JCB Journal of Cell Biology

# SNX10 functions as a modulator of piecemeal mitophagy and mitochondrial bioenergetics

Laura Trachsel-Moncho[1,2,3], Chiara Veroni[1,2,3]*, Benan John Mathai[1,2,3]*, Ana Lapao[1,2], Sakshi Singh[1,2], Nagham Theres Asp[1,2], Sebastian W. Schultz[2,3], Serhiy Pankiv[1,2], and Anne Simonsen[1,2,3]

We here identify the endosomal protein SNX10 as a negative regulator of piecemeal mitophagy of OXPHOS machinery components. In control conditions, SNX10 localizes to early endocytic compartments in a PtdIns3P-dependent manner and modulates endosomal trafficking but also shows dynamic connections with mitochondria. Upon hypoxia-mimicking conditions, SNX10 localizes to late endosomal structures containing selected mitochondrial proteins, including COX-IV and SAMM50, and the autophagy proteins SQSTM1/p62 and LC3B. The turnover of COX-IV was enhanced in SNX10-depleted cells, with a corresponding reduced mitochondrial respiration and citrate synthase activity. Importantly, zebrafish larvae lacking Snx10 show reduced levels of Cox-IV, as well as elevated ROS levels and ROS-mediated cell death in the brain, demonstrating the in vivo relevance of SNX10-mediated modulation of mitochondrial bioenergetics.

## Introduction

Several diseases are characterized by an imbalance between protein production, sorting, and degradation, making it important to understand the cross talk between the pathways involved in regulation of proteostasis, including autophagy and endocytosis (Vagnozzi and Praticò, 2019). While autophagy entails the sequestration of endogenous cytoplasmic material into double-membrane autophagosomes that fuse with lysosomes for cargo degradation (Feng et al., 2014; Melia et al., 2020), endocytosis generally involves lysosomal sorting of exogenous material. The autophagic and endocytic pathways are, however, closely connected, as autophagosomes can fuse with endocytic vesicles before their fusion with lysosomes, and endosomes can contribute membrane to the growing autophagosome (Hyttinen et al., 2013). Moreover, components of the core autophagy machinery have been localized to endosomes, and components of the endosomal sorting complexes required for transport are important for closure of the autophagosome (Rusten and Stenmark, 2009). However, our knowledge about the molecular mechanisms involved in the regulation of the dynamic crosstalk between different types of autophagy and endocytosis remains sparse.

Mitophagy involves the selective degradation of mitochondrial material in lysosomes, preventing the accumulation of dysfunctional mitochondria to mitigate cellular stress. Dysregulation of mitophagy is linked to neurodegenerative diseases,

metabolic disorders, and cancer, and it is therefore important to understand the mechanisms involved in mitophagy. Recently, it has been observed that early and late endosomes can interact with mitochondria (Das et al., 2016; Hamdi et al., 2016; Sheftel et al., 2007; Hammerling et al., 2017; Yamano et al., 2018; Prashar et al., 2024) and that such interactions may play a role in mitochondrial quality control and stress responses (Shutt and McBride, 2013).

The PX domain–containing protein sorting nexin 10 (SNX10) was recently identified in a screen for lipid-binding proteins involved in Parkin-independent mitophagy (Munson et al., 2021). SNX10 is one of the simplest SNX proteins, consisting of a single PX domain that interacts with PtdIns3P (Chandra et al., 2019) and an intrinsically disordered region (IDR) in its C-terminal domain, as elucidated by AlphaFold (Jumper et al., 2021; Varadi et al., 2022). The IDR may account for the diverse cellular roles attributed to SNX10 and its implication in various pathologies (Holehouse and Kragelund, 2024). SNX10 has been found to play a role in cilium biogenesis and promote the localization of vacuolar H+-ATPase subunits and RAB8A to the cilium (Chen et al., 2012). Moreover, several single nucleotide polymorphisms in SNX10 have been linked to autosomal recessive osteopetrosis (ARO) (Pangrazio et al., 2013; Amirfiroozy et al., 2017; Koçak et al., 2019; Stattin et al., 2017), a rare and heterogeneous genetic disease characterized by abnormally

[1]Department of Molecular Medicine, Institute of Basic Medical Sciences Faculty of Medicine, University of Oslo, Oslo, Norway;   [2]Center for Cancer Cell Reprogramming, Institute of Clinical Medicine, Faculty of Medicine, University of Oslo, Oslo, Norway;   [3]Department of Molecular Cell Biology, Institute for Cancer Research, Oslo University Hospital, Oslo, Norway.

*C. Veroni and B.J. Mathai contributed equally to this paper.   Correspondence to Anne Simonsen: anne.simonsen@medisin.uio.no.



dense bone, where dysfunctional or absent osteoclasts fail at performing bone resorption (Pangrazio et al., 2013). Beyond ARO, SNX10 has emerged as a multifaceted player in various pathologies, including gastric cancer, glioblastoma, and colorectal cancer (Deng and Yuan, 2024; Gimple et al., 2023; Feng et al., 2023). Moreover, SNX10 deficiency has been reported to reshape macrophage polarization toward an anti-inflammatory M2 phenotype (You et al., 2016), while its upregulation during bacterial infection enhances phagosome maturation and bacterial killing (Lou et al., 2017). Additionally, SNX10 expression seems to correlate with the severity of Crohn's disease in both human and mouse models, suggesting its potential role in inflammatory bowel diseases (Bao et al., 2023). SNX10 has also been found to influence human adipocyte differentiation and function (Hansen et al., 2023). The involvement of SNX10 in diverse diseases necessitates a deeper exploration of its molecular mechanisms and therapeutic potential.

Here, we show that SNX10 localizes to early and late endocytic compartments and that it modulates the turnover of selected mitochondrial proteins involved in respiration and ATP production, thereby preventing reactive oxygen species (ROS) production and cell death.

## Results

### SNX10 localizes to early and late endocytic compartments in a PtdIns3P-dependent manner

To characterize the mechanisms underlying the normal function of SNX10 and its role in disease development, we generated U2OS cell lines with stable inducible expression of EGFP-tagged WT SNX10 or SNX10 having mutations corresponding to ARO-linked single nucleotide polymorphisms (Y32S, R51P, or R51Q), all located in the PX domain (Elson et al., 2021) of the canonical SNX10 isoform (Fig. 1, A and B; and Fig. S1 A). We observed small cytosolic puncta and larger ring-shaped SNX10-positive structures in cells expressing WT SNX10-EGFP that were absent in cells expressing the ARO-linked mutants, all showing diffuse cytosolic localization (Fig. 1 C). The ARO mutant proteins seem more unstable, as several degradation products not present in the WT cell lysate were detected (Fig. 1 D).

The small SNX10-EGFP–positive puncta co-localized extensively with early endosome antigen 1 (EEA1) and were often seen in very close proximity to larger SNX10-positive ring structures, suggesting these might fuse (Fig. 1 E). The diffuse cytosolic staining of the SNX10 PX mutants (Fig. 1, C and E) indicates that the membrane localization of SNX10 is PtdIns3P dependent. Indeed, when treating cells with the PIK3C3-specific inhibitor VPS34-IN1, both the SNX10-EGFP–positive puncta and the ring structures disappeared, as well as most of the EEA1 staining, as expected since EEA1 has a PtdIns3P-binding FYVE domain (Simonsen et al., 1998) (Fig. 1 E). The SNX10-positive vesicles were also positive for endogenous CD63 (a marker of late endosomes) (Fig. 1 F). Quantification revealed that SNX10-EGFP–positive structures co-localized to a similar extent with EEA1-positive and CD63-positive structures (Fig. 1 G). In line with a role for SNX10 at multiple stages of the endocytic pathway, SNX10-EGFP co-localized with mScarlet-RAB5 (Pankiv et al., 2024) (early

endosomes), mScarlet-RAB7A (Pankiv et al., 2024) (a marker of early to late endosomes), and mScarlet-RAB9A (Pankiv et al., 2024) (involved in endosome-trans Golgi network trafficking) when stably expressed in SNX10-EGFP cells (Fig. S1 B). However, SNX10-EGFP did not co-localize with the recycling endosome markers mScarlet-RAB4 (Pankiv et al., 2024) and mScarlet-RAB11 (Pankiv et al., 2024), with mScarlet-RAB6 (Pankiv et al., 2024) (secretory pathway) or mScarlet-RAB43 (Pankiv et al., 2024) (retrograde transport) (Fig. S1 B).

As previously shown (Qin et al., 2006), expression of SNX10-EGFP led to the formation of giant juxtanuclear vacuoles (Fig. 1 C and Fig. S1 A). Remarkably, these big vacuoles were often negative for SNX10-EGFP and did not form in cells expressing mutant SNX10-EGFP (Y32S, R51P, or R51Q) (Fig. 1 C and Fig. S1 A). The limiting membrane of the large vacuoles stained positive for CD63 (Fig. 1 F), LAMP1 (Fig. 1 H), and RAB7 (Fig. S1 B), and LysoTracker Red stained the vacuole lumen (Fig. S1 C), indicating an acidic lysosomal nature of the large SNX10-induced vacuoles.

To further characterize the nature of SNX10 positive vesicles, SNX10-EGFP cells were processed for correlative light and EM (CLEM) analysis (Fig. 1 I). When focusing on one of the SNX10-EGFP–positive structures, we observed endocytic vesicles containing membranous material that were surrounded by an electron-dense environment of small vesicles (Fig. 1, I iii and iv), where many appeared to be clathrin-coated vesicles (Fig. 1, I iv and v, arrows). Indeed, using confocal imaging, we confirmed that SNX10-EGFP localizes with clathrin-positive structures (Fig. S1 D).

### SNX10 promotes endocytic trafficking

The localization of SNX10 to both early and late endocytic compartments led us to investigate the potential role of SNX10 in endosomal trafficking. U2OS SNX10-EGFP cells were incubated with an antibody against the EGF receptor (EGFR) on ice, followed by EGF-mediated stimulation of EGFR internalization (Fig. 2 A). As expected, SNX10-EGFP puncta showed a clear co-occurrence with endocytosed EGF and with EGFR after EGF stimulation (Fig. 2 B). To assess the potential role of SNX10 in EGFR trafficking, control and SNX10-depleted cells were incubated with EGF for 15 min to activate EGFR signaling and internalization, followed by a chase for up to 120 min. Intriguingly, the level of EGFR and the level of phosphorylated EGFR were higher in SNX10-depleted cells compared with control cells (Fig. 2, C and D), suggesting that SNX10 may affect early endocytic trafficking. In line with such a model, SNX10 silencing with two independent oligos significantly increased the numbers of EEA1-positive puncta (Fig. 2, E and F). Moreover, quantification of endosomes containing immunogold-labeled EGFR in sections from EM images revealed that EGFR-containing endosomes were significantly smaller in SNX10-depleted cells than in control cells (Fig. 2, G and H). Together, our data indicate a role for SNX10 in promoting early endocytic trafficking.

### SNX10-positive endosomes contain mitochondrial material and co-localize with autophagy markers

To further understand the cellular function of SNX10, we set out to identify the interactome of SNX10 and compare it to the

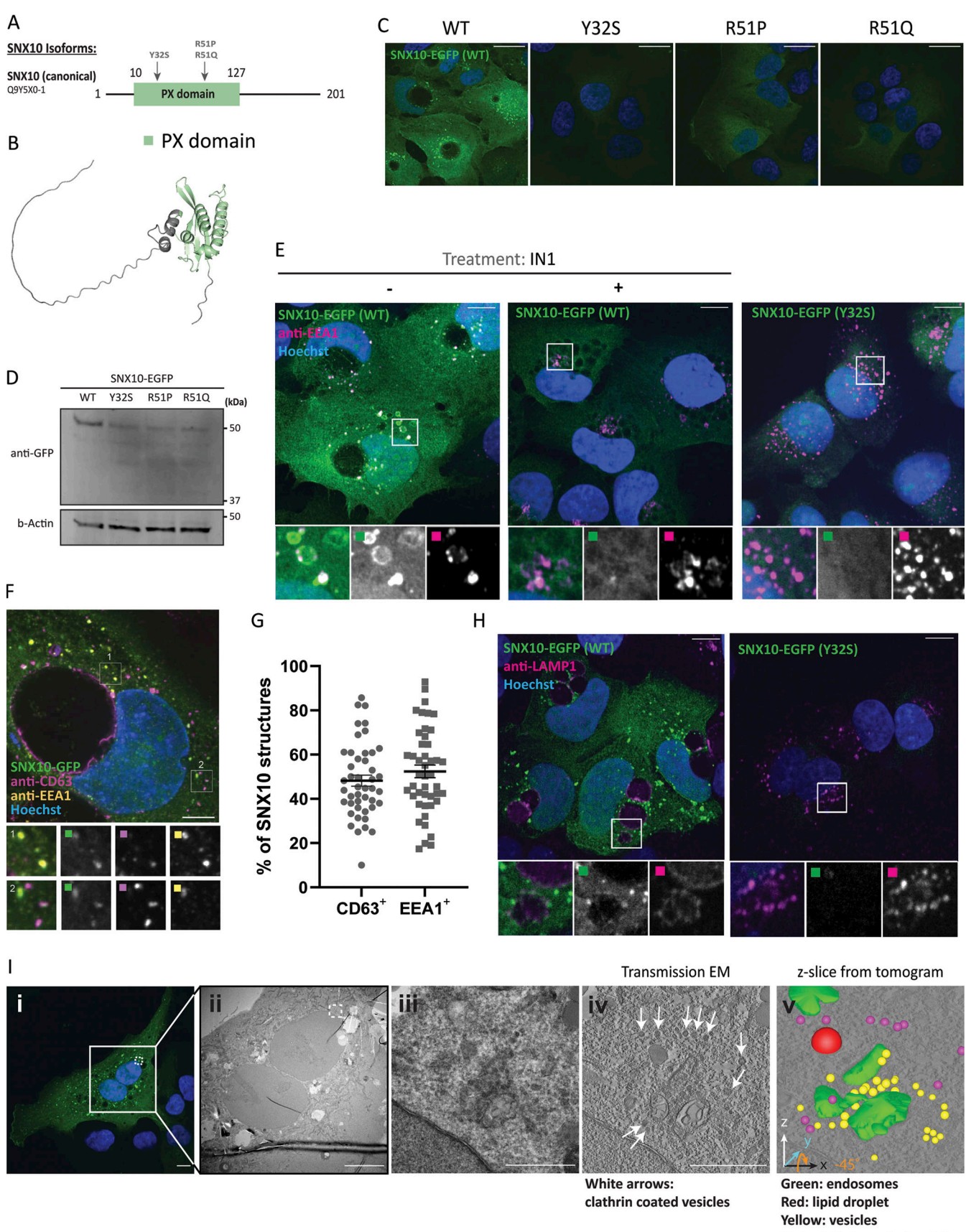

Figure 1.   **SNX10 localizes to early and late endocytic compartments. (A)** Graphical view of the SNX10 isoforms annotated in UniProt. The PX domain is represented in green, and the numbers indicate the number of amino acids. The arrows indicate the position of the natural variants (SNPs) linked to ARO (Y32S,

R51P, and R51Q). **(B)** The figure displays the predicted protein structure of SNX10 generated using AlphaFold, showcasing its three-dimensional conformation. **(C)** Confocal imaging of U2OS cell lines stably expressing doxycycline-inducible SNX10-EGFP WT or the indicated ARO-linked mutants. Nuclei were stained with Hoechst. Scale bar: 20 µm. **(D)** Representative immunoblot showing the expression levels of SNX10-EGFP and the indicated ARO mutants. The membrane was blotted using an anti-GFP antibody and using actin as a loading control. **(E)** Representative immunofluorescence images of U2OS cells stably expressing SNX10-EGFP WT or the Y32S mutant (green) immunostained with anti-EEA1 (magenta) after treating cells with 5 µM VPS34-IN1 for 2 h. Nuclei were stained with Hoechst. Scale bar: 10 µm. Insets: 9.35 × 9.35 µm. **(F)** Representative image of U2OS cells stably expressing SNX10-EGFP and immunostained with anti-CD63 (magenta) and anti-EEA1 (yellow) antibodies. Images were taken with a Nikon CREST X-Light V3 spinning disk microscope using a 60× oil objective (NA 1.42). Scale bar: 10 µm. Insets: 5.12 × 5.12 µm. **(G)** Quantification of F represented as the percentage of SNX10 structures that are either CD63- or EEA1-positive. Data are mean ± SEM with individual data points corresponding to a single field of view (n > 300 cells, four experiments). The significance was assessed by unpaired t test. Data distribution was assumed to be normal, but this was not formally tested. **(H)** Representative immunofluorescence images of U2OS cells stably expressing SNX10-EGFP WT or the Y32S mutant (green) immunostained with anti-LAMP1. Scale bar: 10 µm. Insets: 10.43 × 10.43 µm. **(I)** U2OS SNX10-EGFP cells fixed for CLEM analysis. The area analyzed and shown in (ii) is indicated with a square in the confocal image (i). (iii) Shows the transmission EM and (iv) the z-slide from the tomogram from the white dotted area shown in (ii). The white arrows indicate clathrin-coated vesicles. (v) Green: endosomes; red: lipid droplets; yellow: vesicles; and pink: clathrin-coated vesicles. Scale bars: 10 µm (i and ii), 1 µm (iii and iv). SNPs; single nucleotide polymorphisms. Source data are available for this figure: SourceData F1.

interactome of the ARO mutant SNX10 Y32S that does not bind to membranes. U2OS cells stably expressing SNX10-EGFP WT, SNX10-EGFP Y32S, or EGFP (control) were subjected to GFP-trap pulldown experiments, followed by analysis of their respective interactomes by mass spectrometry. A total of 53 proteins were identified as significant SNX10-EGFP interactors compared with the EGFP control, as analyzed by R lima using a cut-off value of $\log_2$FC >1 and adjusted P value <0.05 (Fig. 3 A). Of these, 29 proteins were specific for WT SNX10 and 24 were common with the Y32S mutant (Fig. 3 A). Gene Ontology term analysis showed that these 53 proteins included proteins with predicted localization (cellular component) to endomembrane compartments (e.g., LAMTOR1, SQSTM1/p62, and MVB12A), mitochondria (e.g., ATP5J, ATPIF1, and COX-IV), endoplasmic reticulum (e.g., SERPINH1 and TMEM109), and the extracellular space (e.g., YBX1 and EEF1G). The 53 proteins were associated with biological processes such as mitochondrial organization (e.g., COX-IV, UQCRFS1, and SLC25A6), autophagy (e.g., LAMTOR1, GBA, and SQSTM1), and endosome organization (e.g., SQSTM1, LAMTOR, and MVB12A) (Fig. 3, A and B).

Given the interaction of SNX10 with mitochondrial (e.g., COX-IV, ATP5J, and ATPIF1), autophagic (e.g., SQSTM1/p62), and endolysosomal (e.g., MVB12A and LAMTOR1) proteins (Fig. 3, A and B), as well as its localization to early and late endosomes (Fig. 1 and Fig. S1), we speculated that SNX10 may have a function in the lysosomal turnover of mitochondrial material. To address this, U2OS SNX10-EGFP cells were labeled with MitoTracker Red and subjected to live-cell imaging. SNX10-EGFP–positive structures were found to localize near the mitochondrial network and move along mitochondria in a highly dynamic manner, with occasional MitoTracker Red signal detected within SNX10-positive vesicles (Fig. 3 C and Video 1). The SNX10 and MitoTracker Deep Red positive vesicles became noticeably bigger upon induction of mitophagy by the hypoxia-mimicking drugs deferiprone (DFP) (Fig. 3, D and E) or dimethyloxalylglycine (DMOG) (Fig. 4), both known to trigger mitophagy in a HIF1a-dependent manner (Allen et al., 2013). These vesicles also stained positive for mScarlet-RAB5 (Fig. 3 D) and the autophagy membrane protein LC3B (Fig. 3 E). The co-occurrence of LC3B-positive structures with SNX10 and with MitoTracker/SNX10-positive puncta was significantly increased in cells treated with DFP for 24 h (Fig. 3, F and G),

indicating a functional shift in SNX10-positive structures upon induction of mitophagy. Indeed, while SNX10 vesicles largely co-localized with EEA1 under control conditions, with minimal co-localization with mitochondria or LC3B (Fig. 4, A and B), SNX10 vesicles showed reduced co-localization with EEA1, along with increased incorporation of mitochondria, LC3B, and CD63 following DFP or DMOG treatment (Fig. 4, C and D). Taken together, our data indicate that SNX10 primarily localizes to early endosomes in control conditions and to mitochondria-containing LC3B and CD63-positive structures under hypoxia-mimicking conditions.

## SNX10 modulates mitochondrial protein degradation

To decipher the nature of the mitochondrial cargo included in SNX10-positive vesicles, SNX10-EGFP cells were fixed and stained with antibodies against distinct mitochondrial proteins. Notably, the incorporation of ATP5J, SAMM50, and COX-IV into SNX10-EGFP vesicles was evident upon DFP treatment (Fig. 5 A), while TOMM20, TIMM23, and PDH were rarely detected (Fig. S2 A), although their mitochondrial staining was clear. SNX10-positive vesicles containing COX-IV also stained positive for endogenous LC3B (Fig. 5, B and C) and LAMP1 (Fig. 5, D and E) both in cells treated with DFP and DMOG (Fig. 5, B–E), suggesting a role for SNX10-positive vesicles in mitophagy.

To investigate whether SNX10 might regulate the turnover of mitochondrial proteins, we assessed the abundance of selected mitochondrial proteins in cells transfected with control or SNX10 siRNA that were treated or not with DFP for 24 h (Fig. 6 A). Interestingly, SNX10 depletion seems to reduce the abundance of several mitochondrial proteins, both at basal levels and upon mitophagy induction, with a significant effect on COX-IV levels (Fig. 6, A–D). Immunofluorescence staining for COX-IV also revealed a significant reduction of the staining intensity in cells depleted of SNX10, which was further reduced in DFP-treated cells (Fig. 6, E and F). In contrast, the level of the outer mitochondrial membrane protein TOMM20 (Fig. S2, B and C) was unaffected by the depletion of SNX10. The reduced level of COX-IV seen in SNX10-depleted cells was not due to changes in COX-IV transcription (Fig. S2 D) or proteasomal degradation of COX-IV (Fig. S2 E), suggesting a role for SNX10 in the lysosomal clearance of selected mitochondrial proteins.

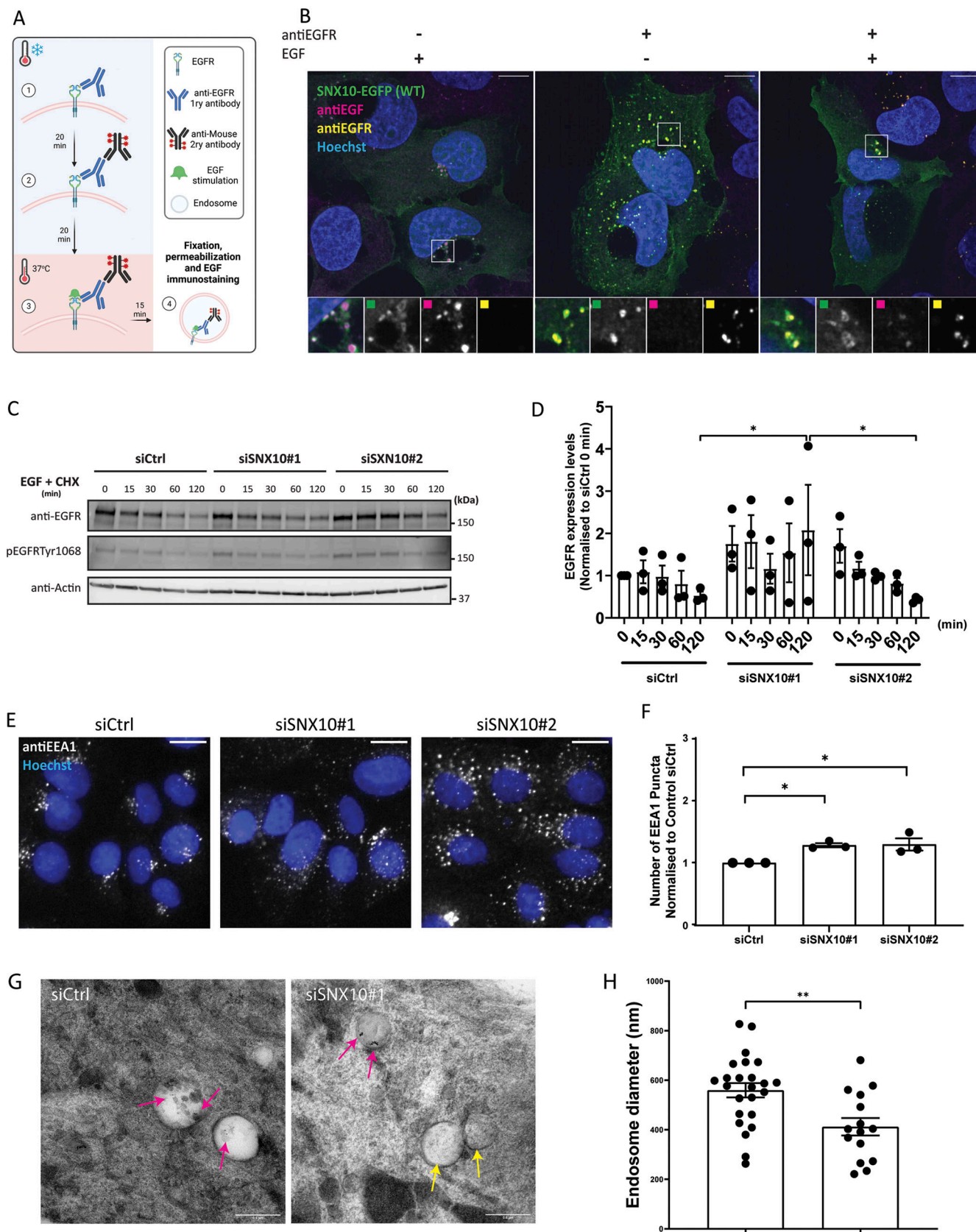

**Figure 2.** **SNX10 regulates endocytic trafficking. (A)** Graphical description of the plasma membrane EGFR staining. Live cells are put on ice and (1) incubated for 20 min with the primary anti-EGFR antibody, then washed and (2) incubated with a secondary antibody for 20 min, followed by (3) incubation with EGF for 15 or 50 min at 37°C before fixation and imaging. **(B)** U2OS SNX10-EGFP cells were incubated with anti-EGFR antibody as described in A, then

stimulated with EGF and fixed. Cells were stained with an anti-EFG antibody after fixation. Scale bar: 10 µm. Insets: 7.24 × 7.24 µm. **(C)** After 72 h of siRNA transfection with siCtrl (control) or two different siSNX10 oligoes (siSNX10#1 and siSNX10#2), U2OS cells were serum starved for 2 h and then incubated with 50 ng/ml EGF + 10 µg/ml cycloheximide (CHX) for the indicated times. The cells were lysed, followed by western blotting for the indicated proteins. **(D)** Quantification of EGFR protein levels normalized to actin in $n$ = 3 independent experiments ± SEM. Significance was determined by two-way ANOVA followed by Tukey's multiple comparisons test. Normality was assumed but not formally tested. **(E)** Cells were transfected with siCtrl, siSNX10#1 or siSNX10#2 prior to fixation and staining for endogenous EEA1. Images were taken with Zeiss Axio Observer widefield microscope (Zen Blue 2.3; Zeiss), and a 20× objective was used. Scale bar: 10 µm. **(F)** Quantification of the data shown in E was performed using CellProfiler software. The values were obtained from analyzing >1,000 cells per condition, and they were normalized to control siRNA (siCtrl). The graphs display the mean values ± SEM from $n$ = 3 independent experiments. The significance was assessed by ordinary one-way ANOVA followed by Bonferroni's post hoc test. Data distribution was assumed to be normal but was not formally tested. **(G)** Representative EM images of endosomes in U2OS cells (control and siSNX10 #1). Pink arrows: Protein A conjugated with 10 nm gold (PAG10)-labeling EGFR that has been taken up into endosomes. Yellow arrows: endosome not containing internalized PAG10-labeled EGFR. Scale bar: 0.5 µm. **(H)** Measurements of EGFR-containing endosome diameter in control versus siSNX10-treated cells from one experiment. The graph shows the endosomal diameter (nm) of a total 24 PAG10-labeled EGFR endosomes in siCtrl cells and 15 PAG10-labeled EGFR endosomes in siSNX10 cells. The graph displays the mean values ± SEM. Significance was determined by unpaired $t$ test with Welch's correction in all graphs, and data distribution was assumed to be normal but was not formally tested. * = P < 0.05 and ** = P < 0.01, nonsignificant differences are not depicted. Source data are available for this figure: SourceData F2.

## SNX10 is a negative modulator of piecemeal mitophagy

Given the co-localization of SNX10 with LC3B, we speculated that SNX10 may play a role in mitophagy. To address this, we used two different stable mitophagy reporter U2OS cell lines, expressing the mitochondrial targeting signal of the inner mitochondrial membrane (IMM) protein pSu9 fused Halo-mGFP (pSu9-Halo-mGFP) (Yim et al., 2022), or the mitochondrial targeting signal of the matrix protein NIPSNAP1 fused to EGFP-mCherry (referred to as iMLS) (Princely Abudu et al., 2019). The pSu9-Halo-mGFP cells allow a measure of mitophagy flux upon mitophagy induction with DFP or DMOG, as the cleaved Halo tag is stable in lysosomes when bound to the ligand (Fig. 7, A and B). The ratio of cleaved versus full-length pSu9-Halo-mGFP was analyzed by western blotting (relative to a loading control), demonstrating a significant increase in mitophagy under both DFP and DMOG conditions in cells depleted of SNX10 compared with control cells (Fig. 7, A and C). In contrast, SNX10 depletion did not affect the lysosomal transport of the mCherry-EGFP–tagged matrix reporter, as analyzed by the area of red only puncta (representing mitolysosomes due to quenching of the EGFP signal in acidic lysosomes) (Fig. S3, A and B). Thus, our data indicate a role for SNX10 as a negative regulator of lysosomal turnover of selected mitochondrial membrane components in response to HIF1 activation.

In addition to macromitophagy, other quality control pathways can facilitate the disposal of mitochondrial proteins for lysosomal degradation, including piecemeal mitophagy (Le Guerroué et al., 2017; Abudu et al., 2021), mitochondria-derived vesicles (MDVs) (Soubannier et al., 2012), and vesicles derived from the IMM (VDIM) (Prashar et al., 2024). Besides co-localizing with LC3B, SNX10 vesicles containing mitochondrial material were found to co-localize with p62 (Fig. 7, D), which argues against a role for SNX10 in the MDV or VDIM pathways, as these are negative for LC3B and p62 and independent of the core autophagy machinery (Soubannier et al., 2012; Prashar et al., 2024). Indeed, the depletion of SNX10 neither affected the number of PDH⁺TOMM20⁻ nor PDH⁻TOMM20⁺ MDVs (Fig. S3, C and D).

Piecemeal mitophagy targets selected mitochondrial proteins to lysosomes, including components of the sorting and assembly machinery complex and the mitochondrial contact site and cristae organizing system complex, and has been found to rely on p62 and the LC3 conjugation machinery proteins (Le Guerroué et al., 2017; Abudu et al., 2021). Indeed, COX-IV levels were significantly increased in p62-depleted cells (Fig. 7, E and F). Interestingly, neither the DFP-induced COX-IV clearance nor the reduced COX-IV levels seen in SNX10-depleted cells were recovered in cells treated with inhibitors of VPS34 (IN1 [Bago et al., 2014]) or ULK1 (MRT [Petherick et al., 2015]) (Fig. 7, G–I), both kinases being important for macroautophagy (Feng et al., 2014), but it is yet unknown whether they are required for piecemeal mitophagy. In line with this, the DFP-induced co-occurrence of LC3B with SNX10 and SNX10/MitoTracker-positive structures was not affected in cells treated with the ULK1 inhibitor compared with control cells (Fig. 3, E–G). Taken together, our data indicate that SNX10 modulation of COX-IV turnover is independent of canonical macroautophagy.

## SNX10 loss decreases mitochondrial bioenergetics

Given the interaction of SNX10 with ATP5J, a subunit of the mitochondrial ATP synthase (Fig. 3 A), and the increased degradation of the pSu9-Halo-mGFP reporter (Fig. 7 A), and key electron transport chain proteins (COX-IV) (Fig. 6) in SNX10-depleted cells, we next investigated whether SNX10 depletion affects mitochondrial bioenergetics. Indeed, both the baseline oxygen consumption rate (an indicator of mitochondrial respiration) and the ATP production-linked respiration were decreased in U2OS cells lacking SNX10 compared with control cells, as analyzed by a Seahorse XF Analyzer (Fig. 8, A–C).

Interestingly, also the activity of citrate synthase (CS), an enzyme that catalyzes the first reaction of the Krebs cycle in the mitochondrial matrix, was decreased in SNX10-depleted cells compared with control cells, both under basal conditions and upon treatment with DFP for 24 h (Fig. 8 D). The level of CS is often used as a read-out for the total mitochondrial level (Shepherd and Garland, 1969). However, the reduction in CS activity did not correlate with reduced CS levels as determined by western blotting (Fig. 8, E and F), suggesting a lower activity of the Krebs cycle and further demonstrating the selectivity of SNX10 in the degradation of mitochondrial proteins. Taken together, our data indicate that SNX10 is important for normal mitochondrial bioenergetics.

## Snx10 is partially conserved in zebrafish and expressed during early embryogenesis

To validate the potential role of SNX10 in piecemeal mitophagy in vivo, we employed zebrafish as a model system. Zebrafish

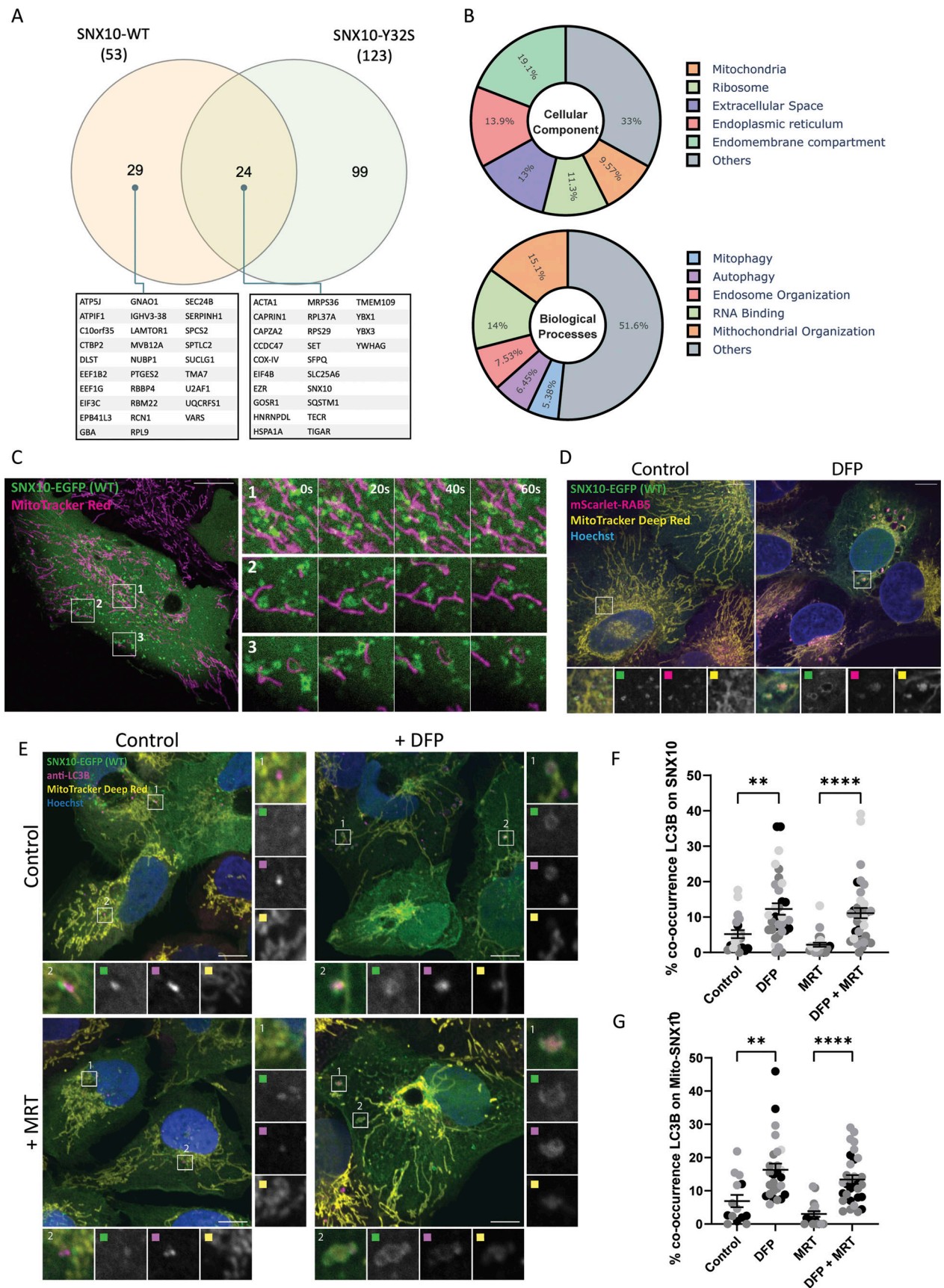

Figure 3. **SNX10 localizes nearby mitochondria. (A)** SNX10-EGFP WT or the Y32S mutant, stably expressed in U2OS cells, underwent GFP pulldown for the subsequent analysis of their interactome using mass spectrometry assays. A total of 53 proteins were identified as significant SNX10-EGFP interactors

compared with the EGFP control, as analyzed by R lima using a cut-off value of log$_2$FC >1 and adjusted P value <0.05. The Venn diagram illustrates the distinct and shared interactors between the SNX10 WT and Y32S mutant, where the identified significant SNX10-EGFP–interacting proteins are listed below. **(B)** The interacting proteins of SNX10 WT were enriched for GO term analysis. Their cellular component (upper pie chart) and biological processes (lower pie chart) enrichment is expressed in percentage toward the significant hits. Graphs were plotted using Plotly (Python package). **(C)** U2OS with stable inducible expression of SNX10-EGFP were treated with doxycycline for 24 h before incubation with MitoTracker Red for 30 min, followed by live imaging with an acquisition speed of 1 frame every 500 ms. Scale bar: 10 μm. **(D)** U2OS cells stably expressing mScarlet-RAB5 and with inducible expression of SNX10-EGFP were stained with MitoTracker Deep Red FM in the presence or absence of DFP (1 μM). Scale bar: 10 μm. **(E)** U2OS SNX10-EGFP cells were treated or not with DFP (1 μM) for 24 h in the absence or presence of the ULK1 inhibitor MRT68921 (1 μM) for 1 h. MitoTracker Deep Red FM (100 nM) was added for 1 h, followed by immunofluorescence staining with antibody against LC3B. Scale bars: 10 μm. Insets 5.77 × 5.77 μm. **(F and G)** Quantification of the percentage of the co-occurrence of LC3 on SNX10 structures (F) and of LC3 on MitroTracker-SNX10–positive structures (G). Data are mean ± SEM with individual data points corresponding to a single field of view ($n$ = 4 [F] and $n$ = 3 [G], corresponding experiment shown in the same color, >150 cells per experiment). The statistical significance was calculated with ordinary one-way ANOVA, followed by Tukey's multiple comparison test. Data distribution was assumed to be normal, but this was not formally tested. * = $P$ < 0.05, ** = $P$ < 0.01, *** = $P$ < 0.001, and **** = $P$ < 0.0001; nonsignificant differences are not depicted. GO; Gene Ontology.

contain SNX10 paralogues (*snx10a* and *snx10b*) that are ~60% similar to one another and ~59% homologous to human SNX10, with the N-terminal PX domain being the most highly conserved (~71% and ~63.5%, respectively) (Fig. 9 A). We first sought to examine the spatiotemporal expression pattern of *snx10a* and *snx10b* during the early development of zebrafish. While both *snx10a* and *snx10b* were consistently expressed from 1 day postfertilization (dpf) to 5 dpf as determined by real-time qPCR, *snx10a* transcripts were more abundant than *snx10b* transcripts (Fig. 9 B), suggesting the importance of *snx10a* in the early development of the zebrafish larvae. Whole-mount in situ hybridization (WM-ISH) of *snx10a* and *snx10b* transcripts in 2–5 dpf WT zebrafish larvae revealed expression of *snx10a* in the retina, optical tectum, intestine, midbrain, and hindbrain at all time points, whereas *snx10b* transcripts were found mainly in the retina and notochord at 2–5 dpf and in the whole brain at 5 dpf (Fig. 9 C and Fig. S4 A). There was no staining of the control sense probe at any time points.

### Increased turnover of mitochondrial proteins in zebrafish larvae lacking Snx10

To explore a possible role for Snx10-regulated mitophagy in the early development of zebrafish larvae, we employed CRISPR/Cas9-mediated genome editing in zebrafish embryos using guide sequences targeting *snx10a* and *snx10b* individually or together (*snx10ab* double knockout [DKO]) (Fig. S4 B). The loss of Snx10 protein in DKO larvae compared with larvae injected with control oligo was verified via immunoblotting (Fig. 9 D).

Control and *snx10ab* DKO larvae were treated or not with 100 μm DMOG for 24 h at 2 dpf to induce mitophagy (Munson et al., 2021), as demonstrated by a significant upregulation of *bnip3* mRNA levels (Fig. S4 C). In line with our data obtained in SNX10-depleted U2OS cells, the Cox-IV protein level was significantly reduced in *snx10ab* DKO larvae when compared with controls (Fig. 9, E and F). Samm50 was also reduced in *snx10ab* DKO larvae at the basal level and upon DMOG treatment (Fig. 9, E–G), but neither *cox-IV* nor *samm50* mRNA levels were significantly reduced (Fig. S4 D), indicating a role for snx10 in the turnover of zebrafish Cox-IV and Samm50 proteins.

### Snx10 affects cell death and oxidative stress in zebrafish

Most *snx10ab* DKO larvae showed a dysmorphic phenotype (data not shown) with necrotic tissue damage in the head and trunk region, suggesting possible cell death. To further investigate

this, we performed TUNEL staining on sections of fixed control and *snx10ab* DKO larvae, treated with or without DMOG, followed by confocal microscopy and quantification of the mean fluorescent intensity in the brain (region of interest, indicated) (Fig. 9, H and I), having high *snx10* expression (Fig. 9 C). TUNEL staining showed increased cell death in *snx10ab* DKO larvae relative to control (Fig. 9, H and I). Interestingly, DMOG-treated control larvae also showed increased cell death as compared with control larvae (Fig. 9, H and I), which may suggest that increased turnover of mitochondria, caused by snx10 depletion or DMOG treatment, promotes cell death. To assess whether the increased cell death seen in *snx10ab* DKO larvae could be attributed to increased oxidative stress, control and *snx10ab* DKO zebrafish larvae were trypsinized and treated with deep red MITOSOX dye before FACS analysis. Indeed, there was a significant increase in ROS levels in *snx10ab* DKO larval cells as compared with the control (Fig. 9 J and Fig. S4 E). Importantly, the increased TUNEL staining seen in *snx10ab* DKO larvae could be rescued with the addition of the antioxidant N-acetyl cysteine (Fig. 9, K and L). Taken together, our in vivo data show an important role for snx10 in preventing cell death and oxidative stress.

## Discussion

In this manuscript, we identify the small PX domain protein SNX10 as a novel modulator of piecemeal mitophagy and mitochondrial bioenergetics. SNX10 localizes to early and late endocytic compartments in a PtdIns3P-dependent manner and modulates trafficking in the endosomal pathway. Upon hypoxia, SNX10 localizes to CD63-positive late endosomes containing mitochondrial material and autophagy proteins. We demonstrate that SNX10 functions as a negative regulator of piecemeal mitophagy of OXPHOS machinery components (including COX-IV) in response to hypoxia, allowing the cell to regulate oxidative phosphorylation in response to metabolic needs without compromising overall mitochondrial structure. Zebrafish larvae lacking Snx10 are characterized by increased ROS and ROS-induced cell death, demonstrating an important role for Snx10 in vivo (Fig. 10).

Mutations in SNX10 are linked to ARO (Pangrazio et al., 2013; Amirfiroozy et al., 2017; Koçak et al., 2019; Stattin et al., 2017), a life-threatening rare type of skeletal dysplasia characterized by increased bone density. All described ARO-linked mutations in

Figure 4. **SNX10 structures containing mitochondria mature into late endosomes. (A and C)** U2OS cells with inducible expression of SNX10-EGFP were treated with DFP (1 µM) or DMOG (1 µM) for 24 h, stained with MitoTracker Deep Red FM (100 nM) for 30 min, followed by immunofluorescence with

antibodies anti-LC3B and anti-EEA1 (A) or anti-CD63 (C), prior to acquisition with a Nikon CREST X-Light V3 spinning disk microscope using a 60× oil objective (NA 1.42). Scale bar: 10 µm. Insets: 5.52 × 5.52 µm (A), 6.62 × 6.62 µm (C). **(B–D)** Pixel intensity plots for line in control, DFP, and DMOG insets, respectively for A and C.

SNX10 are found in the PX domain (Amirfiroozy et al., 2017; Koçak et al., 2019; Stattin et al., 2017), and we here show that three such mutants (Y32S, R51P, and R51Q) fail to localize to endocytic compartments. The exact endosomal function of SNX10 is unclear, but the presence of enlarged LAMP1-positive vacuoles in cells expressing WT SNX10, but not ARO mutants, suggests that SNX10 may regulate fission/fusion events in the endocytic pathway. This is in line with previous data showing that mutation of the predicted Ptdns3P-binding residue of SNX10 (Arg/R[53]) prevents the formation of large vacuoles (Qin et al., 2006), suggesting that PtdIns3P-mediated endosome recruitment of SNX10 is important for its normal function and prevention of ARO. SNX10 is the smallest of the PX domain proteins, containing only a PX domain plus a short IDR, generally known for their dynamic and adaptable nature, existing in a repertoire of structurally distinct conformations that rapidly interconvert based on their cellular context (Holehouse and Kragelund, 2024). SNX10 pulldown experiments demonstrated that the ARO mutant (Y23S) interacted with more proteins than the WT SNX10, so it is also possible that SNX10 PX mutants may cause ARO through its C-terminal IDR.

Mitochondria are crucial for the regulation of nutrient metabolism and maintenance of bone homeostasis. Currently, a potential association between ARO and mitochondrial dysfunction is not known. However, mitochondrial dysfunction, including impaired mitochondrial autophagy and OXPHOS activity, as well as ROS accumulation, has been linked to osteoporosis (Liu et al., 2024).

We found that SNX10 interacts with proteins belonging to the mitochondrial cell compartment and that depletion of SNX10 promotes the degradation of selected mitochondrial proteins, including the IMM protein COX-IV and the pSu9-Halo-EGFP reporter, with no effect on a mCherry-EGFP–tagged matrix marker or the total mitochondrial abundance. Cells lacking SNX10 also had decreased OXPHOS activity. These results are in line with previous reports showing increased glucose metabolism upon SNX10 depletion in intestinal epithelial cells (Feng et al., 2023) and increased levels of LAMP2A (You et al., 2018; Lee et al., 2022).

Our data indicate that SNX10 functions as a negative modulator of piecemeal mitophagy of COX-IV. Using hypoxia-mimicking drugs (DFP and DMOG), we observed staining of COX-IV within SNX10-positive vesicles that co-localized with markers of late endosomes (CD63) and lysosomes (LAMP1), as well as the autophagy proteins LC3B and p62/SQSTM1. Intriguingly, neither the DFP- nor the siSNX10-induced degradation of COX-IV nor the co-localization of SNX10 with LC3B was reduced by inhibition of the core autophagy machinery components ULK1 or VPS34, suggesting a noncanonical type of mitophagy. It is not known whether piecemeal mitophagy depends on ULK1 or VPS34, but it has been shown to rely on p62/SQSTM1 (Le Guerroué et al., 2017;

Abudu et al., 2021). Indeed, we show that p62 co-localizes with COX-IV and that depletion of p62 leads to increased COX-IV levels. The seemingly selective effect of SNX10 on the turnover of proteins involved in mitochondrial oxidative phosphorylation and ATP production is reminiscent of the recently described VDIM pathway (Prashar et al., 2024). VDIMs are formed by IMM herniation through pores in the outer mitochondrial membrane, followed by their engulfment by lysosomes in proximity to mitochondria in a microautophagy-like process. VDIM formation was found to increase upon oxidative stress, leading to selective degradation of IMM proteins (including COX4 and other proteins involved in OXPHOS) while sparing the remainder of the organelle. However, in contrast to the observed MitoTracker/COX-IV–containing SNX10-positive vesicles, VDIMs seem to lack LC3 and p62. We also show that MDVs form independently of SNX10. Thus, we conclude that SNX10 functions as a negative modulator of piecemeal mitophagy of COX-IV upon hypoxia. We speculate that the SNX10 interactome may change in response to various metabolic conditions, thereby modulating the lysosomal transport of endocytic and mitochondrial cargo to allow the cell to respond to the metabolic needs of the cells.

Importantly, the depletion of Snx10 in zebrafish larvae also caused a reduction of Cox-IV and Samm50 protein levels, with a corresponding increase in ROS, indicating a conserved role of Snx10 in vivo as a modulator of mitochondrial homeostasis. Elevation of cell death in *snx10ab*-depleted fish could be attributed to ROS-mediated cell death as it was rescued by antioxidant treatment. While decreased levels of mitophagy previously have been linked to increased ROS and cell death (Baechler et al., 2019; Lin et al., 2019), our data indicate that also increased turnover of selected mitochondrial proteins can cause ROS and cell death in vivo. In line with such a model, induction of mitophagy using the hypoxia-mimetic drug DMOG in control larvae significantly increased cell death, suggesting that excess mitophagy may trigger cell death (Ma et al., 2024). Dynamic interactions between mitochondria and endolysosomes have also been linked to pathogen killing by the transfer of mitochondrial ROS to phagosomes containing bacteria (Abuaita et al., 2018). It is interesting to note that SNX10 has been related to pathologies connected to monocyte/macrophage malfunctioning (You et al., 2016; Lou et al., 2017).

In conclusion, we here present data showing that SNX10 modulates piecemeal mitophagy of selective OXPHOS machinery proteins under hypoxia-mimicking conditions, allowing cells to dispose of surplus mitochondrial components without affecting the larger mitochondrial structures. Lack of SNX10 leads to an accumulation of ROS and ROS-induced cell death, uncovering a previously unknown role of SNX10 in mitochondrial homeostasis. This unexpected association opens avenues for further exploring the intricate interplay between endocytic and mitochondrial pathways, emphasizing SNX10's crucial role

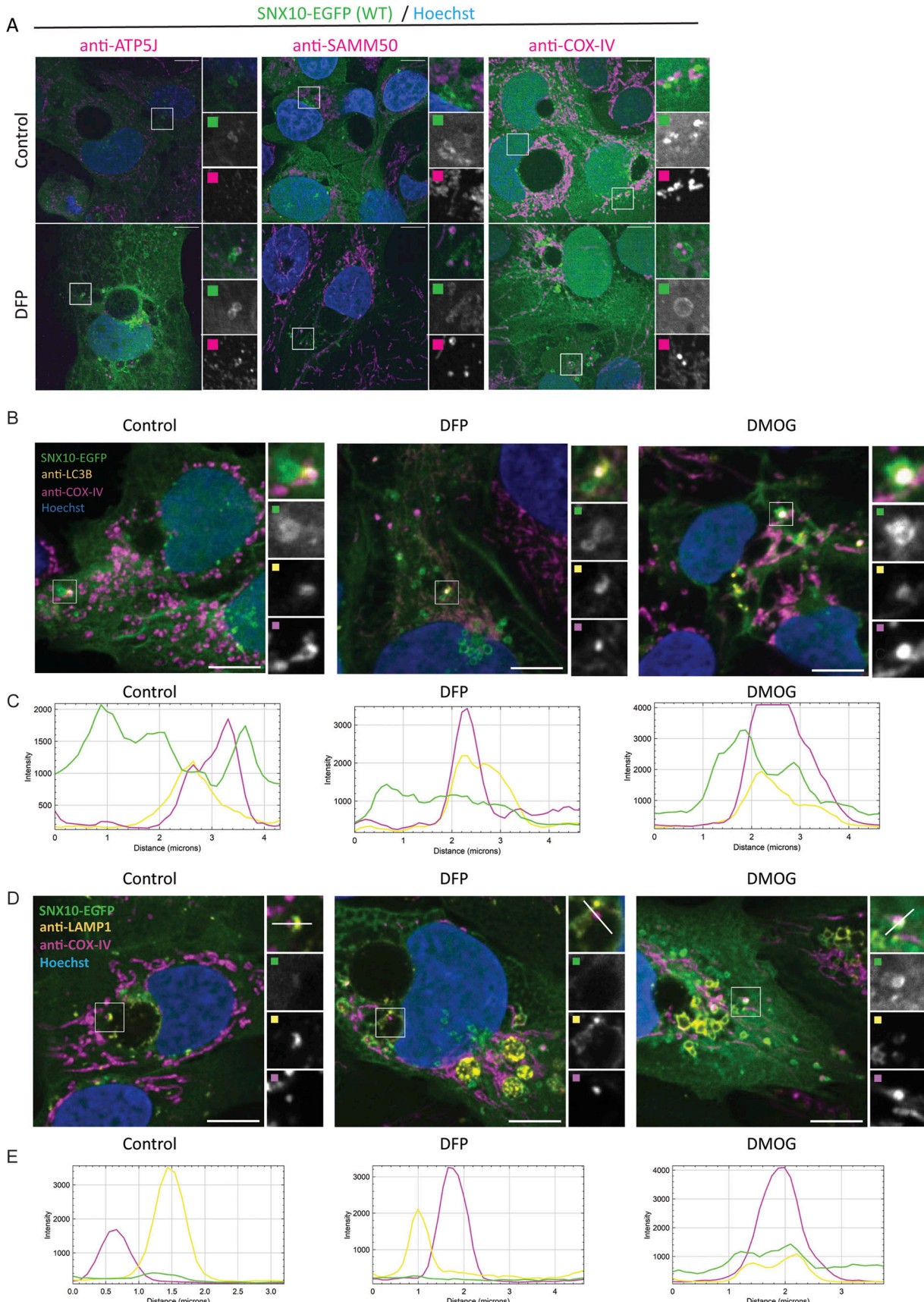

Figure 5. **SNX10 vesicles contain mitochondrial proteins and LC3B. (A)** U2OS cells with stable inducible expression of SNX10-EGFP were pre-treated with doxycycline for 16 h before the addition of DFP (1 μM) for 24 h. The cells were fixed and stained with antibodies against mitochondrial proteins. Scale bars: 10

µm. Insets: 8.57 × 8.57 µm. **(B and D)** U2OS cells with inducible expression of SNX10-EGFP were treated with DFP (1 µM) or DMOG (1 µM) for 24 h and stained with antibodies anti–COX-IV and anti-LC3B in B or anti-LAMP1 in D, prior to acquisition with a Nikon CREST X-Light V3 spinning disk microscope using a 60× oil objective (NA 1.42). Scale bar: 10 µm. Insets: 4.41 × 4.41 µm (B), 5.52 × 5.52 µm (D). **(C–E)** Pixel intensity plots for line in control, DFP, and DMOG insets, respectively for (B and D).

in cellular homeostasis and providing a foundation for future investigations in both physiological and pathological contexts.

## Materials and methods

### Antibodies and dyes
Primary antibodies used: Anti-HaloTag (G9211, 1:1,000, ms; Promega), ATP5J (HPA031069, 1:200, rb; Atlas Antibodies), β-Actin (#3700, 1:5,000, ms; Cell Signaling Technology), CD63 (#H5C6, 1:200, ms; DHB) Citrate Synthase (#14309, 1:1,000, rb; Cell Signaling Technology), Clathrin (#SM5011P, 1:200; Acris), COX-IV (#4850S, 1:1,000 for WB and 1:200 for IF, rb; Cell Signaling Technology), EEA1 (#610457; 1:250, ms; BD Biosciences), GAPDH (Ab9484, 1:5,000, ms; Abcam), GAPDH (#5174, 1:1,000, rb; Cell Signaling), GFP (# 632569, 1:1,000, ms; Takara) EGFR (#20-ES04, 1:1,000 for WB and 1:200 for IF, sh; Fitzgerald), P-EGFR (Tyr1068) (#3777, 1:1,000, rb; Cell Signaling), PDH (#2784S, 1:1,000 for WB and 1:200 for IF, rb; Cell Signaling Technology), LAMP1 (#sc-20011, 1:250, ms; Santa Cruz Biotechnology), LC3B (#PM036, 1:250, rb; MBL), SAMM50 (NBP1–84509, 1:200, rb; Novus Biologicals), SNX10 (HPA015605, 1:1,000, rb; Atlas Antibodies), alpha Tubulin (GTX628802, 1:5,000, ms; GeneTex), TIMM23 (# 611223, 1:1,000 for WB 1:250 for IF, ms; BD Biosciences), and TOMM20 (#17764, 1:200, ms; Santa Cruz).

Secondary antibodies used for immunoblotting: DyLight800 mouse and rabbit (#SA5–10172 and #SA5–10044; Invitrogen), StarBright Blue 700 mouse and rabbit (#12004158 and #12004161; Bio-Rad), Peroxidase AffiniPure Goat Anti-Mouse IgG (H+L) (#115035003, #1:5,000; Jackson), and Peroxidase AffiniPure Goat Anti-Rabbit IgG (H+L) (#111035144, 1:5,000; Jackson). Fluorescent dyes used include MitoTracker Deep Red FM (#M22425; Thermo Fisher Scientific), MitoTracker Red CMXRos (#M7512; Thermo Fisher Scientific), Lyso-Tracker Red DND-99 (# L7528; Thermo Fisher Scientific), and Hoechst 33342 (#H1399; Invitrogen).

### Cell culture and reagents
U2OS TRex FlpIn cells (kindly provided by Steve Blacklow, Harvard Medical School, Boston, MA, USA) were used for the generation of stable inducible cell lines. They were maintained in a complete medium of DMEM (12–741F; Lonza) supplemented with 10% FBS (#F7524; Sigma-Aldrich), 100 U/ml penicillin, and 100 µg/ml streptomycin (#15140122; Thermo Fisher Scientific). Cells were kept in a humidified incubator at 37°C and 5% $CO_2$. For starvation conditions, the cells were incubated for 2–4 h in Earle's balanced salt solution (# 24010043; Gibco). Bafilomycin A1 (#CM110; BML) was used at 100 nM. DFP (#379409; Sigma-Aldrich) was used at 1 mM. Doxycycline (#631311; Clontech) was used at 100 ng/ml. EGF VPS34-IN1 (# S7980; Selleckchem) was used at 5 µM. Tetramethylrhodamine-conjugated ligand (G8251; Promega) was used at 100 nM for 20 min.

### Cell transfection
For the generation of stable cell lines, U2OS FlpIn TRex cells were transfected with pcDNA5/FRT/TO-SNX10-EGFP WT or mutant constructs (Table S1) using Lipofectamine 2000 (#11668019; Invitrogen). Positively transfected cells were selected with hygromycin B (#400052; EMD Millipore). Doxycycline was used at 100 ng/ml for 24 h to induce protein expression. Cell lines expressing mScarlet-tagged RAB proteins were created with lentiviral transduction. Briefly, 293FT cells were co-transfected with psPAX2 and pCMV-VSVG plasmids to generate lentiviral particles. These particles were transduced, after harvesting and filtering, into the SNX10-EGFP stable cell line together with 10 µg/ml polybrene (sc-134220; Santa Cruz) and selected using puromycin (#P7255; Sigma-Aldrich). The plasmids used for the cell lines creation are compiled in Table S1. Point mutations were introduced by using the QuickChange site-directed mutagenesis kit (Agilent Technologies) and the primers used for this are listed in Table S1.

For siRNA transfections, cells were incubated with Silencer Select siRNA oligonucleotides targeting each gene at 20 nM, in combination with RNAiMAX (#13778150; Invitrogen) through reverse transfection. The cells were fixed/harvested for further procedures after 72 h of knockdown.

### CS assay
Biochemical quantification of mitochondrial abundance was done by analysis of CS activity. Briefly, U2OS cells were plated and grown as described in the figure legend. The cells were harvested after washing twice with PBS on ice and then lysed in NP-40 lysis buffer (50 mM HEPES, pH 7.4, 150 mM NaCl, 1 mM EDTA, and 10% [vol/vol] NP-40 + 1 mM DTT; it also contained 1x phosphatase inhibitors and 1x protease inhibitors added fresh). Cell lysates were centrifuged at 21,000 × $g$ for 10 min at 4°C, and the supernatant was obtained. The protein concentration was quantified by Bradford assay. To determine the CS activity, 0.4 µl of cell lysate was added to 198 µl of CS assay buffer (100 mM Tris, pH 8, 0.1% Triton X-100, 0.2 mM 5,5'dithiobis [2-nitrobenzoic acid], and 0.1 mM acetyl CoA). At the starting point, 2 µl of 20 mM oxaloacetate was added, and the plate was incubated at 35°C. The reactions were monitored in a FLUOstar OPTIMA plate reader (BMG Labtech) at $\lambda_{Abs}$ = 440 nm. The reactions were scanned each 87 s for a total of 40 cycles. The values were compared with the oxaloacetate-lacking controls. The rate of the reactions was plotted to determine the CS activity, and the activity rate was calculated using the linear part of the curve before saturation was reached. Then the values were normalized to protein concentration and then normalized to the control value. The graph displays relative values.

### Immunofluorescence
The cells were seeded onto glass coverslips or 96-well plates for high-throughput imaging. siRNA treatment lasted for 72 h, and

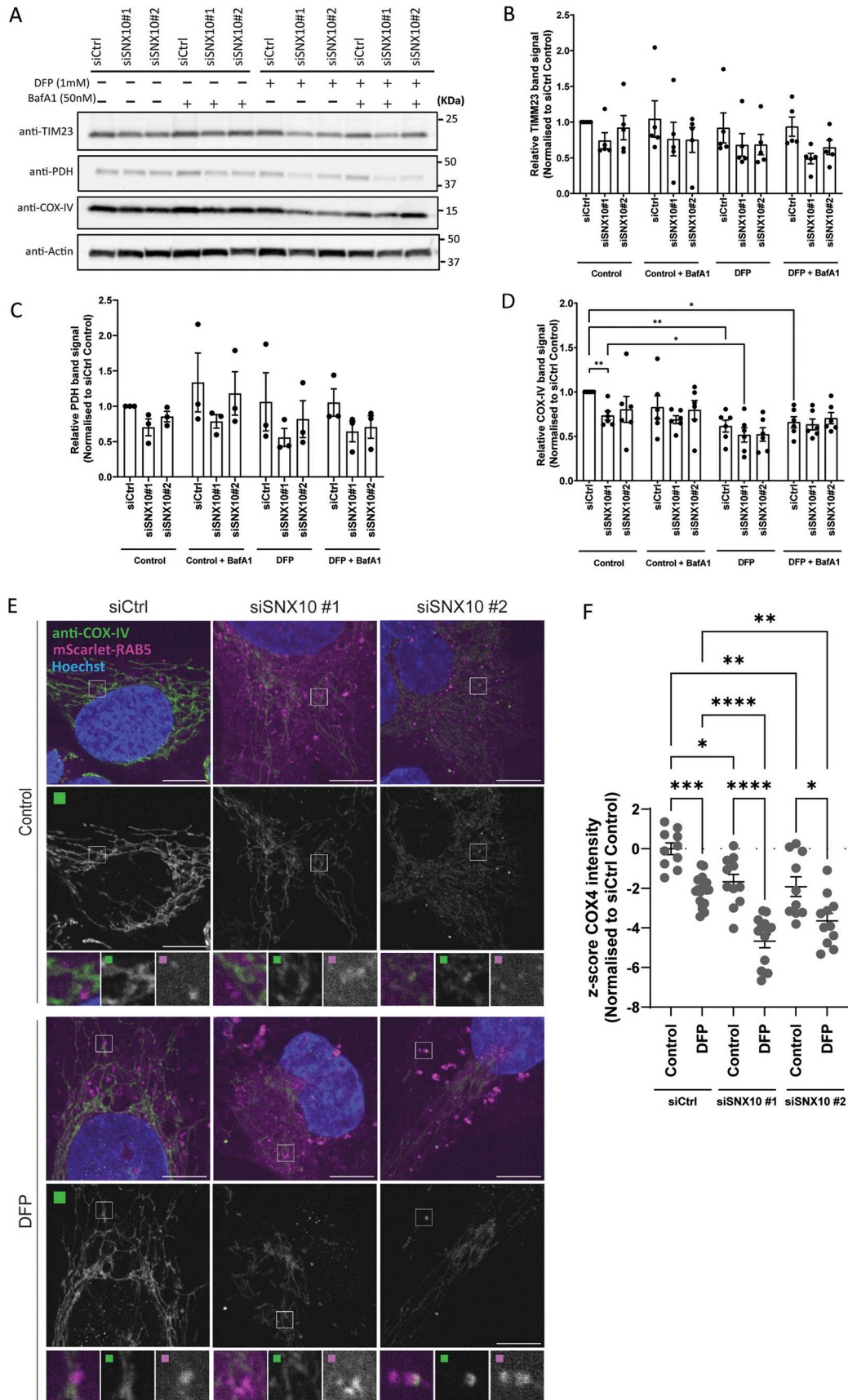

Figure 6.  **SNX10 is a negative modulator of COX-IV turnover. (A)** U2OS cells were reverse transfected with the indicated siRNA (20 nM) for 72 h, then treated or not with DFP (1 µM) for 24 h and with BafA1 (50 nM) the last 16 h, followed by western blotting for the indicated proteins. **(B–D)** Quantification of

the data in A from *n* = 5, 3, and 6 independent experiments. Bars show mean values of the protein levels normalized to actin relative to control conditions (siCtrl control) ± SEM. Significance is assessed by two-way ANOVA followed by Tukey's post hoc test. Data distribution was assumed to be normal. **(E)** U2OS cells with stable expression of mScarlet-RAB5 were reverse transfected with the indicated siRNA (20 nM) for 72 h, then treated or not with DFP for 24 h. The cells were fixed and stained with anti–COX-IV antibody before image acquisition. Scale bar: 10 µm. Insets: 3.69 × 3.69 µm. **(F)** Quantification of COX-IV intensity from E represented as z-score from two independent experiments (>250 cells per experiment). The statistical significance between the control and the other conditions was calculated with ordinary one-way ANOVA followed by Tukey's multiple comparison test. Data distribution was assumed to be normal but was not formally tested. * = $P < 0.05$, ** = $P < 0.01$, *** = $P < 0.001$, and **** = $P < 0.0001$; nonsignificant differences are not depicted. BafA1; bafilomycin A1. Source data are available for this figure: SourceData F6.

stable cell lines were subjected to doxycycline for 24 h prior to treatments to allow protein expression, as indicated in figure legends. After treatments, the cells were washed with PBS and then fixed with PFA fixation buffer (3.7% [wt/vol] PFA and 200 mM HEPES, pH 7.0) for 20 min. After fixing, the cells were washed twice with PBS and then permeabilized for 5 min with 0.2% (vol/vol) NP-40 in PBS. Cells were then washed once and then incubated for blocking for 15 min with 1% BSA in PBS solution (IF-blocking solution). After blocking, the coverslips were incubated with the primary antibodies for 1 h at 37°C (antibodies were diluted in IF-blocking solution at the dilutions stated in the section "Antibodies and dyes") and then washed 3 × 10 min in IF-blocking solution. After washing, the cells were incubated with their corresponding secondary antibodies again diluted in IF-blocking solution at the concentrations above stated for 30 min at RT. Finally, the cells were washed again with IF-blocking solution and then incubated with Hoechst 33342 diluted in 1x PBS at 1 µg/ml for 30 min. Coverslips were mounted in ProLong Diamond Antifade Mountant (#p36965; Invitrogen).

### Live-cell imaging, high-content immunofluorescence microscopy, and confocal microscopy

Imaging settings (laser intensity, time exposure, number of sections per z-stack, and the stepsize between them; time intervals during live imaging) were set identically within each experiment to allow an optimal and non-saturated fluorescent imaging. For widefield imaging, we used an ImageXpress Micro Confocal microscope from Molecular Devices at 20× magnification. Live imaging and imaging of fixed cells were acquired using a Dragonfly 505 spinning disk confocal microscope (Andor Technology) utilizing a 63× or 100× oil objective (1.45 NA) and Zyla 4.2 sCMOS 2,048 × 2,048 camera. The spinning disc confocal mode was used for all pictures. The microscope was equipped with an Okolab cell incubator with temperature, $CO_2$, and humidity control for live-cell imaging. Fixed cells were imaged at RT. If any other microscope was used, it will be specified in the figure legend.

### EGF treatment

U2OS FlpIn TRex pcDNA5-SNX10-EGFP cells transfected with control or SNX10 siRNA for 72 h were then serum starved for 2 h in media supplemented with 10 µg/ml cycloheximide, followed by the addition of 50 ng/ml EGF (sc-4552; Santa Cruz Biotechnology) in combination with 10 µg/ml cycloheximide (C7698; Sigma-Aldrich) for 0, 15, 30, 60, and 1,200 min. The cells were either fixed after 15 min for immunofluorescence or harvested at the aforementioned time points and lysed as described in the western blotting section.

### Image analysis

The identification of relevant structures was carried out using CellProfiler software (v4.2) (Stirling et al., 2021). The images were separated into individual channels and enhanced to improve the visibility of key features. Nuclei, cells, and puncta were detected by applying a manual threshold to segment the areas of interest, allowing for the calculation of puncta counts per cell. Co-localization was assessed by relating the structures of interest and applying a mask that defined regions based on previously identified objects in the analysis pipeline. The percentage of co-localization was calculated using the measurements generated by these modules. In the IMLS cell analysis for mitophagy, mCherry only structures were distinguished by calculating the ratio of the mCherry signal to the EGFP signal on a per-pixel basis, following background noise reduction. The intensity of the red signal was adjusted to correspond with the green signal in controls treated with bafilomycin A1. The intensity was measured by using the "MeasureObjectIntensity" module and by taking into consideration "Integrated intensity" as a final measurement that represents the sum of pixel intensities within each object. These data were plotted as a z-score, in which the values for each field of view were normalized by subtracting the mean and dividing by the standard deviation of the control group.

### Oxygen consumption rate measurement

SNX10 was depleted as described above for a total of 72 h. Prior to starting the experiment, the cells were resuspended in complete DMEM and seeded into Seahorse XFe24 Cell Culture microplates where they reached confluency. The plate was incubated at 37°C in a humidified incubator. The media was then replaced with DMEM without sodium bicarbonate (pH 7.4) before analysis with the Seahorse XFe24 Analyzer (XF mito stress test; Agilent). Specific mitochondrial inhibitors were dissolved in DMEM and loaded into the injector ports of the Seahorse Sensor Plates. The final concentrations of the mitochondrial inhibitors were CCCP: 1 µM, oligomycin: 1.5 µM, rotenone: 0.5 µM, and antimycin A: 0.5 µM. After the analysis, the cells were washed in ice-cold PBS and lysed with NP-40 lysis buffer for protein quantification using BCA Assay (Thermo Fisher Scientific). Quantification was performed using the Seahorse Analytics software (https://seahorseanalytics.agilent.com; Agilent), with normalization based on the measured protein concentration from each well.

### RNA isolation, cDNA synthesis, and qPCR

For quantifying the knockdown efficiency throughout the different experiments, RNA was isolated using the RNeasy Plus

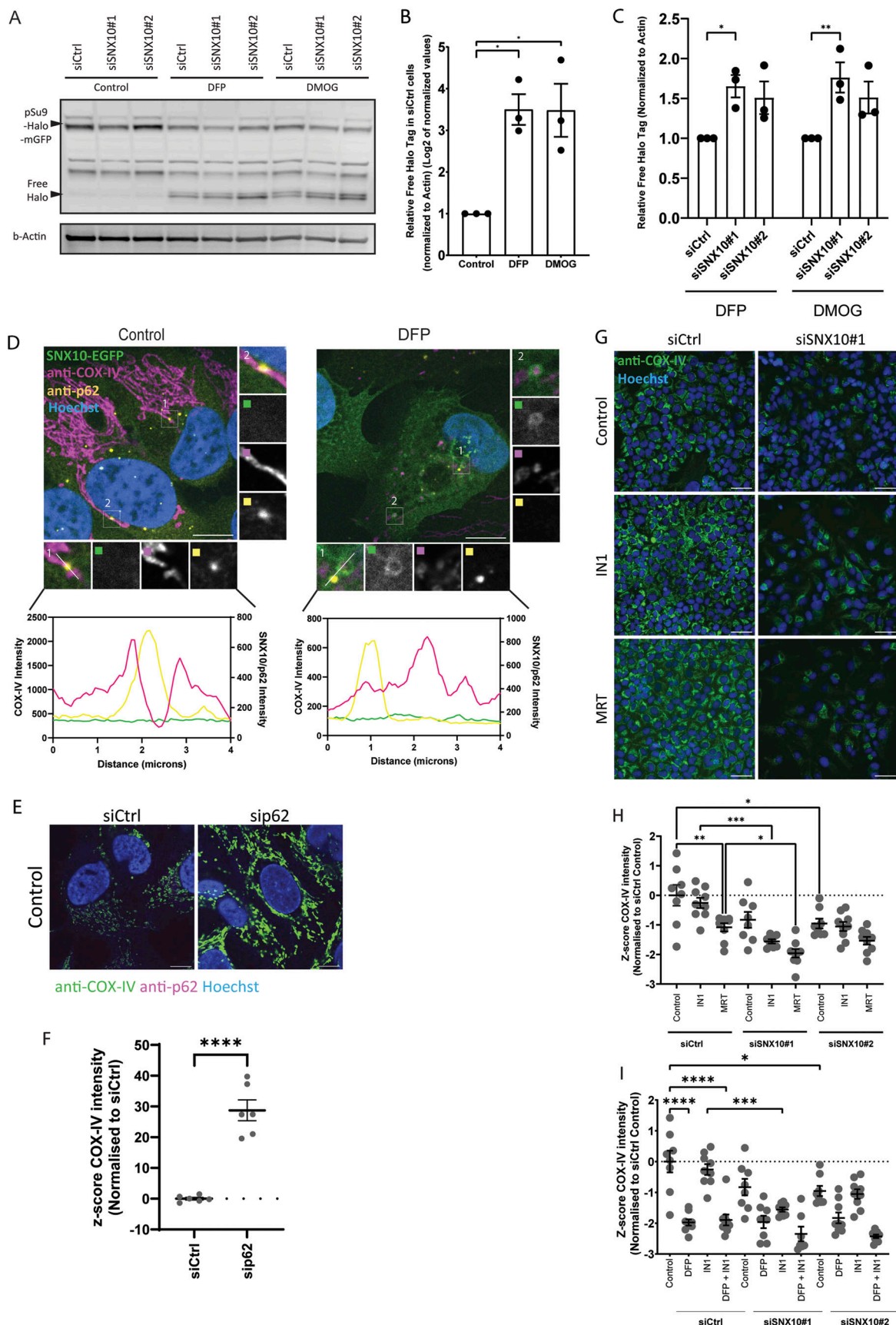

Figure 7. **SNX10 modulates piecemeal mitophagy of OXPHOS components. (A)** U2OS cells stably expressing the reporter pSu9-Halo-mGFP were reverse transfected with siCtrl or siSNX10 for 72 h. Cells were treated with TMR (100 nM) for 20 min, washed three times with PBS, and then treated with DFP (1 μM),

DMOG (1 μM), or left untreated (control) for 24 h before lysis. **(B and C)** The relative Free Halo Tag expression was quantified using the formula (Free Halo/[Free Halo + Full Length]) normalized to actin. Data in B were $\log_2$ transformed. Statistical analysis was performed using one-way ANOVA followed by Dunnett's multiple comparison tests to compare treatment groups to the control. Data represent mean ± SEM from three independent experiments. Data distribution was assumed to be normal but was not formally tested. **(D)** U2OS cells with inducible expression of SNX10-EGFP were treated with or without DFP (1 μM) for 24 h and stained with antibodies anti–COX-IV and anti-p62, followed by acquisition with a Nikon Ti2-E microscope with a Yokogawa CSU-W1 SoRa spinning disk 100×/1.45 NA oil immersion objective. Pixel intensity plot line graphs from control and DFP insets were generated with GraphPad Prism using two different y axis to enhance visualization. Scale bar: 10 μm. Insets: 4.08 × 4.08 μm. **(E)** U2OS cells subjected to reverse transfection with sip62 (20 nM) for 72 h were stained with an anti–COX-IV and anti-p62 antibody, prior to acquisition with a Nikon Ti2-E microscope with a Yokogawa CSU-W1 SoRa spinning disk 100×/1.45 NA oil immersion objective. 100×/1.45 NA oil immersion objective. Scale bar: 10 μm. **(F)** Quantification of COX-IV intensity from E represented as z-score from one experiment with individual data points corresponding to a single field of view (>30 cells per siRNA). Significance was determined by an unpaired two-tailed *t* test. Data distribution was assumed to be normal, but this was not formally tested. **(G)** Representative images of U2OS cells subjected to reverse transfection with siSNX10 (20 nM) for 72 h, followed by treatment with either IN1 or MRT for 24 h before fixation. After fixation, cells were stained with a COX-IV antibody, and images were captured using an ImageXpress Micro Confocal (Molecular Devices) at 20× magnification. **(H and I)** Quantification of COX-IV intensity from G represented as z-score from one independent experiment, with individual data points corresponding to a single field of view (>200 cells were analyzed for each condition). Significance was determined by one-way ANOVA followed by Šídák's multiple comparisons test. Data distribution was assumed to be normal, but this was not formally tested. * = $P < 0.05$, ** = $P < 0.01$, *** = $P < 0.001$, and **** = $P < 0.0001$; nonsignificant differences are not depicted. Source data are available for this figure: SourceData F7.

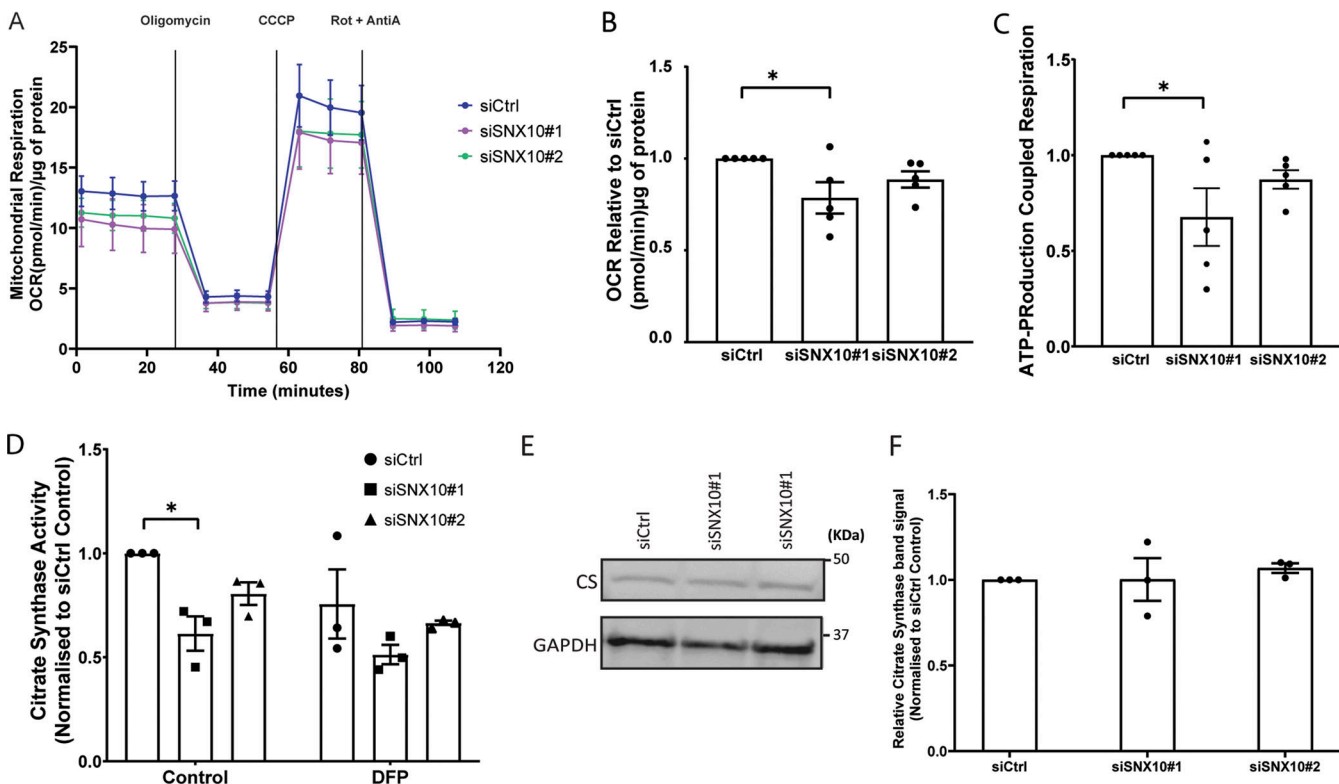

Figure 8.  **SNX10 is important for mitochondrial bioenergetics. (A)** Mitochondrial oxygen consumption rate (OCR) was assessed in control and SNX10 knocked down cells using the Seahorse XFe24 Analyzer. OCR was measured following sequential addition of oligomycin, carbonyl cyanide m-chlorophenyl hydrazone (CCCP), and rotenone/antimycin A (Rot/AntiA). **(B)** The four basal OCR measurements per well were averaged to determine the basal OCR value, and non-mitochondrial respiration was subtracted to ascertain the basal respiration associated with each condition. **(C)** ATP production was calculated by subtracting the proton leak from the maximal respiratory capacity. Error bars represent the mean ± SEM from $n = 5$. Statistical significance was determined using one-way ANOVA followed by Dunnett's multiple comparison test. Data distribution was assumed to be normal but was not formally tested. **(D)** CS activity was determined by spectrophotometry from lysates of U2OS cells transfected with siRNA for 72 h, in the presence or absence of DFP for the last 24 h. The graph displays mean values normalized to siCtrl. Significance was determined from $n = 3$ independent experiments by two-way ANOVA followed by Tukey's multiple comparison test. Data distribution was assumed to be normal but was not formally tested. **(E and F)** Expression levels of CS were measured in control (siCtrl) and SNX10 knockdowns (siSXN10#1 and siSXN10#2) across three independent experiments. Band densities of CS were normalized to the housekeeping gene GAPDH. Data are presented as mean ± SEM. Statistical analysis was performed using one-way ANOVA followed by Dunnett's post hoc test to compare each knockdown group to the control group. Data distribution was assumed to be normal but was not formally tested. * = $P < 0.05$, ** = $P < 0.01$, *** = $P < 0.001$, and **** = $P < 0.0001$; nonsignificant differences are not depicted. Source data are available for this figure: SourceData F8.

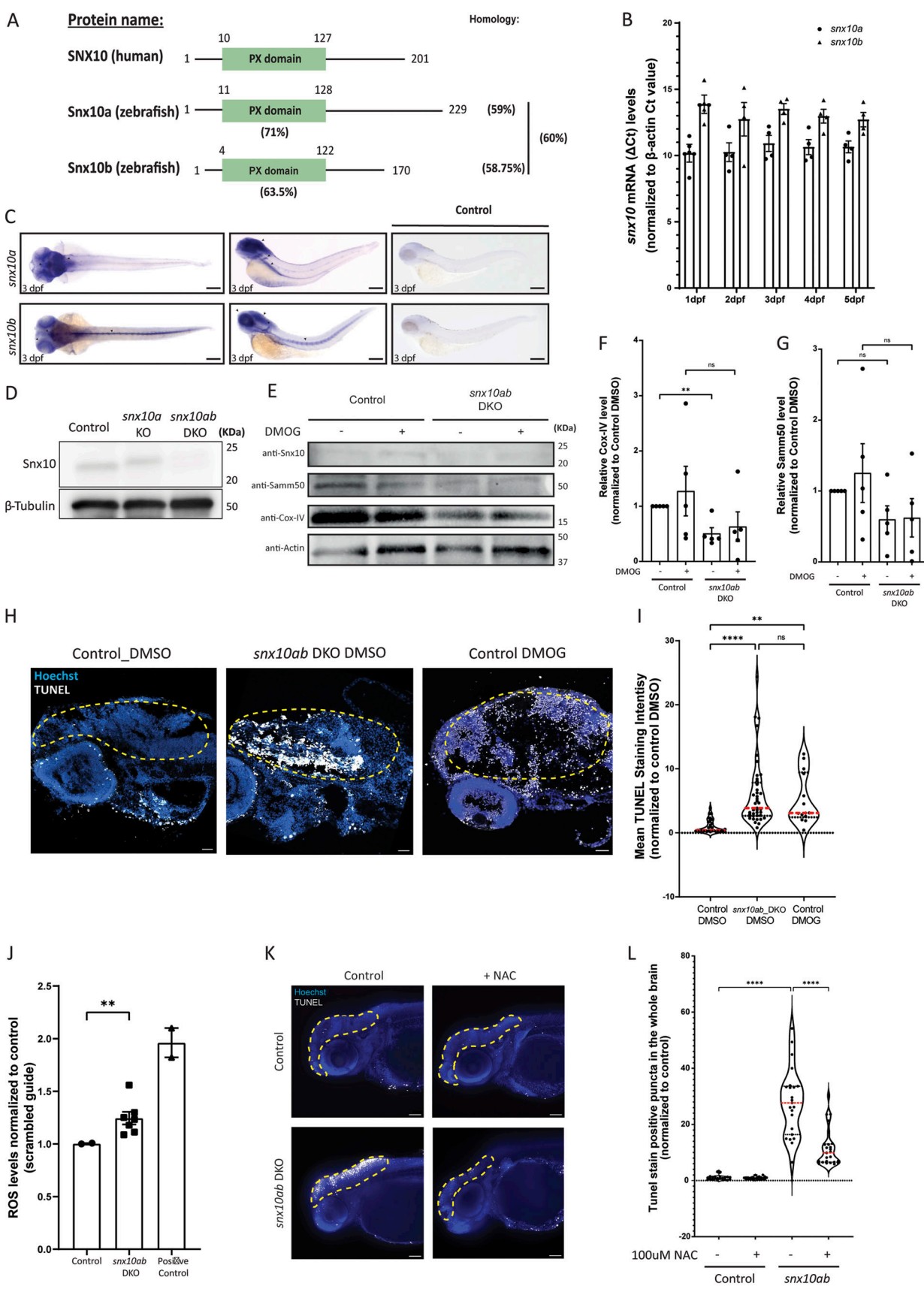

Figure 9. **SNX10 regulates mitochondrial homeostasis and cell death in vivo. (A)** Schematic diagram of human SNX10, zebrafish Snx10a, and Snx10b proteins. The percentage identity of the orthologues amongst each other and to the human counterpart is indicated. Also, the percentage identity of the

zebrafish PX domains in comparison with the PX domain of human SNX10 is shown. **(B)** Temporal expression pattern of *snx10a* and *snx10b*. The graph shows the fold change in transcript levels relative to β-actin in whole zebrafish embryos from 1 to 5 dpf. Error bars indicate mean ± SEM. Data are collected from three individual experiments using 30 larvae for each experiment. **(C)** Dorsal and lateral view of the spatial expression pattern of *snx10a* and *snx10b* at 3 dpf as demonstrated by WM-ISH using an internal probe. Scale bars: 200 µm. Images are representative from three experiments. **(D)** Representative immunoblots of Snx10 and β-tubulin on whole embryo lysates from control (scrambled guide), single *snx10a* KO, and *snx10ab* DKO animals. β-tubulin served as a loading control. **(E)** Representative immunoblots of Snx10, Samm50, Cox-IV, and actin on whole embryo lysates from control (scrambled guide) and *snx10ab* DKO treated with 100 µM DMOG or DMSO control for 24 h at 2 dpf. **(F)** Quantification of the Cox-IV signal intensity from blots in E normalized to control DMSO signal intensity from $n = 4$ experiments. Error bars indicate mean ± SEM, unpaired Student's $t$ test was performed to assess significance. **(G)** Quantification of the Samm50 signal intensity from blots in E normalized to control DMSO signal intensity from $n = 4$ experiments. Error bars indicate mean ± SEM, unpaired Student's $t$ test was performed to assess significance. Data distribution was assumed to be normal but was not formally tested. **(H)** Representative images of TUNEL assay on control (scrambled sgRNA) and *snx10ab* DKO larvae treated with 100 µM DMOG or DMSO control for 24 h at 3 dpf. Orientation lateral. Scale bar: 500 µm. **(I)** Quantification of the mean fluorescent intensity from demarcated brain regions of images in H. A total of 45 control larvae (scrambled sgRNA) and 41 *snx10ab* DKO larvae were used for quantification, respectively. Values were normalized to control DMSO values. Control larvae were treated with DMOG as a comparison to *snx10ab* DKO larvae. $n = 2$ independent experiments. Plots demonstrate data distribution and median value (red line). Significance was determined by two-way ANOVA followed by Tukey's post hoc test to compare all groups. Data distribution was assumed to be normal but was not formally tested. **(J)** Quantification of ROS levels obtained via FACS analysis of control (scrambled sgRNA), *snx10ab* DKO, and positive control larvae at 3 dpf incubated with MitoSOX. The values were presented as relative values after normalizing to control. Error bars indicate mean ± SEM. Quantification was from at least two independent experiments. Data distribution was assumed to be normal but was not formally tested. **(K)** Representative whole mount images shown as maximum intensity projection from z-stack of TUNEL assay performed on control (scrambled sgRNA) and *snx10ab* DKO larvae treated with or without 100 µM NAC at 3 dpf. Orientation lateral. Scale bar: 500 µm. **(L)** Quantification of the number of white puncta (dots) from the demarcated whole brain region shown in K. A total of >20 control larvae (scrambled gRNA) and >20 *snx10ab* DKO larvae treated or not with 100 µM NAC were used for quantification. Values were normalized to control values. Data are collected from three individual experiments. Plots show data distribution and median value (red line). Significance was determined by one-way Brown–Forsythe and Welch's ANOVA tests to compare all groups. Data distribution was assumed to be normal but was not formally tested. * = P < 0.05, ** = P < 0.01, *** = P < 0.001, and **** = P < 0.0001; nonsignificant differences are not depicted. NAC; N-acetyl cysteine. Source data are available for this figure: SourceData F9.

Mini Kit (Qiagen), except for RNA from experiments containing DFP, which was isolated using Trizol (#15596026; Thermo Fisher Scientific). cDNA was synthesized using Superscript III reverse transcriptase (#18080085; Thermo Fisher Scientific) and amplified using KAPA SYBR FAST qPCR Kit and designed primers targeting specific genes. For experiments performed in a 96-well format, the Power SYBR Green Cells-to-CT kit (#4402955; Thermo Fisher Scientific) was used for extraction and amplification according to the manufacturer's instructions. All amplification experiments were run in a CFx96 real-time PCR system (Bio-Rad). Transcript levels were normalized to TATA-box–binding protein, and the quantifications were performed using the $2^{-\Delta\Delta Ct}$ method. The primers used for real-time PCR in this study are BNIP3: 5′-GGCCATCGGATTGGGGATCT-3′ (fwd) and 5′-GGCCACCCCAGGATCTAACA-3′ (rev). BNIP3L: 5′-TCC ACCCAAGGAGTTCCACT-3′ (fwd) and 5′-GTGTGCTCAGTCGCT TTCCA-3′ (rev). COX-IV: 5′-CAGTGGCGGCAGAATGTTG-3′ (fwd) and 5′-GATAACGAGCGCGGTGAAAC-3′ (rev). HI1a: 5′-GGCAGCAACGACACAGAAAC-3′ (fwd) and 5′-GCAGGGTCA GCACTACTTCG-3′ (rev). SNX10: 5′-TTGAGGCGTGTGTTTCTG GG-3′ (fwd) and 5′-CCAAGCCCAGAGGATGAACTTT-3′ (rev). TATA-box–binding protein: 5′-CAGAAAGTTCATCCTCTGGGCT-3′ (fwd) and 5′-TATATTCGGCGTTTCGGGCA-3′ (rev). TIMM23: 5′-GATACCATGGAAGGAGGCGG-3′ (fwd) and 5′-ATCCCTCGAAGA CCACCTGT-3′ (rev). TOMM20: 5′-GCTGGGCTTTCCAAGTTACC-3′ (fwd) and 5′-AGTAACTGCTGTGGCTGTCC-3′ (rev). ULK1: 5′-GTTCCAAACACCTCGGTCCT-3′ (fwd) and 5′-TCCACCCAGAGA CATCTTCCT-3′.

RNA isolation from zebrafish larvae was performed using Trizol reagent (#15596026; Thermo Fisher Scientific). cDNA was synthesized using Superscript III reverse transcriptase (#18080085; Thermo Fisher Scientific) and amplified using KAPA SYBR FAST qPCR Kit. Primers targeting specific genes were designed.

Transcript levels were normalized to beta-actin, and the quantifications were performed using the $2^{-\Delta\Delta Ct}$ method. Primers used for real-time PCR for zebrafish are: snx10a—5′-AGCTATGAGATCTGCCTTCACACC-3′ (fwd) and 5′-TGACGCAGC CAAACAAACTCAC-3′ (rev); snx10b—5′-AGATTTCTGGCATGCCT TCATGG-3′ (fwd) and 5′-AGTGAACGCCAAGCTGTTTGTATG-3′ (rev); cox-iv—5′-CTGCCTTCGTGGTGCACATG-3′ (fwd) and 5′-GTCCTCGACCTTCGCAACT-3′ (rev); samm50—5′-GGAA CAACGAGGGCAGCATG-3′ (fwd) and 5′-GGCAGTTTGATCCCC AGGAC-3′ (rev); and bnip3l—5′-AGCAGCTCGTCCTGCAACAG-3′ (fwd) and 5′-GGACTGTGTGGCCGTGGAGG-3′ (rev). bnip3 and β-actin primers were from Qiagen (QT02053933 and QT02174907, respectively).

## CLEM

For CLEM, U2OS FlpIn cells, stably expressing SNX10-EGFP, were seeded on photo-etched coverslips (Electron Microscopy Sciences). Cells were fixed in 4% formaldehyde, 0.1% glutaraldehyde in 0.1 M PHEM (240 mM PIPES, 100 mM HEPES, 8 mM MgCl$_2$, and 40 mM EGTA), pH 6.9, for 1 h. The coverslips were washed with 0.1 M PHEM buffer and mounted with Mowiol containing 1 µg/ml Hoechst 33342. The cells were examined with a Dragonfly 505 spinning disk confocal microscope (Andor Technology) using an oil-immersed 60× magnification objective. Cells of interest were identified by fluorescence microscopy, and a z-stack covering the whole cell volume was acquired. The relative positioning of the cells on the photo-etched coverslips was determined by taking a low-magnification DIC image. The coverslips were removed from the object glass, washed with 0.1 M PHEM buffer, and fixed in 2% glutaraldehyde/0.1 M PHEM overnight. Cells were postfixed in osmium tetroxide and potassium ferry cyanide, stained with tannic acid and uranyl acetate, and thereafter dehydrated stepwise to 100% ethanol,

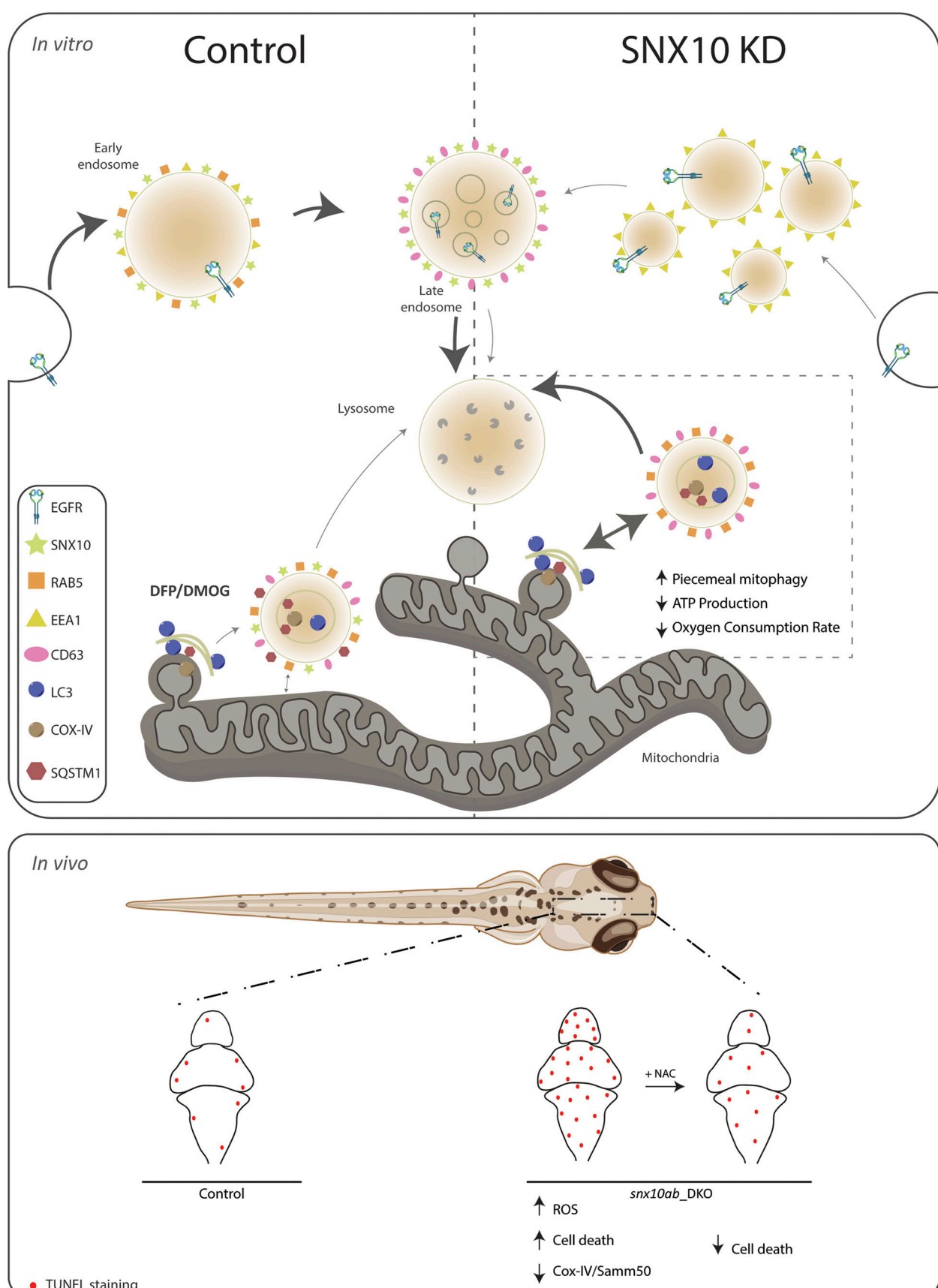

Figure 10. **Model showing the role of SNX10 as a modulator of endocytic transport and piecemeal mitophagy.** In vitro: Under normal conditions (control), SNX10 localizes to early endosomes (RAB5 and EEA1 positive) and late endosomes (CD63 positive) together with endocytic cargo (as EGFR). SNX10-

positive structures are also observed near mitochondria. Upon hypoxia-mimicking conditions (induced by DFP or DMOG) SNX10 vesicles co-localize with CD63, LC3B, and p62 and incorporate selected mitochondrial components (including COX-IV), indicating a role for SNX10 in selective mitochondrial degradation. Upon SNX10 knockdown (SNX10 KD), early endosomes appear smaller and more numerous, with a corresponding reduced degradation of EGFR. In contrast, the turnover of mitochondrial COX-IV and ATP synthase is increased, along with reduced oxygen consumption rate and ATP production, reflecting impaired mitochondrial function. The arrow thicknesses indicate the extent of the different pathways under different conditions. In vivo: In zebrafish larvae, *Snx10ab* DKO leads to decreased levels of mitochondrial proteins (Cox-IV and Samm50), increased levels of ROS, and elevated cell death in the brain region, as shown by TUNEL staining. The snx10ab DKO–mediated cell death can be rescued by treatment with the antioxidant NAC, suggesting that Snx10 modulates piecemeal mitophagy to limit oxidative stress and maintain mitochondrial homeostasis. NAC; N-acetyl cysteine.

---

followed by flat-embedding in Epon. Serial sections (200 nm) were cut on an Ultracut UCT ultramicrotome (Leica) and collected on formvar-coated slot grids.

Sections were observed at 200 kV in a Thermo Fisher Scientific Talos F200C microscope and recorded with a Ceta 16 M camera. For tomograms, image series were taken between –60° and 60° tilt angles with 2° increment. Single-axes tilt series were recorded with a Ceta 16M camera. Tomograms were computed using weighted back projection using the IMOD package. Display, segmentation, and animation of tomograms were also performed using IMOD software version 4.9 (Kremer et al., 1996).

To assess the diameter of endosomes, U2OS cells subjected to SNX10 silencing were grown on poly-l-lysine–coated sapphire discs. To label newly internalized EGFR following EGF stimulation, cells were first washed in ice-cold PBS and incubated with an antibody recognizing the extracellular part of EGFR (mouse anti-EGFR; Pharmingen) on ice. After washing four times with ice-cold PBS, cells were incubated with Protein A-10 nm gold conjugate (UMC Utrecht Dept. of Cell Biology), which recognizes the Fc region of the mouse IgG2b primary antibody. Cells were again washed four times with ice-cold PBS, stimulated with EGF in warm DMEM for 40 min, and finally high-pressure frozen using a Leica HPM100. Freeze substitution was performed as follows: sample carriers designed for sapphire discs were filled with 4 ml of freeze substituent (0.1% [wt/vol] uranyl acetate in acetone, 1% $H_2O$) and placed in a temperature-controlling AFS2 (Leica) equipped with an FPS robot. Freeze substitution occurred at –90°C for 48 h before the temperature was raised to –45°C over a time span of 9 h. The samples were kept in the freeze substituent at –45°C for 5 h before washing three times with acetone, followed by a temperature increase (5°C per hour) to –35°C. Samples were stepwise infiltrated with increasing concentrations of Lowicryl HM20 (10%, 25%, and 75%, 4 h each). During the last two steps, the temperature was gradually raised to –25°C before infiltrating three times with 100% Lowicryl (10 h each). Subsequent UV polymerization was initiated for 48 h at –25°C, and the temperature was then evenly raised to +20°C (5°C per hour). Polymerization then continued for another 24 h at 20°C. Serial sections (250 nm) were cut on an Ultracut UCT ultramicrotome (Leica) and collected on formvar-coated slot grids.

Single-axis tilt tomograms were collected in a Thermo Scientific Talos F200C microscope equipped with a Ceta 16 M camera, and image series were taken between –60° and 60° tilt angles with 2° increment. Tomograms were reconstructed using weighted back projection using the IMOD package software version 4 (Kremer et al., 1996).

## SNX10 pull-down for interactome analysis

To analyze the SNX10-EGFP interactome, we used mass spectrometry assays in biological triplicates. U2OS cells expressing SNX10-EGFP, SNX10(Y32S)-EGFP, or EGFP were lysed with NP-40 lysis (50 mM HEPES, pH 7.4, 150 mM NaCl, 1 mM EDTA, 10% glycerol, and 0.5% NP-40) supplemented with PhosStop phosphatase inhibitor (Sigma-Aldrich) and cOmplete protease inhibitor cocktail (Merck). The protein purification was performed using ChromoTek GFP-Trap, following the vendor's specifications. The mass spectrometry experiments were performed at the Proteomics Core Facility at Oslo University Hospital (Rikshospitalet). Beads were washed twice with 50 mM ammonium bicarbonate. Proteins on beads were reduced and alkylated and further digested by trypsin for overnight at 37°C. Digested peptides were transferred to a new tube, acidified, and the peptides were desalted for MS analysis.

## LC-MS/MS

Peptide samples were dissolved in 10 μl of 0.1% formic buffer, and 3 μl were loaded for MS analysis. LC-MS/MS analysis of the resulting peptides was performed using an Easy nLC1000 liquid chromatography system (Thermo Electron) coupled to a QExactive HF Hybrid Quadrupole-Orbitrap mass spectrometer (Thermo Electron) with a nano-electrospray ion source (EasySpray; Thermo Electron). The LC separation of peptides was performed using an EasySpray C18 analytical column (2-μm particle size, 100 Å, 75-μm inner diameter, and 25 cm; Thermo Fisher Scientific). Peptides were separated over a 60-min gradient from 2 to 30% (vol/vol) ACN in 0.1% (vol/vol) FA, after which the column was washed using 90% (vol/vol) ACN in 0.1% (vol/vol) FA for 20 min (flow rate 0.3 μl/min). All LC-MS/MS analyses were operated in data-dependent mode, where the most intense peptides were automatically selected for fragmentation by high-energy collision-induced dissociation.

Raw files from LC-MS/MS analyses were submitted to MaxQuant 1.6.17.0 software (Cox and Mann, 2008) for peptide/protein identification. Parameters were set as follows: Carbamidomethyl was set as a fixed modification and PTY, protein N-acetylation, and methionine oxidation as variable modifications. First search error window of 20 ppm and mains search error of 6 ppm. Trypsin without the proline restriction enzyme option was used, with two allowed miscleavages. Minimal unique peptides were set to one, and the false discovery rate (FDR) allowed was 0.01 (1%) for peptide and protein identification. The UniProt human database was used. The generation of reversed sequences was selected to assign FDR rates.

The analysis of label-free intensities was performed with Perseus (Tyanova et al., 2016). The proteins with contaminants were omitted from the analysis. We used R (R Core Team, 2020) (Team 2020) limma (Ritchie et al., 2015) for performing imputation of missing values, fold change analysis, and P values were adjusted using the default adjust method used by topTable() command of limma R package. Gene Ontology enrichment analysis for biological processes and cellular components was performed using Database for Annotation, Visualization, and Integrated Discovery with FDR correction to identify significantly enriched terms. Pie charts and Venn diagrams were plotted using ggplot2 R package and InteractiVenn (Heberle et al., 2015), respectively. The mass spectrometry proteomics data have been deposited to the ProteomeXchange Consortium via the PRIDE (Perez-Riverol et al., 2022) partner repository with the dataset identifier PXD056720.

## Western blotting

For western blotting, the cells were treated as indicated in each experiment and then moved onto ice prior to lysing. Briefly, the cells were washed twice with PBS and then lysed on ice in NP-40 lysis buffer (50 mM HEPES, pH 7.4, 150 mM NaCl, 1 mM EDTA, 10% glycerol, and 0.5% NP-40) supplemented with protease and phosphatase inhibitors (unless specified otherwise). The cells were incubated for 10 min and then collected. The samples were then centrifuged at $21,000 \times g$ at 4°C for 10 min, and then the supernatant was collected. Protein quantitation was assessed by Bradford assay (#5000006; Bio-Rad) using serial dilutions of BSA as standards. Samples were then normalized and added to sample buffer (2.8x NuPage LDS Sample buffer + DTT 0.30 M). After running the samples by an SDS polyacrylamide gradient gel (4–20% Criterion TGX Precast Midi Protein Gel, Bio-Rad) and transferring them to a PVDF membrane (350 mA/1 h), the samples were blocked with casein buffer (0.5% casein, 0.1%, NaAzide, and PBS) for 1 h at RT and then incubated overnight at 4°C with primary antibodies. Primary antibodies were diluted in a 5% BSA in 1X PBS, 0.1% Tween-20 Detergent solution (PBST). Membranes were washed 3 × 10 min in PBST and then incubated with the secondary antibodies 1 h at RT. Membranes were washed again 3 × 10 min in PBST and a last wash in PBS prior to image acquisition with an Odyssey CLx Infrared Imaging system. For HRP-conjugated secondary antibodies, the membranes were developed using the ECL kit and scanned with ChemiDoc MP Imaging system (Bio-Rad). The blots were quantified with Image Studio Lite software, and GraphPad Prism was used for statistical analysis (unpaired $t$ test) of the data.

## WM-ISH

WM-ISHs for snx10a and snx10b were performed as previously described (Thisse and Thisse, 2008) using digoxigenin-labeled riboprobes. Primer sequences for snx10a sense and antisense probes were ATG probe: forward primer (5′-ATGGATAACACA AGCTTTGAG-3′), reverse primer (5′-TCATATAACTGAATGGCC ATG-3′), and for snx10b sense and antisense probes were ATG probe: forward primer (5′-TATGCAGGAATTCACTGGCG-3′), reverse primer (5′-CACTCTGATTCACATTGCAC-3′).

## Zebrafish maintenance

WT zebrafish (AB strain) were housed at the zebrafish facility at the Center for Molecular Medicine Norway (AVD.172) using standard practices. Embryos were incubated in egg water (0.06 g/liter salt [Red Sea]) or E3 medium (5 mM NaCl, 0.17 mM KCl, 0.33 mM CaCl$_2$, and 0.33 mM MgSO$_4$, equilibrated to pH 7.0). Embryos were held at 28°C in an incubator following collection. Experimental procedures followed the recommendations of the Norwegian Regulation on Animal Experimentation ("Forskrift om forsøk med dyr" from 15.jan.1996). All experiments conducted on WT zebrafish larvae were done at 5 dpf or earlier.

## CRISPR/Cas9 genome editing in zebrafish

Zebrafish snx10a and snx10b KO embryos were generated using CRISPR/Cas9 technology as described (Kroll et al., 2021). Two guide RNAs were designed, each for snx10a and snx10b, respectively, based on predictions from the online sgRNA prediction web tool CHOPCHOP (https://chopchop.cbu.uib.no/). Genomic DNA sequences retrieved from Ensembl GRCz10 or z11 (https://www.ensembl.org/index.html) were used for the target site searches. The two guides, designated as sgRNA#1 and sgRNA#2, were predicted to target exon 2 and exon 3 of the snx10a gene and exon 1 and 3 of the snx10b gene, respectively. sgRNA is the annealed product of a target-specific element called the crRNA and a trans-activating CRISPR element called tracrRNA (both of which are ordered from Integrated DNA Technologies). 1 µl of each of these nucleotide elements were annealed together with 1.28 µl of duplex buffer in a thermocycler at 95°C for 5 min. After annealing, 1 µl of sgRNA and 1 µl of Cas9 nuclease (Integrated DNA Technologies) were incubated together in a thermocycler at 37°C for 5 min sgRNA assembled with Cas9 is RNP. snx10a RNP was pooled together with snx10b RNP to create a mix of snx10ab RNP to create DKOs. 1 nl (267.5 pg) of this mix was injected in the yolk sac at the single cell stage before the cell inflated. Oligonucleotides used for snx10a sgRNA synthesis were sgRNA#1 (5′-ACGGGAUCCUCAGGUUCACAG UUUUAGAGCUAUGCU-3′) and sgRNA#2 (5′-GGGUUCCAAGGA GGAAGUUUGUUUUAGAGCUAUGCU-3′); for snx10b sgRNA synthesis were sgRNA#1 (5′-AGAAAUUGGAGCCCGUAUGGG UUUUAGAGCUAUGCU-3′) and sgRNA#2 (5′-CAUGCCAGAAAU CUUCCUUCGUUUUAGAGCUAUGCU-3′). TracrRNA used was (5′-AAAAGCACCGACUCGGUGCCACUUUUUCAAGUUGAUAAC GGACUAGCCUUAUUUUAACUUGCUAUUUCUAGCUCUAAAAC-3′).

## Zebrafish cryosectioning, cell death assay by TUNEL staining, confocal microscopy, and image analysis

Control and *snx10ab* DKO zebrafish larvae were treated or not with 100 µM DMOG for 24 h in the absence or presence of 100 µM N-acetyl cysteine for the last 6 h of DMOG treatment at 3 dpf. At experimental endpoints, larvae were washed once in embryo water and fixed with 4% PFA (in HEPES) at RT for 2 h. After fixation, larvae were washed three times in PBS. The larvae were then cryopreserved in a 2-ml tube in increasing amounts of sucrose in 0.1 M PBS with 0.01% sodium azide. Cryopreservation was done first in 15% sucrose solution for 1 h at RT or up until larvae dropped to the bottom of the tube and then

in 30% sucrose solution at 4°C overnight with gentle shaking. Cryopreserved larvae were oriented in a cryomold (Ref: 4565; Tissue-Tek Cryomold, Sakura) with optimal cutting temperature compound (Ref: 4583; OCT compound; Tissue-Tek Sakura). Larvae were oriented with the ventral side down, and additional OCT was added to fill the mold and frozen on dry ice. A solid block of OCT with larvae oriented in the desired way was taken out from the mold, and 12-μm coronal slices were sectioned on a cryostat (Thermo Fisher Scientific). Sections were collected on Superfrost Plus glass slides (Ref: J1800AMNZ; Thermo Fisher Scientific) and kept at RT for at least 2 h to firmly tether slices onto the glass slide.

Slides were rehydrated three times in PBS at RT for 3 min each. The area of interest was circled by a hydrophobic PAP pen (ab2601; Abcam), and the slides were placed in a humidified chamber to avoid drying out. Rest of the procedure for TUNEL staining (Thermo Fisher Scientific) was performed according to the manufacturer's instructions. Confocal images were obtained using an Apochromat ×20/0.8 or ×40/1.0 oil DIC objective on an LSM 800 microscope (Zeiss). Image analysis was performed using Fiji, by which mean fluorescent intensity was measured in the brain region of the respective sections of zebrafish larvae.

### ROS analysis in zebrafish
The analysis of ROS levels in zebrafish was performed as described (Mugoni et al., 2014). Briefly, after dissociation into single cells of the respective larvae, cells were treated with 5 μM of MitoSOX (Thermo Fisher Scientific) in HBSS for 15 min at 28°C in the dark. Cells were then centrifuged for 5 min at 250 × $g$ at 4°C, the supernatant discarded, and the cells washed with HBSS. FACS estimations were done on NovoCyte Flow Cytometer Systems, using an excitation peak of 396 nm for selective detection of mito superoxide.

### Statistics
Experimental values were used for statistical analysis using Prism (v8.0.1) and post hoc tests as indicated in figure legends. All data values come from distinct samples. * = $P < 0.05$, ** = $P < 0.01$, *** = $P < 0.001$, and **** = $P < 0.0001$; nonsignificant differences are not depicted.

### Online supplemental material
Fig. S1 shows the additional data for Fig. 1. Fig. S2 shows the additional data for Figs. 5 and 6. Fig. S3 shows the additional data for Fig. 7. Fig. S4 shows the additional data for Fig. 9. Table S1 includes the plasmids used in this study. Video 1 is the time-lapse video of the data shown in Fig. 3 C.

### Data availability
All data used in this study are available upon request.

## Acknowledgments

We would like to thank the Simonsen lab for their support and critical discussion throughout the project development, especially Patricia González-Rodríguez and Laura Rodriguez de la Ballina (Department of Molecular Cell Biology, Institute for Cancer Research, Oslo University Hospital and Centre for Cancer Cell Reprogramming, University of Oslo, Oslo, Norway) for providing U2OS Su9-HaloTag7-mGFP cells and for assistance with data analysis, respectively. We would like to acknowledge Ulrikke Dahl Brinch for assistance with EM sample preparation. We acknowledge the Norwegian Core Facility for Human Pluripotent Stem Cells at the Norwegian Center for Stem Cell Research for mycoplasma testing and the ALM core facility, Gaustad node, where high-content imaging was performed.

Mass spectrometry–based proteomic analyses were performed by the Proteomics Core Facility, Department of Immunology, University of Oslo/Oslo University Hospital, which is supported by the Core Facilities program of the South-Eastern Norway Regional Health Authority. This core facility is also a member of the National Network of Advanced Proteomics Infrastructure, which is funded by the Research Council of Norway INFRASTRUKTUR-program (project number: 295910). This project was carried out with funding from the Norwegian Cancer Society (grants no. 171318 and 223278), from the European Union's Horizon 2020 research and innovation program under the Marie Skłodowska-Curie (grant no. 801133), the South-Eastern Norway Regional Health Authority (grant no. 2020032), and the Research Council of Norway through its Centers of Excellence funding scheme (grant no. 262652) and FRIPRO (grant no. 249753).

Author contributions: L. Trachsel-Moncho: conceptualization, data curation, formal analysis, investigation, methodology, project administration, supervision, validation, visualization, and writing—original draft, review, and editing. C. Veroni: formal analysis and investigation. B.J. Mathai: data curation, formal analysis, investigation, methodology, validation, visualization, and writing—review and editing. A. Lapao: investigation and validation. S. Singh: data curation, formal analysis, investigation, methodology, resources, software, visualization, and writing—original draft. N.T. Asp: data curation and formal analysis. S.W. Schultz: data curation, investigation, and visualization. S. Pankiv: investigation and resources. A. Simonsen: conceptualization, funding acquisition, investigation, methodology, project administration, resources, supervision, validation, visualization, and writing—original draft, review, and editing.

Disclosures: The authors declare no competing interests exist.

Submitted: 5 April 2024

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

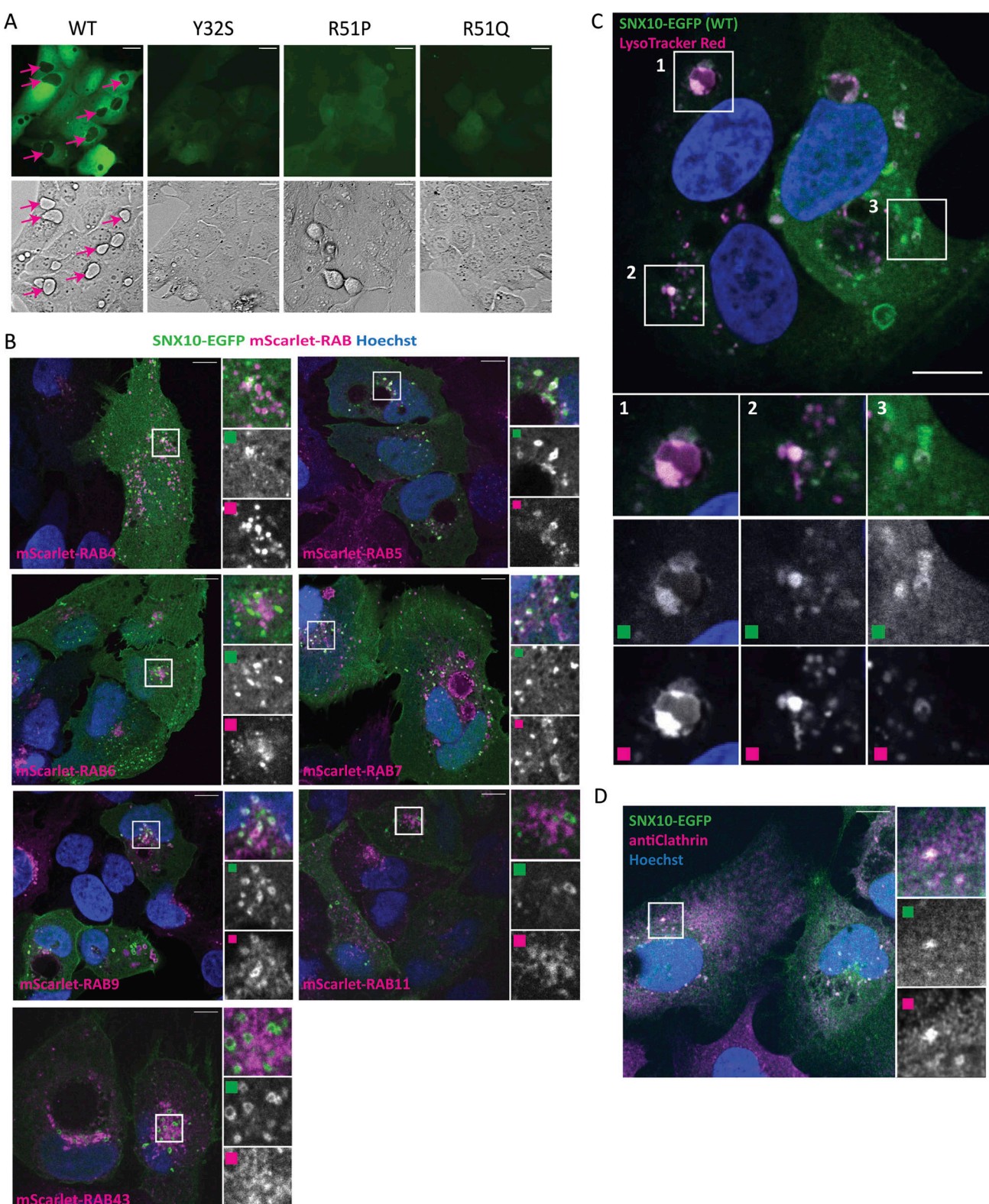

Figure S1.   **SNX10 localizes to endocytic structures. (A)** Fluorescence imaging of U2OS cell lines stably expressing doxycycline-inducible SNX10-EGFP WT or the indicated ARO-linked mutants was acquired at 20× magnification using a Zeiss Axio Observer widefield microscope (Zen Blue 2.3; Zeiss). Corresponding brightfield images are displayed below the fluorescence images. Pink arrows correspond to observed vacuoles. Scale bar = 20 μm. **(C)** U2OS cells stably expressing SNX10-EGFP were treated with LysoTracker Red prior to fixation. Scale bars: 10 μm. Insets: 8.40 × 8.40 μm. **(B)** U2OS cells with stable inducible expression of SNX10-EGFP were infected with lentiviral particles to express mScarlet-RAB4/RAB5/RAB6/RAB7/RAB9/RAB11/RAB43. Nuclei were stained with Hoechst. Scale bars: 10 μm. Insets: 10.92 × 10.92 μm. **(D)** U2OS cells with stable inducible expression of SNX10-EGFP were fixed and stained against endogenous clathrin. Nuclei were stained with Hoechst. Scale bar: 10 μm. Insets: 9.48 × 9.48 μm.

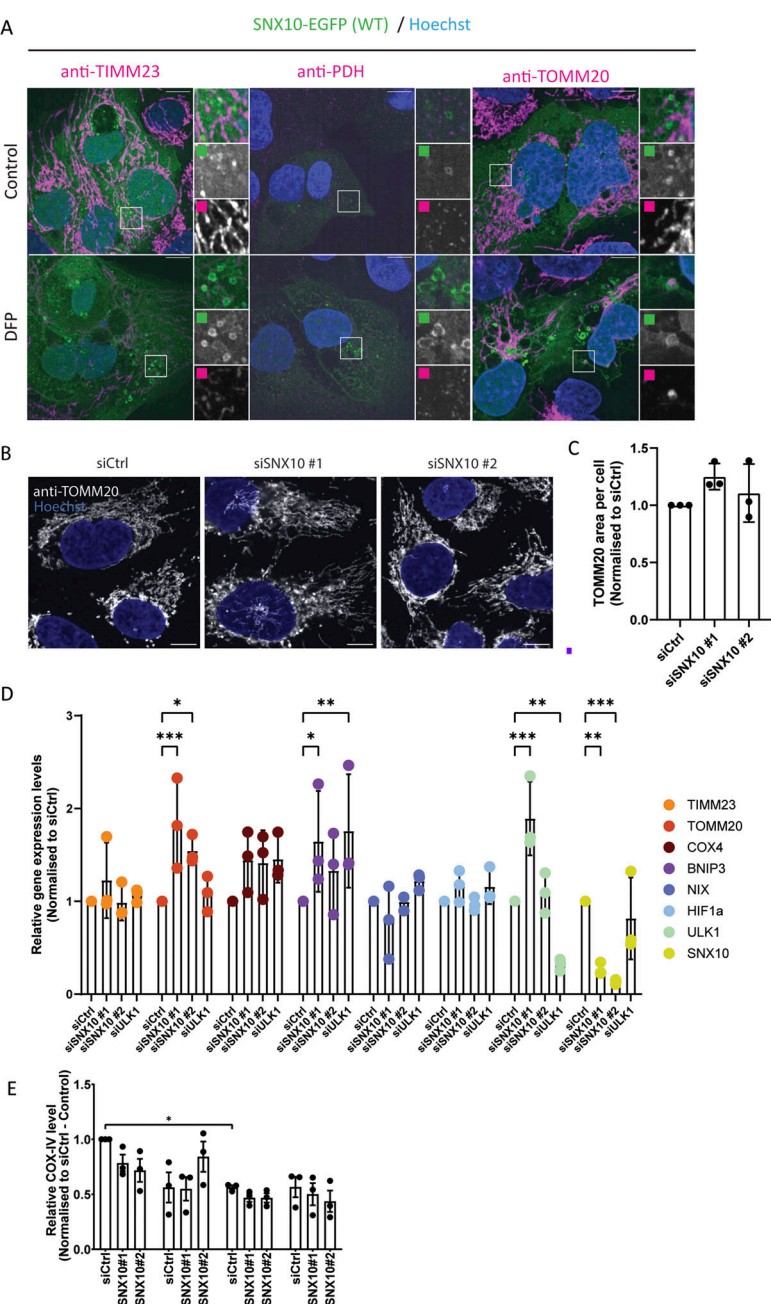

Figure S2. **SNX10 modulates COX-IV protein levels. (A)** U2OS cells with stable inducible expression of SNX10-EGFP were pre-treated with doxycycline for 16 h before the addition of DFP for 24 h. The cells were fixed and stained with antibodies against the indicated mitochondrial proteins for subsequent analysis. Scale bars: 10 μm. Insets: 8.57 × 8.57 μm. **(B)** Representative images of U2OS cells transfected with 20 nM siRNA: siCtrl (control) and two different siSNX10 oligoes (siSNX10 #1 and siSNX10 #2). Cells were stained with an anti-TOMM20 antibody after fixation. Images were acquired using a Nikon CREST X-Light V3 spinning disk microscope utilizing a 60× oil objective. Scale bar: 10 μm. **(C)** Quantification of the data shown in B, performed using CellProfiler software. The graph displays the area occupied by TOMM20 per cell ($n$ = 3, >100 cells per condition in each replicate). Significance was assessed by ordinary one-way ANOVA followed by Tukey's multiple comparison test. Data distribution was assumed to be normal but was not formally tested. **(D)** U2OS cells were reverse transfected with the indicated siRNA (20 nM) for 72 h. The cells were lysed in the well, and the RNA was extracted prior to cDNA synthesis. The graph shows the difference in expression levels upon KD of the different proteins (mean values ± SEM). The values were normalized to TBP using the $2^{-\Delta\Delta Ct}$ method and then compared with siCtrl control. Significance was determined from $n$ = 2 independent experiments by one-way ANOVA followed by Dunnetts's multiple comparison test. Data distribution was assumed to be normal but was not formally tested. **(E)** Quantification of COX-IV protein expression levels in control (siCtrl) and SNX10 (siSXN10#1, siSXN10#2) depleted cells upon treatment of MG132 and/or DFP across three independent experiments. Band densities were normalized to the housekeeping gene actin. Data are presented as mean ± SEM. Statistical analysis was performed using one-way ANOVA followed by Šídák's multiple comparisons test to compare each knockdown group to the control group. Data distribution was assumed to be normal but was not formally tested. * = P < 0.05, ** = P < 0.01, *** = P < 0.001, and **** = P < 0.0001; nonsignificant differences are not depicted. TBP; TATA-box–binding protein and KD; knockdown.

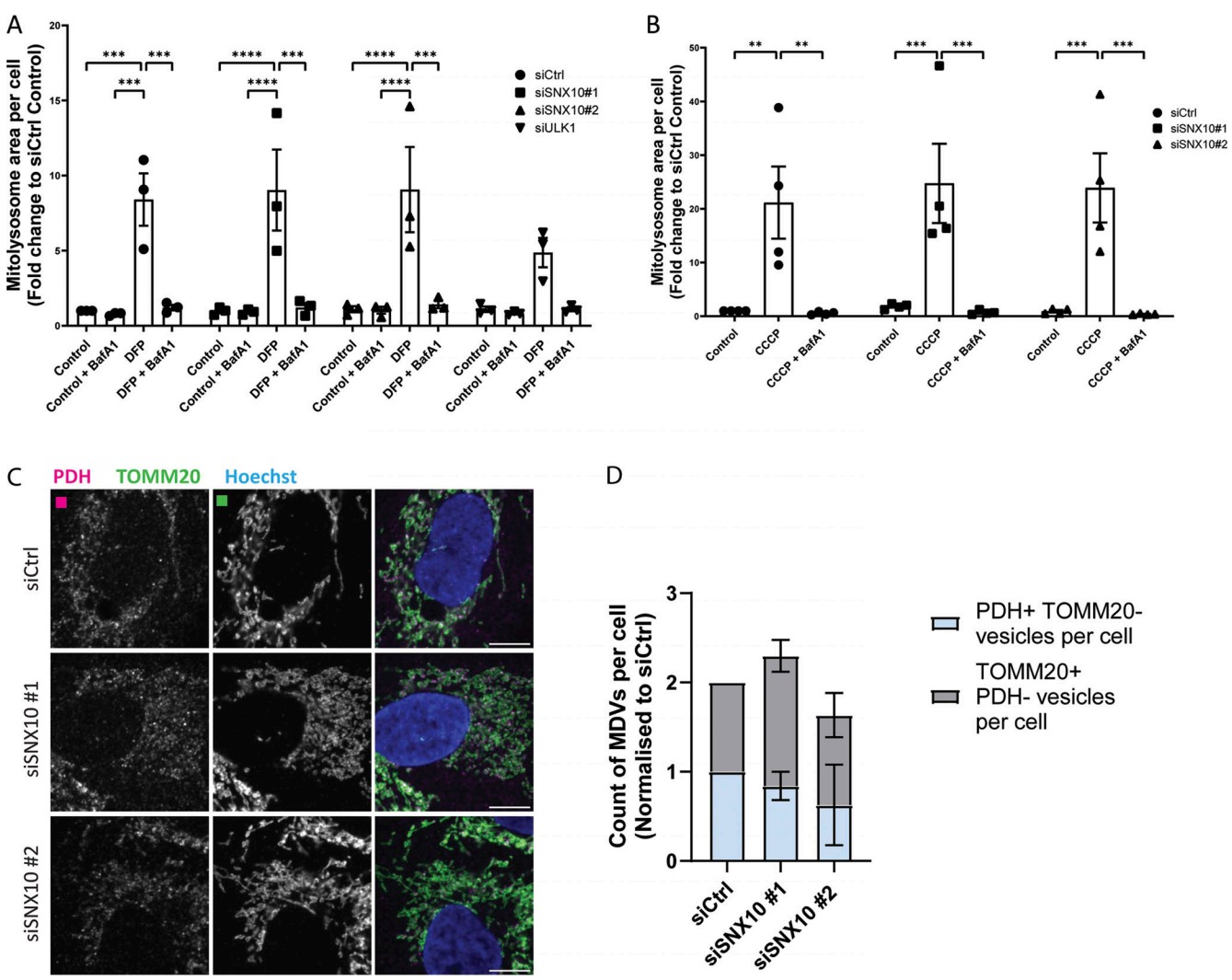

Figure S3. **SNX10 is dispensible for mitophagy of a matrix reporter and MDV formation. (A)** U2OS cells stably expressing iMLS-GFP-mCherry were reverse transfected with Ctrl, SNX10, or ULK1 siRNAs (20 nM) for 72 h. DFP was added for the last 24 h and 50 nM BafA1 was added 16 h before fixation. Scale bar: 20 µm. The graph represents the mitolysosome area per cell from >1,000 cells based on images taken with ImageXpress Micro Confocal (Molecular devices) at 20× magnification. The bars show the means normalized to the control (siCtrl) cells ± SEM (*n* = 3). Significance was determined by two-way ANOVA followed by Tukey's multiple comparison test. **(B)** U2OS cells stably expressing MLS-GFP-mCherry and non-tagged Parkin were reverse transfected with Ctrl and SNX10 siRNAs (20 nM) for 72 h. CCCP was added for the last 24 h and 50 nM BafA1 was added 2 h before fixation. Scale bar: 20 µm. The graph represents the mitolysosome area per cell from >1,000 cells based on images taken with ImageXpress Micro Confocal (Molecular devices) at 20× magnification. The bars show the means normalized to the control (siCtrl) cells ± SEM (*n* = 3). Significance was determined by two-way ANOVA followed by Tukey's multiple comparison test. **(C)** Representative images of U2OS cells transfected with 20 nM siRNA: siCtrl (control) and two different siSNX10 oligoes (siSNX10 #1 and siSNX10 #2). Cells were stained with an anti-PDH and anti-TOMM20 antibody after fixation. Images were acquired using a Nikon CREST X-Light V3 spinning disk microscope utilizing a 60× oil objective. Scale bar: 10 µm. **(D)** Quantification of the data shown in C, performed using CellProfiler software. The graph displays the number of MDVs per cell, calculated as vesicles positive for TOMM20 only or PDH only (*n* = 3, >100 cells per condition in each replicate). Significance was assessed by two-way ANOVA followed by Dunnett's multiple comparison test. Data distribution was assumed to be normal but was not formally tested. BafA1; bafilomycin A1 and CCCP; carbonyl cyanide m-chlorophenyl hydrazine.

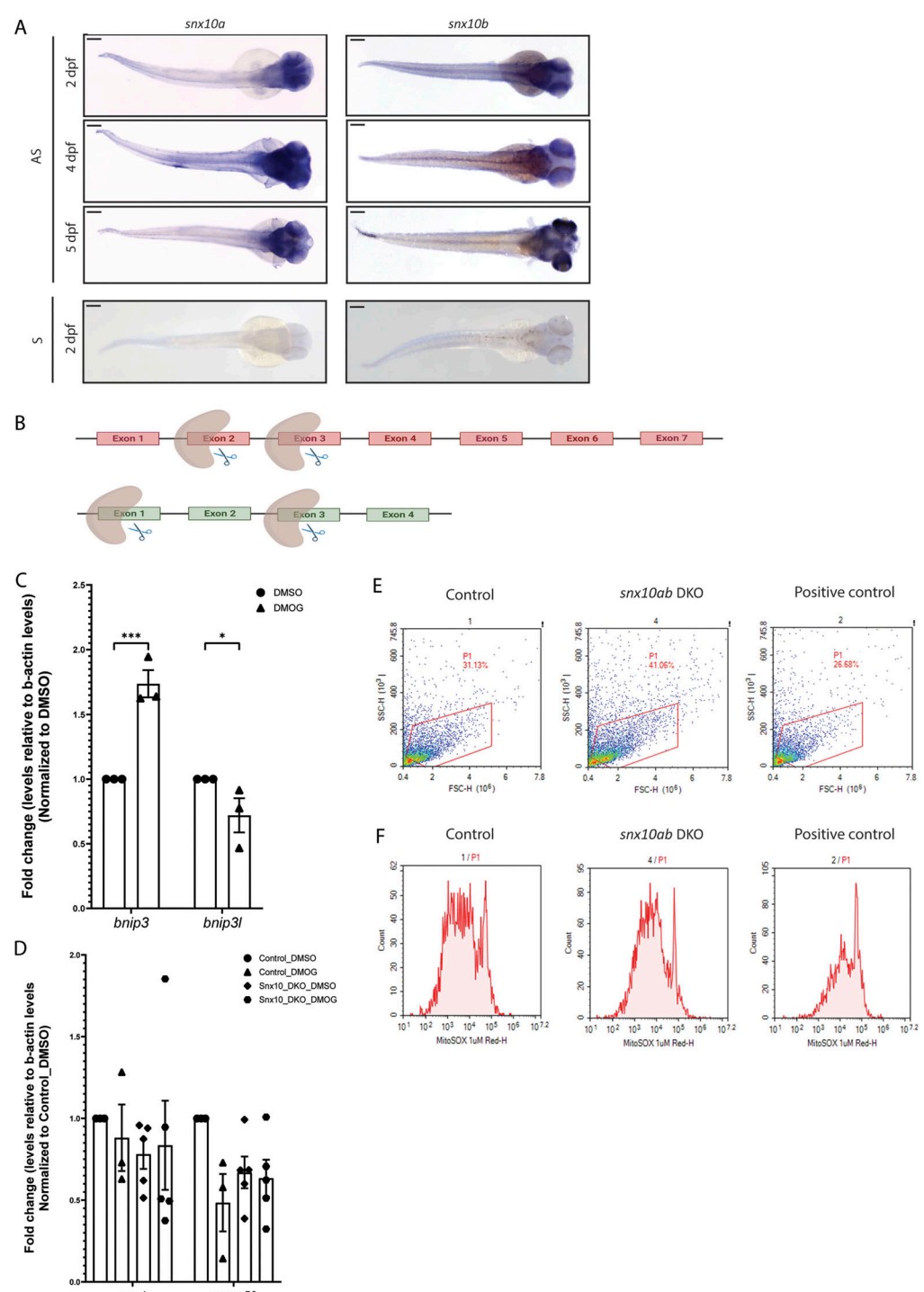

Figure S4. **snx10 is highly expressed in the brain of zebrafish larvae and its depletion elevates oxidative stress. (A)** Dorsal view of spatial expression pattern of snx10a and snx10b at 2, 4, and 5 dpf as demonstrated by WM-ISH using an internal antisense (AS) probe. Scale bar = 200 μm. Images are representative of three experiments. Control 2 dpf larvae hybridized to a sense probe (S). **(B)** Illustration of sgRNA-binding regions on *snx10a* and *snx10b* gene, respectively. **(C)** Temporal expression levels of *bnip3* and *bnip3l* transcripts in DMSO- and DMOG-treated WT zebrafish larvae at 3 dpf. The graph shows the fold change in transcript levels relative to β-actin and normalized to DMSO $2^{-\Delta\Delta Ct}$ levels. Error bars indicate mean ± SEM. Data are collected from three individual experiments. Significance was determined by two-way ANOVA test to compare all groups with the two variables. **(D)** Temporal expression levels of *cox-iv* and *samm50* transcripts in control and *snx10ab*_DKO zebrafish larvae treated with or without 100 μm DMOG at 3 dpf. The graph shows the fold change in transcript levels relative to β-actin and normalized to DMSO $2^{-\Delta\Delta Ct}$ levels. Error bars indicate mean ± SEM. Data are collected from three individual experiments. Significance was determined by one-way ANOVA test to compare all groups with the individual variable. Data distribution was assumed to be normal but was not formally tested. * = P < 0.05, ** = P < 0.01, *** = P < 0.001, and **** = P < 0.0001; nonsignificant differences are not depicted. **(E)** Representative dot plots showing the region selected for FACS analysis from control and snx10ab DKO zebrafish larvae at 3 dpf using MitoSOX reagent. $H_2O_2$ was added to the water for 1 h as a positive control. **(F)** Representative FACS plot showing oxidative stress in control and snx10ab DKO zebrafish larvae at 3 dpf using the MitoSOX reagent. $H_2O_2$ added in water served as positive control.

Video 1.   **Time-lapse fluorescence imaging of SNX10-EGFP–expressing U2OS cells treated with MitoTracker Red.** This movie depicts live imaging of U2OS cells with inducible expression of SNX10-EGFP and treated with MitoTracker Red 30 min prior to imaging. The acquisition lasted 2 min at intervals of 500 ms, and the playback rate is six frames per second (fps). Scale bar = 10 μm. Related to Fig. 3 C.

**Provided online is Table S1. Table S1 summarizes the plasmids utilized throughout the study, detailing their names and respective descriptions.**

