## [Peer Review File · The Journal of Cell Biology]

SNX10 functions as a modulator of piecemeal mitophagy and mitochondrial bioenergetics

Laura Trachsel-Moncho, Chiara Veroni, Benan Mathai, Ana Lapao, Sakshi Singh, Nagham Asp, Sebastian Schultz, Serhiy Pankiv, and Anne Simonsen

Corresponding Author(s): Anne Simonsen, Oslo University Hospital

Review Timeline:

Submission Date:	2024-04-05
Editorial Decision:	2024-05-13
Revision Received:	2024-12-20
Editorial Decision:	2025-01-23
Revision Received:	2025-01-31

Monitoring Editor: Michael Lazarou

Scientific Editor: Andrea Marat

Transaction Report:

DOI: <https://doi.org/10.1083/jcb.202404009>

May 13, 2024

Re: JCB manuscript #202404009

Prof. Anne Simonsen

University of Oslo

Centre for Cancer Cell Reprogramming, Division for Cancer Medicine, Inst for Clinical Medicine and Dept of Molecular Cell Biology, Inst for Cancer Research, Oslo University Hospital

The Norwegian Radium Hospital, Montebello,

Oslo N-0379

Norway

Dear Prof. Simonsen,

Thank you for submitting your manuscript entitled "SNX10 regulates the clearance of mitochondrial proteins and mitochondrial bioenergetics". The manuscript was assessed by expert reviewers, whose comments are appended to this letter. We invite you to submit a revision if you can address the reviewers' key concerns, as outlined here.

You will see that the reviewers all appreciate your study provides interesting observations regarding a role for SNX10 in endosomal and mitochondrial functions. While they have noted that the molecular roles of SNX10 are not clearly established by the current work, we do not expect a precise mechanism to be established in a revised study. The reviewers have provided constructive experimental suggestions, which should however be attempted as this will provide greater insight into your model.

Specifically, in a revised manuscript it is essential to address: Rev 1 comments 2B (clarifying whether the structures might be MDVs) and 4 (having a model figure) are quite important. Rev 2, a response and or clarification to point 5 is very important, also very important is point 6, and then RNAi rescue or KOs highlighted by Rev 2 (7) and Rev 3 (15).

While Rev 1 Point 2a is interesting, it is not essential, but it would at least require addressing through re-writing of the discussion or consideration in the discussion that an iron chelator was used and how this might be involved in the interpretation of SNX10 trafficking iron.

Otherwise, we hope you will be able to address the remaining reviewer comments in a revised study.

GENERAL GUIDELINES:

Text limits: Character count for an Article is < 40,000, not including spaces. Count includes title page, abstract, introduction, results, discussion, and acknowledgments. Count does not include materials and methods, figure legends, references, tables, or supplemental legends.

Figures: Articles may have up to 10 main text figures. Figures must be prepared according to the policies outlined in our Instructions to Authors, under Data Presentation, <https://jcb.rupress.org/site/misc/ifora.xhtml>. All figures in accepted manuscripts will be screened prior to publication.

Supplemental information: There are strict limits on the allowable amount of supplemental data. Articles may have up to 5 supplemental figures. Up to 10 supplemental videos or flash animations are allowed. A summary of all supplemental material should appear at the end of the Materials and methods section.

Please note that JCB now requires authors to submit Source Data used to generate figures containing gels and Western blots with all revised manuscripts. This Source Data consists of fully uncropped and unprocessed images for each gel/blot displayed in the main and supplemental figures. Since your paper includes cropped gel and/or blot images, please be sure to provide one Source Data file for each figure that contains gels and/or blots along with your revised manuscript files. File names for Source Data figures should be alphanumeric without any spaces or special characters (i.e., SourceDataF#, where F# refers to the associated main figure number or SourceDataFS# for those associated with Supplementary figures). The lanes of the gels/blots should be labeled as they are in the associated figure, the place where cropping was applied should be marked (with a box),

and molecular weight/size standards should be labeled wherever possible.

The typical timeframe for revisions is three to four months. While most universities and institutes have reopened labs and allowed researchers to begin working at nearly pre-pandemic levels, we at JCB realize that the lingering effects of the COVID-19 pandemic may still be impacting some aspects of your work, including the acquisition of equipment and reagents. Therefore, if you anticipate any difficulties in meeting this aforementioned revision time limit, please contact us and we can work with you to find an appropriate time frame for resubmission. Please note that papers are generally considered through only one revision cycle, so any revised manuscript will likely be either accepted or rejected.

Thank you for this interesting contribution to Journal of Cell Biology. You can contact us at the journal office with any questions at cellbio@rockefeller.edu.

Sincerely,

Michael Lazarou, PhD
Monitoring Editor

Andrea L. Marat, PhD
Senior Scientific Editor

Journal of Cell Biology

Reviewer #1 (Comments to the Authors (Required)):

This study describes a novel role for sorting nexin 10 (SNX10) in influencing mitochondrial biology. The authors begin by finding that SNX10 co-localizes with EEA1 in a manner that is dependent on VPS34 activity, suggesting its PI(3)P dependence. Consistently, the authors find that RNAi-mediated knockdown of SNX10 alters EGF-simulated internalization of EGFR - a process well known to depend on endosomal trafficking. To further understand the functions of SNX10, the authors transitioned to a discovery-based approach using GFP-trap pulldown experiments to identify SNX10 interaction partners. Surprisingly, the authors find an enrichment in mitochondrial, autophagic, and endolysosomal targets, suggesting that SNX10 may play a role in mitophagy. The remainder of the manuscript focuses on characterizing the intersection between SNX10 and mitochondrial biology. First, the authors find that fluorescently labeled SNX10 colocalizes with mitochondria and LC3, an effect that is enriched upon treatment with the mitophagy-inducing compound DFP. Consistently, the authors find select mitochondrial markers associated with SNX10-positive vesicles upon DFP treatment, but perhaps surprisingly, SNX10 RNAi reduced the protein expression of multiple mitochondrial markers, as well as mitochondrial content and respiratory activity. Interestingly, these phenotypes seem to be independent of classical mitophagy, as there is no alterations in a pH-sensitive fluorescence reporter in SNX10 knockdown cells, nor is the loss of mitochondrial proteins rescued by autophagic inhibitors. Finally, the authors find a conserved role for SNX10 in mitochondrial homeostasis, as CRISPR mediated KO of both orthologs of *snx10* in zebrafish decreases mitochondrial protein levels and causes elevated cell death and ROS. The manuscript is interesting but can be difficult to follow as the model proposed for SNX10 function is unclear.

Major points:

1. In Figures 1 and 3, the authors demonstrate that SNX10-EGFP co-localizes with EEA1 and Rab5, demonstrating its presence on early endosomes. In Figure 3, they show that SNX10 associates with LC3, and in Figure 4, SNX10 is shown to co-localize with COX-IV under DFP treatment. It is unclear, however, whether the authors believe these populations of SNX10 are independent of one another (i.e., a subpopulation of SNX10 localizes to early endosomes and a separate, distinct population localizes to LC3-positive vesicles that contain mitochondrial material), or whether these are the same vesicular structures caught at different timepoints within their "life cycle". If the latter, do the authors propose directionality to this process? i.e., does SNX10 start out in early endosomes and, upon interaction with mitochondria, recruit LC3? Or does SNX10 interact with mitochondria

and deliver the contents to early endosomes? Perhaps co-staining for these various markers together could clarify how SNX10 is distributed, which could inform its underlying mechanism.

2. The authors propose in the discussion that SNX10-depleted cells may be undergoing a non-canonical type of mitophagy. This is due to the lack of full rescue of COXIV in the presence of autophagic inhibitors, as well as the lack of phenotype seen with the pH-sensitive GFP/mCherry reporter (which I'm assuming is similar to or analogous to mito-QC). The authors should further explore this claim:

2a. The authors heavily use DFP, an iron chelator, to induce PINK1/parkin-independent mitophagy. However, they also comment in the discussion that SNX10 may play a role in the trafficking of iron between endosomes and mitochondria, which would likely be influenced by DFP and in a manner independent of mitophagy, consistent with the authors' model. The authors could delineate the mitophagic contributions versus the potential iron chelation contributions of these treatments by ablating expression of BNIP3/NIX, which are well-known targets of DFP through pseudohypoxia (PMID: 32420530), or by inducing BNIP3/NIX expression independent of iron chelation, such as through bona fide hypoxia, to see if the phenotypes seen in the presence of DFP persist. If not, the authors will have identified another set of proteins within the SNX10 pathway. If so, this is further evidence to substantiate their claims of non-canonical mitophagy. The authors could also test the role of the PINK1/parkin pathway (albeit indirectly) through the use of mitochondrial toxins such as antimycin A or CCCP.

2b. The intersection between SNX10 and the endosomal pathway, along with its lack of full response to autophagic inhibition may be consistent with SNX10 influencing the formation of mitochondrial derived vesicles (MDVs). Do the authors see evidence of MDV formation in any of their EM work? Even if not, they could comment on the evidence for or against the model that SNX10 influencing MDV formation (even if speculative).

2c. The interaction studies in Figure 3 identify p62/SQSTM1 as an interaction partner of SNX10. Are the SNX10 phenotypes seen dependent on p62 expression/activity?

3. The authors use total citrate synthase (CS) activity to measure mitochondrial content. While well established, evidence exists that CS does not always correlate with total mitochondrial content (PMID: 33077793). The authors should bolster this claim with orthogonal methods, such as mitotracker staining or the use of additional mitochondrial markers. Additionally, the authors should western blot for CS, as its reduction in activity in SNX10 knockdown cells could be due to a loss of total protein, which could implicate it as a potential SNX10 target.

4. The model put forth by the authors is difficult to grasp, and clarifying the role that SNX10 plays in mitochondrial biology would be helpful. The authors propose that SNX10 does many things (e.g., endosomal maturation, housing mitochondrial cargo, non-canonical autophagy, influencing mitochondrial functions), but how this occurs collectively is not clear. For instance, SNX10 co-localizes to LC3-positive vesicles that contain mitochondrial content, which would be consistent with a positive role in mitophagy. However, loss of SNX10 depletes mitochondrial content, which would be consistent with a negative role in mitophagy. Having a clear model figure of what is shown for SNX10 functionality in this manuscript would be really helpful to the reader. Additionally, more discussion on the intersection of the phenotypes seen in the paper is warranted.

Minor points:

1. In the abstract, the authors state "SNX10-positive vesicles contain COX-IV and SAMM50, both proteins being important for mitochondrial respiratory chain function, while other mitochondrial proteins are excluded." This should be amended to "while other tested mitochondrial proteins are excluded", as the authors have performed experiments on limited SNX10 mitochondrial cargos.

2. The section titles, as well as the title of the paper claiming SNX10 regulation of mitochondrial functions should be reworded, as the authors have not shown direct regulation of these processes. Perhaps something along the lines of "Loss of SNX10 decreases mitochondrial bioenergetics" rather than "SNX10 regulates mitochondrial bioenergetics" would be more appropriate.

Reviewer #2 (Comments to the Authors (Required)):

In the manuscript by Trachsel-Moncho et al., the authors examine the role of SNX10 in endosome and mitochondrial function. SNX10 was identified as a potential mitophagy regulator from a previously published screen from the lab and now the authors characterize its role in greater detail here. SNX10 localises primarily to early endosomal structures in a manner dependent on VPS34 activity and its loss impaired endosome maturation and EGFR trafficking. Interestingly the authors find close association of SNX10 +ve compartments with mitochondria and even mitochondrial markers inside of them. siRNA mediated depletion of SNX10 results in loss of COX-IV protein but no significant changes were observed with other mitochondrial markers analysed. How this loss occurs is not clear. Regardless, disruption of SNX10 reduces mitochondrial respiration and this also may be the case in vivo, using a Zebra fish CRISPR model.

This is certainly an interesting manuscript, and the use of cell biology and in vivo work nicely highlights the complexity of SNX10 role(s), including an unexpected role in mitochondrial function. However, I was a little confused over the mechanism of SNX10 in endosomal vs mitochondrial regulation and whether these were two separate or the same pathway.

Main points:

- 1) From the data in Fig.1, I get the impression that the majority of SNX10 is on early endosomes, with a lesser amount on later structures. Can the authors quantify co-localisation with the markers to confirm this?
- 2) In Figure 2, the authors use EGFR trafficking and propose that loss of SNX10 impairs endosomal maturation. If this is the case, then EGFR degradation in lysosomes should be impaired following EGF stimulation. However, the authors do not see any turnover within the 60 min tested and should perform a longer CHX chase time.
- 3) The authors perform IP mass-spec of SNX10 to identify potential SNX10 interactors. Have the authors been able to independently validate any of these hits? The authors go on to imply mitochondrial binding, so can any of the mitochondrial proteins be detected by western blot of SNX10 IPs?
- 4) The authors very nicely show close proximity of the SNX10 compartment with mitochondria and even structures that contain mitochondria (mitotracker stain). Are these early endosomes or late endosomes or another compartment? If they are endosomal, does loss of SNX10 reduce mitochondrial proximity/engulfment?
- 5) In Fig.4, and related to above point, the authors find multiple mitochondrial components within the SNX10 compartment, yet loss of SNX10 only appears to affect COX-IV. This suggests to co-localization is just a result of mitophagy processes transiting endosomal (and hence SNX10+ve) compartments, rather than any direct involvement of SNX10 in the process. Indeed mitophagy, by reporter assay, is not altered by loss of SNX10. It was not clear if this was the point the authors were trying to make, or something else. Perhaps this could be better explained in the text.
- 6) In Fig.4F, the authors show loss of COX-IV but where is it going? The authors could include bafilomycin and a proteasome inhibitor, as well as another mitochondrial marker as a control. This could help with mechanism as to why only this protein is significantly reduced upon SNX10 loss. If the additional treatments are negative, could COX-IV be being secreted via an MDV/exosome-like pathway?
- 7) In Fig. 5, the authors see significant effects with one siRNA but not the other. This makes it difficult to draw any firm conclusions. Can the authors knock out SNX10 by CRISPR or rescue effects by re-expression of siRNA resistant forms?
- 8) In Fig.6D and E, the blots of SNX10 and COXIV are very weak and hard to make out by eye - are there any "long exposures"?
- 9) The increase in cell death in the SNX10 KO larvae is interesting and the authors propose that this is caused by increased ROS. If this is the case, then can cell death be rescued via ROS scavengers/anti-oxidants?
- 10) Do the authors think the effect of SNX10 loss on EGFR is related to the mitochondrial phenotype? Some discussion would be helpful.

Minor points

- 11) Many of the charts indicate differences but lack statistics.
- 12) Outside of the initial mitophagy analyses, I found the use of DFP and DFO a little confusing as there was no differences between controls and SNX10 depletions. For simplicity, the authors could consider removing these data.

Reviewer #3 (Comments to the Authors (Required)):

This manuscript explores the role of the PX domain-containing protein sorting nexin 10 (SNX10) in both endosomal trafficking and mitochondrial abundance. Understanding SNX10's biological function is important due to its involvement since it is known to be mutated in autosomal recessive osteopetrosis.

While the figures in this manuscript are visually appealing, the manuscript in its current form would benefit from greater cohesion to present a clear model or mechanism. Much of the data is observational, and at times, the interpretation of results can be confusing. To enhance clarity and ensure accurate alignment between conclusions and presented data, I offer several suggestions for improvement below:

Specific comments

1. Figure 1 shows that SNX10 localises to early and late endosomes in a PtdIns3P-dependent manner.

Point 1. It is possible that vacuoles also form in SNX10 ARO mutant expressing cells but are not visible due to the low expression level of GFP (or that they form but the GFP-SNX10 does not localise there). Suggest rewriting the interpretation of the data for accuracy. Can the vacuoles be observed using another marker in the mutant-expressing cells? To me, this is a minor point and I think removing the statement about vacuoles not forming in the mutant cells would suffice.

Point 2. Including quantification representing the colocalization of SNX10 with various Rab proteins would improve the robustness of the paper.

2. Figure 2 explores SNX10's role in endosomal trafficking, however, the supporting text did not clearly articulate the role of SNX10 in this process.

Point 3. Clarify the accompanying text in the results section for the role of SNX10 in endosomal maturation. I am not convinced that the data presented is conclusive to demonstrate a role for SNX10 in endosomal maturation or trafficking since I would think that this should include a more thorough characterisation of hallmarks of endosomal maturation such as Rab switch, lipid status

and acidification status. Can the authors provide more data to support their statements? Alternatively, moderate the interpretation of the data.

Point 4. WB figure 2 C-D. The statistical test is mentioned in the figure legend, but the result is missing from the figure. If non-sig, include on the graph.

Point 5. Please include an image analysis section in the methods section describing the quantification of the EEA1 for this graph (applies to some other graphs as well).

Point 6. Regarding the EM analysis, I was confused about matching the figure legend text with the graph since the gold-negative and positive endosomes were not separated on the graph. Please clarify.

Point 7. Supplementary Figure 2. Suggest rewriting the figure legend of A for clarity - which SNX10 sirna is used to measure ulk1 levels? Or is it the ULK1 levels being measured in the ULK1 column?

3. Figure 3 highlights the interactome of SNX10.

Point 8. A more comprehensive description of the quantification of the mass spec is required to interpret the proteomic data. Was this done in triplicate? How was the control used to remove background proteins? Was the control GFP alone or the disease-associated mutant SNX10 - the figure suggests it is the mutant SNX10 but the text suggests it is GFP alone. Was there any statistical analysis performed to demonstrate that SNX10 interacts with mito proteins?

Point 9. The conclusion that SNX10 "interact" with mitochondria should be tempered. The data showing that SNX10 and mitochondria are near each other is not conclusive given the widespread localisation of mitochondria throughout the cytoplasm. Suggest changing "interact" to localise nearby.

The data showing that mitochondrial proteins are inside SNX10-positive vesicles is convincing, but I think it is important to clarify in method section that these particular images were acquired by confocal.

Point 10. Supplementary figure 2B. Do SNX10 protein levels change with DFP?

Point 11. The text at the end of this section describing figure 3 "Taken together, our data suggest a role for SNX10 in mitophagy" is confusing to me since there is no direct evidence for this presented in the figure (i.e. would need to deplete SNX10 and show an increase or decrease in mitophagy). The increased LC3-SNX10 colocalisation could be due to the increased LC3 in the DFP condition. Data suggests that SNX10 localises to the mitophagosome however there is no data to suggest it functions there.

4. Figure 4 describes the effect of SNX10 depletion on levels of mitochondrial proteins in steady state and after DFP.

Point 12. In my opinion, although SNX10 may indeed localise to structures that contain mitochondria, the functional depletion of SNX10 does not significantly change mitochondrial protein abundance. The WB for COXIV does not reflect the IF images presented. Is this the most representative WB to show? Is there another exposure?

Point 13. Supplementary Figure 3A. line intensity scan - specify if the left or right Y axis is green or magenta pixel intensity.

Point 14. Supplementary Figure 3C-D. Include a description in methods section or figure legend about how mitolysosomes were measured - is it the magenta only signal or a ratiometric analysis. Stats mentioned in figure legend but not on the graph. If non-sig, include on the graph.

5. Figure 5 looks at the depletion of SNX10 and mitochondrial function.

Point 15. Since only one siRNA appears to have a significant effect (that is my interpretation from the graph but I may be wrong), I believe it is especially important to rescue this phenotype using the si-resistant construct or demonstrate with another siRNA. To understand the disease-relevance of the pathway, the authors could rescue with the disease-associated mutants.

6. Figure 6 - zebrafish system.

Point 16. The interpretation that SNX10 mediated degradation of inner membrane proteins is not demonstrated by the data presented, therefore the text should be rewritten. It could be through transcription, loss of biogenesis etc.

Discussion point: is osteopetrosis seen in mito disease?

Rebuttal letter JCB manuscript #202404009

Dear Editor and Reviewers,

We want to express our gratitude for the thoughtful and constructive feedback on our manuscript titled "SNX10 regulates the clearance of mitochondrial proteins and mitochondrial bioenergetics." We appreciate the time and effort the reviewers have invested in evaluating our work, and we are grateful for the opportunity to revise and improve our manuscript. We have carefully considered each comment and made significant revisions in response, which we believe have strengthened the overall quality and clarity of the paper. Below is our detailed point-by-point response to the reviewer's comments and concerns. Please also find attached a list of all new figures and figure modifications done.

Reviewer #1 comments

Comment 1: In Figures 1 and 3, the authors demonstrate that SNX10-EGFP co-localizes with EEA1 and Rab5, demonstrating its presence on early endosomes. In Figure 3, they show that SNX10 associates with LC3, and in Figure 4, SNX10 is shown to co-localize with COX-IV under DFP treatment. It is unclear, however, whether the authors believe these populations of SNX10 are independent of one another (i.e., a subpopulation of SNX10 localizes to early endosomes and a separate, distinct population localizes to LC3-positive vesicles that contain mitochondrial material), or whether these are the same vesicular structures caught at different timepoints within their "life cycle". If the latter, do the authors propose directionality to this process? i.e., does SNX10 start out in early endosomes and, upon interaction with mitochondria, recruit LC3? Or does SNX10 interact with mitochondria and deliver the contents to early endosomes? Perhaps co-staining for these various markers together could clarify how SNX10 is distributed, which could inform its underlying mechanism.

Response: We thank the reviewer for these important questions. We have used several approaches to further investigate the nature of SNX10-positive vesicles and a possible functional directionality for SNX10-positive structures in response to hypoxia-mimicking conditions. First, to validate our findings with DFP (an iron chelator), we have used DMOG (Dimethylloxalylglycine), another hypoxia-mimicking drug that stabilizes HIF1A and induces mitophagy. We observed similar dynamic kiss-and-run interactions of SNX10-positive endosomes with mitochondria (MitoTracker and COX-IV staining) in response to DMOG treatment as with DFP (Figs. 4 and 5). We also show that SNX10-positive structures are associated with EEA1 and LC3B, as well as CD63 and LAMP1, in DMOG-treated cells (Figs. 4 and 5).

To address whether these populations of SNX10-positive structures are independent of each other and if there is any directionality to the process, we carried out additional experiments where SNX10-EGFP cells were treated or not with DFP or DMOG and stained for EEA1, LC3B, and mitochondria (MitoTracker) (Fig. 4 A-B) or with CD63, LC3B and Mitotracker (Fig. 4 C-D). Under control conditions, we observed little to no MitoTracker or LC3 co-occurrence with SNX10 (Fig. 4 A-B). Following DFP or DMOG treatment, SNX10 vesicles were found to contain mitochondrial material and stain positive for LC3B, with noticeably reduced co-localization with EEA1 (Fig. 4 A-B) and increased co-localization with CD63 (Fig. 4 C-D). Moreover, when

staining SNX10-EGFP cells with antibodies against endogenous COX-IV and LC3 or LAMP1, we observed COX-IV colocalization with LC3 and LAMP1 upon DFP or DMOG treatment (Fig. 5 B-E). Thus, our data suggest that under hypoxia-mimicking conditions, SNX10-positive vesicles undergo a functional shift from early endosomes to later endocytic structures, incorporating mitochondrial material and LC3, likely reflecting a change in vesicular identity.

In line with this, the co-localization of SNX10 with RAB7 was higher in DFP-treated cells compared to control conditions (data not included in the manuscript, see *Reviewer Figure 1* below).

Reviewer Figure 1: U2OS cells stably transfected with SNX10-GFP and mScarlet-RAB7 were imaged live after treatment (or not) with DFP for 24h. (scale bar = 10 μ m)

Comment 2a: The authors heavily use DFP, an iron chelator, to induce PINK1/parkin-independent mitophagy. However, they also comment in the discussion that SNX10 may play a role in the trafficking of iron between endosomes and mitochondria, which would likely be influenced by DFP and in a manner independent of mitophagy, consistent with the authors' model. The authors could delineate the mitophagic contributions versus the potential iron chelation contributions of these treatments by ablating expression of BNIP3/NIX, which are well-known targets of DFP through pseudohypoxia (PMID: 32420530), or by inducing BNIP3/NIX expression independent of iron chelation, such as through bona fide hypoxia, to see if the phenotypes seen in the presence of DFP persist. If not, the authors will have identified another set of proteins within the SNX10 pathway. If so, this is further evidence to substantiate their claims of non-canonical mitophagy. The authors could also test the role of the PINK1/parkin pathway (albeit indirectly) through the use of mitochondrial toxins such as antimycin A or CCCP.

Response. We acknowledge that our original manuscript heavily relied on DFP to induce PINK1/parkin-independent mitophagy and the possible limitations of that. In response to the reviewer's suggestion, we have now used DMOG, another inducer of BNIP3/NIX expression and PINK1/parkin-independent mitophagy, to validate the main phenotypes seen in the presence of DFP (Figs. 4, 5 and 7). As we observe a similar effect of SNX10-depletion on the

turnover of mitochondrial material (MitoTracker, COX-IV, and pSu9-Halo-mGFP) both in DFP and DMOG-treated cells, we conclude that the role of SNX10 in the turnover of mitochondrial components is independent of iron levels or iron transport.

To monitor mitophagy more directly, we have now employed the reporter pSu9-Halo-mGFP (PMID: 35938926) that allows measurement of mitophagy flux as the Halo-tag is stable in lysosomes when bound to its ligand. We generated U2OS cells with stable expression of pSu9-Halo-mGFP and treated the cells (or not) with DFP or DMOG for 24 hrs after the addition of a Halo ligand (TMR) for 20 min. The ratio of cleaved versus full-length pSu9-Halo-mGFP was analyzed by Western blotting (relative to a loading control), demonstrating an increase in the lysosomal degradation of pSu9-Halo-mGFP (mitophagy) under both DFP and DMOG conditions in cells depleted of SNX10 compared to siCtrl (Fig. 7A-C).

We also appreciate the reviewer's suggestion to ablate the expression of BNIP3/NIX to further delineate the contributions of iron chelation/hypoxia-induced mitophagy versus non-canonical mitophagy within the SNX10 pathway. However, neither depletion of BNIP3 nor NIX alone prevented DFP- or DMOG-induced mitophagy in U2OS cells (data not shown), indicating that both genes likely need to be depleted to inhibit mitophagy. Therefore, we decided instead to use DMOG as an alternative approach to induce mitophagy.

The reviewer also suggests that we address the potential role of SNX10 in the PINK1/parkin pathway. U2OS cells stably expressing iMLS (a matrix localized EGFP-mCherry-tagged reporter) and Parkin were transfected with siCtrl or siSNX10 before treatment with/without CCCP, followed by automated imaging and analysis of the level of mitolysosomes (as defined as red-only area per cell, using a Cell Profiler pipeline established in the lab). In these experiments, we observed a strong induction of mitophagy both in control and SNX10-depleted cells, with no major differences in mitophagy levels in SNX10-depleted cells compared to control cells (Fig. S3B).

Although we can largely exclude an iron-chelation effect underlying the role of SNX10 in mitochondrial turnover (as we see similar effects of DFP and DMOG), we anyway decided to analyze the potential role of SNX10 in the trafficking of iron between endosomes and mitochondria. Cells expressing SNX10-EGFP were incubated with Transferrin-FarRed to label early and recycling endosomes. Intriguingly, we could observe transferrin in punctate structures near or within the limiting membrane of SNX10-positive vesicles, occasionally in contact with mitochondria (see arrow in *reviewer Figure 2* below). This data has not been included in the revised manuscript and we have toned down the discussion of this point.

Reviewer Figure 2: U2OS cells stably transfected with SNX10-GFP were imaged live after the addition of MitoTracker and AlexaFluor647-Transferrin to the media. Arrows point to transferrin co-occurrence with both SNX10 and mitochondria.

Taken together, our data indicates that SNX10 regulates the turnover of selected mitochondrial proteins under hypoxia-mimicking conditions in a PINK1/parkin-independent manner.

Comment 2b: The intersection between SNX10 and the endosomal pathway, along with its lack of full response to autophagic inhibition may be consistent with SNX10 influencing the formation of mitochondrial derived vesicles (MDVs). Do the authors see evidence of MDV formation in any of their EM work? Even if not, they could comment on the evidence for or against the model that SNX10 influencing MDV formation (even if speculative).

Response. As suggested, we re-examined the prepared EM samples and indeed can find vesicles in close proximity to mitochondria (see *Reviewer Figure 3* below). However, with the current samples it was not possible to be certain if these vesicles a) are connected to the mitochondria and therefore derived from the adjacent mitochondria or b) if seemingly connected vesicles were MDVs or represented normal branching of mitochondria (see white arrow in image below). In general, EM is not well suited to establish if there is a quantitative difference in MDV frequency in control cells vs siSNX10 cells as this would require immuno-

EM to define the identity of the observed vesicles, 3D imaging and a large number of cells for quantification.

Reviewer Figure 3: U2OS cells stably transfected with SNX10-GFP were imaged live after addition of Mitotracker and AlexaFluor647-Transferrin to the media. Arrows pointing to transferrin interacting with both SNX10 and mitochondria.

We therefore used an alternative approach to investigate whether SNX10 could affect MDV formation. Control and SNX10 depleted U2OS cells were fixed and stained with antibodies recognizing endogenous TOMM20 and PDH followed by imaging and image analysis to quantify vesicles that are positive for either TOMM20 or PDH (according to PMID: 22226745). As shown in Fig. S3C-D, there is no significant difference in the number of MDVs between control and SNX10-depleted cells.

Comment 2c: The interaction studies in Figure 3 identify p62/SQSTM1 as an interaction partner of SNX10. Are the SNX10 phenotypes seen dependent on p62 expression/activity?

Response: We thank the reviewer for this suggestion. To address a potential role of p62/SQSTM1 in the observed SNX10 phenotypes, we first asked whether p62 colocalized with SNX10. SNX10-EGFP cells were treated or not with DFP before fixation, staining with antibodies against COX-IV and p62, and confocal imaging. While there was little or no colocalization of p62 with SNX10 in control cells, p62 colocalized with SNX10 vesicles containing COX-IV in DFP-treated cells (Fig. 7D).

p62 has previously been found to be required for piece-meal mitophagy of selected mitochondrial proteins (PMID: 29149599, PMID: 34037656). The co-localization of p62 with COX-IV and SNX10 in DFP-treated cells, in combination with the increased lysosomal turnover

of COX-IV and the mitophagy reporter pSu9-Halo observed in SNX10 depleted cells (Figs. 6 and 7) prompted us to address whether p62 is required for COX-IV clearance. Indeed, the level of COX-IV was significantly increased in cells upon siRNA-mediated depletion of p62 (Fig. 7E-F).

Comment 3: The authors use total citrate synthase (CS) activity to measure mitochondrial content. While well established, evidence exists that CS does not always correlate with total mitochondrial content (PMID: 33077793). The authors should bolster this claim with orthogonal methods, such as mitotracker staining or the use of additional mitochondrial markers. Additionally, the authors should western blot for CS, as its reduction in activity in SNX10 knockdown cells could be due to a loss of total protein, which could implicate it as a potential SNX10 target.

Response: To address whether SNX10 knockdown affects citrate synthase (CS) expression and/or activity, we performed Western blot analysis on samples from control (siCtrl) and SNX10 knockdown cells (using two different siRNA targeting SNX10, siSNX10#1 and siSNX10#2). The results (from 3 independent experiments) showed no significant differences in CS protein levels between the control and knockdown conditions (Fig. 8E-F), indicating that the observed reduction in citrate synthase activity in SNX10 depleted cells (Fig. 8D) is not due to a loss of total CS protein.

To measure mitochondrial content, we also stained control and SNX10-depleted cells for endogenous TOMM20 and used immunofluorescence microscopy and automated image analysis to measure the area covered by TOMM20 (Fig. S2 B-C). Consistent with the Western blot results, there was no reduction in TOMM20 staining in SNX10-depleted cells. In contrast, the level of COX-IV was reduced in SNX10-depleted cells, as assessed by immunofluorescence microscopy staining of endogenous protein (Fig. 6E-F and 7G-I) and western blot analysis (Fig. 6 A and D), indicating that the increased COX-IV turnover seen in SNX10-depleted cells is not due to a general reduction of mitochondrial content.

Comment 4: The model put forth by the authors is difficult to grasp, and clarifying the role that SNX10 plays in mitochondrial biology would be helpful. The authors propose that SNX10 does many things (e.g., endosomal maturation, housing mitochondrial cargo, non-canonical autophagy, influencing mitochondrial functions), but how this occurs collectively is not clear. For instance, SNX10 co-localizes to LC3-positive vesicles that contain mitochondrial content, which would be consistent with a positive role in mitophagy. However, loss of SNX10 depletes mitochondrial content, which would be consistent with a negative role in mitophagy. Having a clear model figure of what is shown for SNX10 functionality in this manuscript would be really helpful to the reader. Additionally, more discussion on the intersection of the phenotypes seen in the paper is warranted.

Response: We apologize for not including a model figure in the original manuscript (now included as Fig. 10). We propose that SNX10 functions as a negative regulator of piecemeal mitophagy of components of the OXPHOS machinery (including COX-IV and ATP synthase subunits), allowing the cell to regulate oxidative phosphorylation in response to metabolic needs without compromising overall mitochondrial structure. This is based on the following observations:

- i) Endogenous COX-IV, SAMM50, and MitoTracker staining inside SNX10-positive vesicles that are also positive for LC3B, p62 and CD63 upon induction of hypoxia-induced mitophagy using DFP or DMOG (Fig. 3, 4 and 5).
- ii) Increased turnover of endogenous COX-IV (by western blot and immunofluorescence, Fig. 6), as well as a Halo-tagged Su9 (a subunit of the ATP synthase, Fig. 7) in cells lacking SNX10. The levels of Cox-IV and Samm50 are also reduced in zebrafish larvae (Fig. 9).
- iii) SNX10 depletion correlates with reduced respiration, ATP production and citrate synthase activity in cells (Fig. 8), and increased ROS and cell death in zebrafish larvae, which is prevented by the addition of the antioxidant NAC (Fig. 9).
- iv) While SNX10 colocalizes with both EEA1 (early endosomes) and CD63 (late endosomes) in control conditions, there is a shift in the SNX10-positive population to CD63-positive structures containing mitochondrial material and LC3B in cells treated with the hypoxia-mimicking drugs DFP or DMOG. Intriguingly, whereas lysosomal turnover of mitochondrial components increases in SNX10-depleted cells, cells lacking SNX10 show an accumulation of endocytic vesicles and delayed EGFR degradation (Fig. 2), suggesting that SNX10 control cargo trafficking in the endocytic pathway. The exact mechanism underlying this phenotype is not clear, but the presence of enlarged LAMP1-positive vacuoles in cells expressing wild-type SNX10 suggests that SNX10 may regulate fission/fusion events in the endocytic pathway. SNX10 is the smallest of the PX domain proteins, containing only a PX domain plus a short IDR (intrinsically disordered region). The fact that mutations in its PX domain are causative of autosomal recessive osteopetrosis (ARO), a life-threatening rare type of skeletal dysplasia characterized by increased bone density, demonstrates an essential role for this protein.
- v) The fact that SNX10 co-localizes with LC3B (Figs. 3E-G, 4 and 5B-C) and p62 (Fig. 7D), combined with the increased turnover of the inner mitochondrial membrane (IMM) protein pSu9-Halo-EGFP (Fig. 7A-C), with no effect on a mCherry-EGFP-tagged matrix marker (Fig. S3A-B) in SNX10-depleted cells, indicate that SNX10 functions in piecemeal mitophagy. p62/SQSTM1 is known to be required for piecemeal mitophagy (PMID: 29149599, PMID: 34037656). Indeed, we show that p62 colocalizes with COX-IV containing SNX10-positive vesicles (Fig. 7D) and that the COX-IV level is significantly increased in cells lacking p62 (Fig. 7E-F). To our surprise, neither the DFP- nor the siSNX10-induced degradation of COX-IV (Fig. 7) or the co-localization of SNX10 with LC3B (Fig. 3E-G) were reduced by inhibition of the core autophagy machinery components ULK1 or VPS34. The two previous papers published on piecemeal mitophagy (PMID: 29149599, PMID: 34037656) did not investigate the need for these core autophagy machinery components, and it is therefore likely that piecemeal mitophagy may be independent of the ULK1 and VPS34 complexes.

The seemingly selective effect of SNX10 on the turnover of proteins involved in mitochondrial oxidative phosphorylation and ATP production is reminiscent of the recently described VDIM (Vesicles Derived from the Inner Mitochondrial membrane) pathway (PMID: 39169179). VDIMS are formed by IMM herniation through pores in the outer mitochondrial membrane, followed by their engulfment by lysosomes in proximity to mitochondria in a microautophagy-like process. VDIM

formation was found to increase upon oxidative stress, leading to selective degradation of IMM proteins (including COX4 and other proteins involved in OXPHOS) while sparing the remainder of the organelle. However, in contrast to MitoTracker/COX-IV containing SNX10-positive vesicles, VDIMs seem to lack LC3 and p62. We also show that mitochondria-derived vesicles form independently of SNX10 (Fig. S3C-D).

Comment 5: In the abstract, the authors state "SNX10-positive vesicles contain COX-IV and SAMM50, both proteins being important for mitochondrial respiratory chain function, while other mitochondrial proteins are excluded." This should be amended to "while other tested mitochondrial proteins are excluded", as the authors have performed experiments on limited SNX10 mitochondrial cargos.

Response: We agree that this was an overstatement, and we have now changed the text in the abstract to also include the new data. The corresponding sentence now reads "Upon hypoxia-mimicking conditions, SNX10 localizes to late endosomal structures containing selected mitochondrial proteins, including COX-IV and SAMM50, and the autophagy proteins SQSTM1/p62 and LC3B." We have also modified this statement throughout the text.

The section titles, as well as the title of the paper claiming SNX10 regulation of mitochondrial functions should be reworded, as the authors have not shown direct regulation of these processes. Perhaps something along the lines of "Loss of SNX10 decreases mitochondrial bioenergetics" rather than "SNX10 regulates mitochondrial bioenergetics" would be more appropriate.

Response: We agree with the reviewer that we do not have evidence for a direct role of SNX10 in the regulation of mitochondrial functions and we have now changed the title to "Identification of SNX10 as a novel modulator of piecemeal mitophagy and mitochondrial bioenergetics". We have also modified the section titles and text throughout the manuscript, e.g. by using the word "modulates/modulator" instead of "regulates/regulator".

Reviewer #2 comments

Comment 1: From the data in Fig.1, I get the impression that the majority of SNX10 is on early endosomes, with a lesser amount on later structures. Can the authors quantify co-localisation with the markers to confirm this?

Response: We have now carried out several new experiments to further characterize the localization of SNX10 in the endocytic pathway under control conditions and upon hypoxia-mimicking conditions (see also our response to comment 4 below). To quantify the colocalization of SNX10 to early and late endocytic structures, stable U2OS SNX10-EGFP cells were fixed and stained with antibodies against endogenous EEA1 and CD63, followed by imaging using a Nikon CREST X-Light V3 microscope. Quantification of colocalization was performed by segmentation of fluorescent signals using CellProfiler to obtain the percentage of CD63 or EEA1 structures that overlap with SNX10. When plotting the ratio of CD63 or EEA1

structures overlapping with SNX10 to the total number of SNX10-positive structures (Fig. 1 F-G), we find that SNX10 colocalizes with EEA1 and CD63 to a similar extent under basal conditions.

To determine the nature of the SNX10-positive vesicles containing mitochondrial material, SNX10-EGFP cells were treated or not with DFP or DMOG (another hypoxia-mimicking drug) and stained with antibodies against EEA1 or CD63, together with anti-LC3B and MitoTracker (Fig. 4). Under control conditions, we observed little to no MitoTracker or LC3 co-occurrence with SNX10 (Fig. 4 A-B). Following DFP or DMOG treatment, SNX10 vesicles were found to contain mitochondrial material and stain positive for LC3B, with noticeably reduced co-localization with EEA1 (Fig. 4 A-B) and increased co-localization with CD63 (Fig. 4 C). Moreover, when staining SNX10-EGFP cells with antibodies against endogenous COX-IV and LC3B or LAMP1, we observed COX-IV colocalization with LC3 and LAMP1 upon DFP or DMOG treatment (Fig. 5 B-E). Thus, our data suggest that under hypoxia-mimicking conditions, SNX10-positive vesicles undergo a functional shift from early endosomes to later endocytic structures, incorporating mitochondrial material and LC3, likely reflecting a change in vesicular identity.

Comment 2: In Figure 2, the authors use EGFR trafficking and propose that loss of SNX10 impairs endosomal maturation. If this is the case, then EGFR degradation in lysosomes should be impaired following EGF stimulation. However, the authors do not see any turnover within the 60 min tested and should perform a longer CHX chase time.

Response: We thank the reviewer for this suggestion. We have now repeated these experiments with longer CHX chase times (15, 30, 60, and 120 min) following EGF stimulation of control and SNX10-depleted cells. Indeed, while we see a time-dependent turnover of EGFR in control cells up to 120 min chase, the EGFR level remains high over time and is significantly increased in SNX10-depleted cells at 120 min compared to control (Fig. 2C-D). Moreover, the EGFR level is higher at time 0 in cells transfected with both SNX10 siRNA oligos.

Comment 3 The authors perform IP mass-spec of SNX10 to identify potential SNX10 interactors. Have the authors been able to independently validate any of these hits? The authors go on to imply mitochondrial binding, so can any of the mitochondrial proteins be detected by western blot of SNX10 IPs?

Response: We acknowledge the importance of validating potential interactors identified by mass spectrometry analysis of the SNX10-EGFP interactome. We have now performed SNX10-EGFP pulldown assays followed by Western blot analysis of a few of the proteins detected by mass spectrometry, including COX-IV, ATP5J, and p62/SQSTM1, but were unable to validate the interactions. However, as mass spectrometry is significantly more sensitive than Western blotting and also independent of functional antibodies, the lack of protein detection by Western blot does not necessarily indicate the absence of an interaction. We do observe dynamic interactions of SNX10 with mitochondria, suggesting that SNX10 interactions might be transient and challenging to capture through IP and Western blotting. The loss of COX-IV (Figs. 6-7 and 9) and the ATP synthase subunit pSu9-Halo-mGFP (Fig. 7A-C) observed upon SNX10 depletion supports a functional relationship between SNX10 and mitochondria. The autophagy receptor protein p62/SQSTM1 was also found among the SNX10 interactome (Fig.

3A), and we now show that p62 co-localizes with SNX10 on structures containing COX-IV (Fig. 7D) and that p62 is required for COX-IV turnover (Fig. 7E-F).

Comment 4: The authors very nicely show close proximity of the SNX10 compartment with mitochondria and even structures that contain mitochondria (mitotracker stain). Are these early endosomes or late endosomes or another compartment? If they are endosomal, does loss of SNX10 reduce mitochondrial proximity/engulfment?

Response: We thank the reviewer for these interesting questions. We have now carried out several new experiments to further characterize the localization of SNX10 in the endocytic pathway under control conditions and upon hypoxia-mimicking conditions.

SNX10-EGFP cells were treated or not with DFP or DMOG and stained for EEA1, LC3B, and mitochondria (Mitotracker) (Fig. 4A-B) or with CD63, LC3B and Mitotracker (Fig. 4C-D). Under control conditions, SNX10 co-localized both with EEA1 and CD63 (Fig. 1 E-F), with little to no observable Mitotracker or LC3B co-occurrence (Fig. 4A-D). Following DFP or DMOG treatment, SNX10 vesicles were found to contain mitochondrial material and stain positive for LC3B, with noticeably reduced co-localization with EEA1 (Fig. 4A-B) and increased co-localization with CD63 (Fig 4C-D). Thus, our data suggest that under hypoxia-mimicking conditions, SNX10-positive vesicles undergo a functional shift to incorporate mitochondrial material and LC3, likely reflecting a change in vesicular identity. In line with this, treatment with DFP in cells co-expressing SNX10-EGFP and mScarlet-RAB7 resulted in higher SNX10-RAB7 co-localization compared to control conditions (see *reviewer Figure 1* above, included in response to a similar comment from reviewer #1).

To address whether loss of SNX10 reduced the mitochondrial proximity and/or engulfment of mitochondrial material, cells with stable expression of mCherry-RAB5 were transfected with siCtrl or siSNX10 (two different oligos) and then treated or not with DFP, fixed and stained with antibodies against COX-IV. We do not see any difference in the mitochondrial proximity of RAB5 vesicles to mitochondria (anti-COX-IV staining) in SNX10-depleted cells compared to control cells (Fig. 6E). It is, however, difficult to determine from the observed co-occurrence of COX-IV with RAB5 whether COX-IV is engulfed in RAB5 vesicles or not (Fig. 6E), but we do see a significant reduction in COX-IV staining intensity (based on the immunofluorescence microscopy images) in SNX10-depleted cells (Fig. 6E-F). As mentioned above, p62 colocalize with COX-IV containing SNX10-positive vesicles and is required for COX-IV turnover (Fig. 7x), suggesting that SNX10 functions in the downstream trafficking of COX-IV for degradation (see model in Fig. 10).

Comment 5: In Fig.4, and related to above point, the authors find multiple mitochondrial components within the SNX10 compartment, yet loss of SNX10 only appears to affect COX-IV. This suggests that co-localization is just a result of mitophagy processes transiting endosomal (and hence SNX10+ve) compartments, rather than any direct involvement of SNX10 in the process. Indeed mitophagy, by reporter assay, is not altered by loss of SNX10. It was not clear if this was the point the authors were trying to make, or something else. Perhaps this could be better explained in the text.

Response: We agree that we did not explain the model very well in the original manuscript. We have now included a model figure (Fig. 10) and discussed it (page 13). Additionally, we

have performed several new experiments to strengthen the model (including using another mitophagy reporter, as described below).

We propose that SNX10 functions as a negative regulator of piecemeal mitophagy of components of the OXPHOS machinery, allowing the cell to regulate oxidative phosphorylation in response to metabolic needs without compromising overall mitochondrial structure. This is based on the following observations:

- i) Endogenous COX-IV, SAMM50, and MitoTracker staining inside SNX10-positive vesicles that are also positive for LC3B, p62 and CD63 upon induction of hypoxia-induced mitophagy using DFP or DMOG (Fig. 3, 4 and 5).
- ii) Increased turnover of endogenous COX-IV (by western blot and immunofluorescence, Fig. 6), as well as a Halo-tagged Su9 (a subunit of the ATP synthase, Fig. 7) in cells lacking SNX10. The levels of Cox-IV and Samm50 are also reduced in zebrafish larvae (Fig. 9).
- iii) SNX10 depletion correlates with reduced respiration, ATP production and citrate synthase activity in cells (Fig. 8), and increased ROS and cell death in zebrafish larvae, which is prevented by the addition of the antioxidant NAC (Fig. 9).
- iv) While SNX10 colocalizes with both EEA1 (early endosomes) and CD63 (late endosomes) in control conditions, there is a shift in the SNX10-positive population to CD63-positive structures containing mitochondrial material and LC3B in cells treated with the hypoxia-mimicking drugs DFP or DMOG. Intriguingly, whereas lysosomal turnover of mitochondrial components increases in SNX10-depleted cells, cells lacking SNX10 show an accumulation of endocytic vesicles and delayed EGFR degradation (Fig. 2), suggesting that SNX10 control cargo trafficking in the endocytic pathway. The exact mechanism underlying this phenotype is not clear, but the presence of enlarged LAMP1-positive vacuoles in cells expressing wild-type SNX10 suggests that SNX10 may regulate fission/fusion events in the endocytic pathway. SNX10 is the smallest of the PX domain proteins, containing only a PX domain plus a short IDR (intrinsically disordered region). The fact that mutations in its PX domain are causative of autosomal recessive osteopetrosis (ARO), a life-threatening rare type of skeletal dysplasia characterized by increased bone density, demonstrates an essential role for this protein.
- v) The fact that SNX10 co-localizes with LC3B and p62 (Fig. 3, 4, and 5), combined with the increased turnover of the inner mitochondrial membrane (IMM) protein pSu9-Halo-EGFP (Fig. 7), with no effect on a mCherry-EGFP-tagged matrix marker (Fig. S3 A-B) in SNX10-depleted cells, indicate that SNX10 functions in piecemeal mitophagy. p62/SQSTM1 is known to be required for piecemeal mitophagy (PMID: 29149599, PMID: 34037656). Indeed, we show that p62 co-localizes with COX-IV containing SNX10-positive vesicles (Fig. 7D) and that the COX-IV level is increased in cells lacking p62. To our surprise, neither the DFP- nor the siSNX10-induced degradation of COX-IV (Fig. 7G-I) or the co-localization of SNX10 with LC3B (Fig. 3E-G) were reduced by inhibition of the core autophagy machinery components ULK1 or VPS34. The two previous papers published on piecemeal mitophagy (PMID: 29149599, PMID: 34037656) did not investigate the need for these core autophagy machinery components, and it is therefore likely that piecemeal mitophagy may be independent of the ULK1 and VPS34 complexes.

The seemingly selective effect of SNX10 on the turnover of proteins involved in mitochondrial oxidative phosphorylation and ATP production is reminiscent of the recently described VDIM (Vesicles Derived from the Inner Mitochondrial membrane) pathway (PMID: 39169179). VDIMS are formed by IMM herniation through pores in the outer mitochondrial membrane, followed by their engulfment by lysosomes in proximity to mitochondria in a microautophagy-like process. VDIM formation was found to increase upon oxidative stress, leading to selective degradation of IMM proteins (including COX4 and other proteins involved in OXPHOS) while sparing the remainder of the organelle. However, in contrast to MitoTracker/COX-IV containing SNX10-positive vesicles, VDIMS seem to lack LC3 and p62. We also show that mitochondria-derived vesicles form independently of SNX10 (Fig. S3C-D).

Comment 6: In Fig.4F, the authors show loss of COX-IV but where is it going? The authors could include bafilomycin and a proteasome inhibitor, as well as another mitochondrial marker as a control. This could help with mechanism as to why only this protein is significantly reduced upon SNX10 loss. If the additional treatments are negative, could COX-IV be being secreted via an MDV/exosome-like pathway?

Response: As suggested by the reviewer, we have now performed several experiments to address COX-IV turnover in control and SNX10-depleted cells. First, we did qPCR to check for COX-IV mRNA levels, showing a light but not significant increase in *COX-IV* expression in cells lacking SNX10 (Fig. S2D). Upon treatment of control and SNX10-depleted cells with the proteasome inhibitor MG132 (Fig. S2E) or Bafilomycin A1 (Fig. 6A and D), we did not see any significant effect on COX-IV levels by western blot analysis. The latter is rather unexpected, as we do see COX-IV staining within LAMP1-positive SNX10 vesicles (Fig. 5D-E), and can be explained by the known effect of BafA1 on induced secretion due to inhibition of autophagosome-lysosome fusion (see e.g. PMID: 7896890). We have also done mass spectrometry analysis of the secretome of control and SNX10-depleted cells under control conditions and did not detect COX-IV in the secretome of SNX10-depleted cells (not shown).

Finally, to test whether SNX10 depletion affects the formation of MDVs (mitochondria-derived vesicles), cells were stained with antibodies against TOMM20 and PHD, followed by imaging and quantification of PDH⁺TOMM20⁻ and PDH⁻TOMM20⁺ structures, demonstrating no significant differences between control and SNX10 depleted cells (Fig. S3 C-D). Thus, based on these experiments, as well as our findings showing colocalization of p62 with COX-IV-containing SNX10 vesicles (Fig. 7D) and an accumulation of COX-IV in p62-depleted cells (Fig. 7 E-F) we conclude that SNX10 functions as a negative regulator of p62-dependent piecemeal mitophagy of COX-IV.

Comment 7: In Fig. 5, the authors see significant effects with one siRNA but not the other. This makes it difficult to draw any firm conclusions. Can the authors knock out SNX10 by CRISPR or rescue effects by re-expression of siRNA resistant forms?

Response: We acknowledge the need to rescue the phenotypes observed in SNX10-depleted cells and we have tried several approaches to do so:

- CRISPR-Cas9 mediated KO of SNX10: after several attempts, where cells kept dying, we managed to obtain one clone (validated through sequencing). However, the cells grew very slowly and we did not observe a phenotype. As mutations in SNX10 are associated with ARO, a disease linked to childhood mortality, it is not surprising that a full knock-out of SNX10 is lethal.
- Next, we generated stable shRNA cells using two different shRNA constructs (SNX10 shRNA #1, SNX10 shRNA #2), with the latter targeting the 3' UTR (to be used for rescue experiments). Both shRNA cell lines showed a 60-80% knockdown of SNX10 expression (see *Reviewer Figure 4* below). We analyzed the total level of COX-IV in these cells by western blot and while we did observe a similar reduction of COX-IV levels (as with SNX10 siRNA) for the first two experiments, this effect was no longer evident in the next experiments, potentially indicating induction of compensatory mechanisms over time or reduced knockdown efficiency. We have therefore chosen to include the data from the first two experiments for the reviewer only (see *Reviewer Figure 4* below). In these experiments, we also transfected cells (control shRNA, SNX10 shRNA #1, SNX10 shRNA #2) with SNX10-EGFP to address a potential rescue effect of SNX10. As can be seen in the *reviewer Figure 4*, the expression level of SNX10-EGFP was too high to observe a knockdown phenotype of SNX10 shRNA #1 (targeting the coding sequence), but it is interesting to note that the level of COX-IV was higher in cells expressing SNX10-EGFP than in non-transfected cells, indicating that SNX10 overexpression has the opposite effect on COX-IV than SNX10 depletion.
- We also tried transient transfection of the shRNA constructs as the 3' UTR shRNA only gives a 60% knockdown efficiency in stable cells. However, upon transient transfection, all cells died, indicating that targeting all SNX10 isoforms is lethal. SNX10 has several isoforms, that in contrast to the canonical isoform used here, do not contain the full PX domain, but share the IDR tail. It makes sense that knockdown of all isoforms could be lethal, as ARO is a serious disease that leads to early life death.

We would like to point out that the two siRNA oligos used (siSNX10 #1 and siSNX10 #2) both show the same trend for all experiments, although only oligo 1 is significant in several experiments. Importantly, both siSNX10 oligos result in higher DFP-induced degradation of the mitophagy reporter pSu9-Halo-mGFP (Fig. 7A-C) and of COX-IV (Fig. 6E-F). We have anyway tried to generate a siRNA-resistant SNX10 construct to make stable cell lines for transfection with siSNX10 #1. We were, however, not able to amplify the mutant siRNA through several attempts.

The fact that we see a reduction of Cox-IV (and Samm50) levels in zebrafish larvae lacking *Snx10ab* (with corresponding increased levels of ROS and cell death) further strengthens our data on the role of SNX10 in the turnover of COX-IV.

Reviewer Fig. 4: U2OS cells stably expressing shCtrl or two different SNX10 shRNAs. Left) The cells were cultured in growth media, lysed in the well and the RNA was extracted before cDNA synthesis and qPCR. The graph shows the differences in mRNA expression of SNX10. The values were normalized to TBP using the 2- $\Delta\Delta$ Ct method and then compared to shCtrl. Right) The cells were cultured in growth media and transfected with SNX10-EGFP (when indicated). 24 hours post-transfection cells were treated with or without DFP (1 μ M) for 24 hours, followed by followed by western blotting for the indicated proteins. The graph shows COX-IV band signal normalized to the shCtrl sample. Data are presented as mean \pm SEM.

Comment 8: In Fig.6D and E, the blots of SNX10 and COXIV are very weak and hard to make out by eye - are there any "long exposures"?

Response: The blots we had in the original submission were indeed long exposures. We have redone the western blots and made new figures to replace the old ones, although the blots look similar to the previous blots (Fig. 9D-G). The SNX10 antibody does not work well for immunoblotting but we see a consistent reduction of Snx10 protein levels in *snx10ab* double KO larva across several experiments.

Comment 9: The increase in cell death in the SNX10 KO larvae is interesting and the authors propose that this is caused by increased ROS. If this is the case, then can cell death be rescued via ROS scavengers/anti-oxidants?

Response: This was a great suggestion by the reviewer. We have repeated the TUNEL staining of control and *snx10ab* DKO zebrafish larvae, treated or not with the anti-oxidant N-acetyl cysteine (NAC). Indeed, we see a rescue of cell death in the brain of the *snx10ab* DKO larvae when compared to control larvae when treated with NAC (Fig. 9K-L), indicating that the cell death seen in *snx10ab* DKO larvae is likely due to increased oxidative stress.

Comment 10: Do the authors think the effect of SNX10 loss on EGFR is related to the mitochondrial phenotype? Some discussion would be helpful.

Response: We do not think there is a direct correlation between the effect of SNX10 loss on reduced EGFR trafficking and increased turnover of COX-IV. Under basal conditions, SNX10

localizes to both early (EEA1 positive) and late (CD63 positive) endosomes, generally devoid of mitochondrial markers. However, upon induction of mitophagy (with DFP or DMOG), we see a shift in the SNX10 localization towards later endosomes containing mitochondrial material and the autophagy markers LC3B and p62, with a corresponding reduced colocalization with EEA1 (Figs. 4 and 5). Thus, we speculate that SNX10, potentially through interacting with different proteins during various metabolic conditions, can function to promote endocytic trafficking of certain cargo while inhibiting other cargo to allow the cell to respond to the metabolic needs of the cells, e.g. during basal versus hypoxia-mimicking conditions as analyzed in this manuscript. We have included some discussion of this (p. 13).

Comment 11: Many of the charts indicate differences but lack statistics.

Response: Thank you for pointing this out. We decided not to display non-significant (ns) data directly on the graphs to avoid overcrowding with too many lines and elements, which could potentially hinder clear visualization and interpretation of the data. We have now explicitly mentioned in each figure legend that non-significant differences are not depicted on the graphs.

Comment 12: Outside of the initial mitophagy analyses, I found the use of DFP and DFO a little confusing as there was no differences between controls and SNX10 depletions. For simplicity, the authors could consider removing these data.

Response: Thank you for pointing this out. We initially considered removing the mitophagy assays conducted with the IMLS cell line (expressing the matrix reporter NIPSNAP¹⁻⁵³-mCherry-EGFP), showing no effect of SNX10 depletion of the turnover of this matrix reporter (Fig. S3A-B). However, we decided to retain the IMLS data given our new data from the pSu9-Halo-mGFP reporter cell line (expressing a subunit of the ATP synthase), showing increased lysosomal degradation of this inner mitochondrial membrane protein in SNX10-depleted cells, in response to DFP or DMOG (Fig. 7A-C). We believe these results further reinforce our conclusions about the specificity of mitophagy under varying conditions, highlighting that different cargoes may be degraded depending on the metabolic status of the cell.

Reviewer #3 comments

Point 1: Figure 1 shows that SNX10 localises to early and late endosomes in a PtdIns3P-dependent manner. Point 1. It is possible that vacuoles also form in SNX10 ARO mutant expressing cells but are not visible due to the low expression level of GFP (or that they form but the GFP-SNX10 does not localise there). Suggest rewriting the interpretation of the data for accuracy. Can the vacuoles be observed using another marker in the mutant-expressing cells? To me, this is a minor point and I think removing the statement about vacuoles not forming in the mutant cells would suffice.

Response: Thank you for pointing out the possibility that vacuoles may form in SNX10 ARO mutant-expressing cells that are not visible due to low GFP expression levels or that mutant SNX10-EGFP does not localize to vacuoles. To address this, we did brightfield imaging of cells expressing wild-type or ARO mutant SNX10-EGFP. This demonstrates that while such vacuoles

are clearly evident in the SNX10 WT overexpressing cell lines, they are absent in the ARO mutant cells (Fig. S1C). The formation of vacuoles upon SNX10 overexpression has been previously documented by Qin et al. (2006), who demonstrated that these vacuoles appear unique to SNX10 compared to other sorting nexins analyzed. We have anyway toned down this point in the discussion (p.12).

Point 2: Including quantification representing the colocalization of SNX10 with various Rab proteins would improve the robustness of the paper.

Response: We have now carried out several new experiments to further characterize the localization of SNX10 in the endocytic pathway under control conditions and upon hypoxia-mimicking conditions. Instead of quantifying the colocalization of SNX10 with stably overexpressed Rab proteins, we decided to quantify the colocalization of SNX10 with endogenous markers of early (EEA1) and late (CD63) endocytic structures. Stable U2OS SNX10-EGFP cells were fixed and stained with antibodies against endogenous EEA1 and CD63, followed by imaging using a Nikon CREST X-Light V3 microscope. Quantification of colocalization was performed by segmentation of fluorescent signals using CellProfiler to obtain the percentage of CD63 or EEA1 structures that overlap with SNX10. When plotting the ratio of CD63 or EEA1 structures overlapping with SNX10 to the total number of SNX10-positive structures (Fig. 1 F-G), we find that SNX10 colocalizes with EEA1 and CD63 to a similar extent under basal conditions. Intriguingly, there is a shift in the SNX10-positive population to CD63-positive structures containing mitochondrial material and autophagy markers (p62 and LC3B) in cells treated with the hypoxia-mimicking drugs DFP and DMOG (Figs. 4 and 5, see also our response to point 3 below).

Point 3: Figure 2 explores SNX10's role in endosomal trafficking, however, the supporting text did not clearly articulate the role of SNX10 in this process. Point 3. Clarify the accompanying text in the results section for the role of SNX10 in endosomal maturation. I am not convinced that the data presented is conclusive to demonstrate a role for SNX10 in endosomal maturation or trafficking since I would think that this should include a more thorough characterisation of hallmarks of endosomal maturation such as Rab switch, lipid status and acidification status. Can the authors provide more data to support their statements? Alternatively, moderate the interpretation of the data.

Response: We agree with the reviewer that the data presented in the original manuscript was not sufficient to state that SNX10 plays a role in endosomal maturation, a term we may have used incorrectly. Our data demonstrate a key role for SNX10 in cargo trafficking within the endolysosomal pathway, but we have no data to support a direct role for SNX10 in the regulation of an endosomal Rab switch, altered lipid, or acidification status. We have therefore refrained from using the term endosomal maturation.

During the revision, we have carried out several new experiments to further understand the role of SNX10 in the endolysosomal pathway. While SNX10 colocalizes with both EEA1 and CD63 positive vesicles in control conditions (Fig. 1F-G), we observe a shift in the SNX10-positive population to CD63 positive structures containing mitochondrial material (MitoTracker and COX-IV) and autophagy markers (p62 and LC3B) upon induction of mitophagy (Figs. 4-5). In line with this, the co-localization of SNX10 with RAB7 is higher in DFP-treated cells compared to control conditions (see *Reviewer Figure 1* above). Importantly,

SNX10-depleted cells and zebrafish larvae show an increased turnover of mitochondrial material (Fig. 6-7 and 9), whereas SNX10-depleted cells are characterized by delayed EGFR degradation and an accumulation of endocytic vesicles (Fig. 2), suggesting that SNX10 may control cargo trafficking in the endocytic pathway. The exact mechanism underlying this phenotype is not clear, but the presence of enlarged LAMP1-positive vacuoles in cells expressing wild-type SNX10 (Fig. 1) suggests that SNX10 may regulate fission/fusion events in the endocytic pathway. SNX10 is the smallest of the PX domain proteins, containing only the PX domain plus a short IDR (intrinsically disordered region). The fact that mutations in its PX domain are causative of autosomal recessive osteopetrosis (ARO), a life-threatening rare type of skeletal dysplasia characterized by increased bone density, demonstrates an essential role for this protein.

We propose that SNX10 functions as a negative regulator of piecemeal mitophagy of components of the OXPHOS machinery, allowing the cell to regulate oxidative phosphorylation in response to metabolic needs without compromising overall mitochondrial structure.

Point 4: F WB figure 2 C-D. The statistical test is mentioned in the figure legend, but the result is missing from the figure. If non-sig, include on the graph.

Response: Thank you for pointing this out. We decided not to display non-significant data directly on the graphs to avoid overcrowding with too many lines and elements, which could potentially hinder clear visualization and interpretation. We appreciate your feedback, as it has helped us improve the clarity of our manuscript. In response, we have thoroughly clarified in each figure legend that non-significant differences are not depicted on the graphs.

Point 5: Please include an image analysis section in the methods section describing the quantification of the EEA1 for this graph (applies to some other graphs as well).

Response: We have now included an image analysis section in the methods section describing the image quantification results (for EEA1 and other graphs).

Point 6: Regarding the EM analysis, I was confused about matching the figure legend text with the graph since the gold-negative and positive endosomes were not separated on the graph. Please clarify.

Response: We thank the reviewer for pointing out the slightly confusing figure legend text in Figs. 2G-H. We have now updated the figure legend to clarify what was measured in this experiment. The legend to Fig. 2G-H now reads “**G**) Representative electron microscopy images of endosomes in U2OS cells (control and siSNX10 #1). Pink arrows: Protein A conjugated with 10 nm gold (PAG10) labeling EGFR that has been taken up into endosomes, Yellow arrows: endosome not containing internalized PAG10-labeled EGFR. **H**) Measurements of EGFR-containing endosome diameter in control vs siSNX10 treated cells from one experiment. The graph shows the endosomal diameter (nm) of a total 24 PAG10-labeled EGFR endosomes in siCtrl cells and 15 PAG10-labeled EGFR endosomes in siSNX10 cells. The graph displays the mean values \pm SEM. Significance was determined by unpaired t-test with Welch’s correction in all graphs. * = $p < 0.05$, ** = $p < 0.01$, non-significant differences are not depicted.” Thus, only the EGFR gold-containing endosomes were quantified.

Point 7: Supplementary Figure 2. Suggest rewriting the figure legend of A for clarity - which SNX10 sirna is used to measure ulk1 levels? Or is it the ULK1 levels being measured in the ULK1 column?

Response: We appreciate the reviewer pointing out the lack of clarity in the figure legend. As part of the revision process, we have performed qPCR experiments for several mitochondrial transcripts as well as for ULK1, under conditions of SNX10 and ULK1 knockdown (Fig. S2D). As the original graph (Fig. S2A) was redundant with our new qPCR data (Fig. S2D), we opted to remove the original graph to avoid repetition and streamline the presentation of the data.

Point 8: 3. Figure 3 highlights the interactome of SNX10. Point 8. A more comprehensive description of the quantification of the mass spec is required to interpret the proteomic data. Was this done in triplicate? How was the control used to remove background proteins? Was the control GFP alone or the disease-associated mutant SNX10 - the figure suggests it is the mutant SNX10 but the text suggests it is GFP alone. Was there any statistical analysis performed to demonstrate that SNX10 interacts with mito proteins?

Response: Thank you for drawing our attention to the gaps in our original explanation of the proteomic analysis. We realized that it was not adequately described and have now provided a more detailed explanation in the Methods (p. 21-23) sections of the revised manuscript.

We have clarified that the mass spectrometry experiments were performed in triplicate, and the control used to eliminate background proteins was GFP alone, as this allowed us to specifically distinguish SNX10 interactions from non-specific background binding. Furthermore, we have elaborated on the statistical analyses conducted to demonstrate the significance of SNX10's interactions with mitochondrial proteins. Additionally, to enhance transparency and reproducibility, the mass spectrometry proteomics data have been deposited to the ProteomeXchange Consortium via the PRIDE⁵⁶ partner repository with the dataset identifier PXD056720.

Point 9: The conclusion that SNX10 "interact" with mitochondria should be tempered. The data showing that SNX10 and mitochondria are near each other is not conclusive given the widespread localisation of mitochondria throughout the cytoplasm. Suggest changing "interact" to localise nearby.

Response: we have now modified the text to read "SNX10-EGFP-positive structures were found to localize near the mitochondrial network and move along mitochondria in a highly dynamic manner" (p. 7) and the Figure 3 legend title to read "SNX10 localizes nearby mitochondria" (p. 32).

Point 10: Supplementary figure 2B. Do SNX10 protein levels change with DFP?

Response: While we observe a two-fold increase in SNX10 transcript levels in DFP-treated cells, it has been challenging to establish whether this corresponds to higher SNX10 protein levels. Despite extensive efforts, using three different antibodies from various vendors (Atlas Antibodies, Santa Cruz Biotechnologies, and Novus Biologicals) and thorough optimization, we

have only been able to detect endogenous SNX10 on a few occasions using the Atlas antibody, but subsequent batches did not work, even after contacting the company. Therefore, we cannot definitively determine if SNX10 protein levels increase with DFP treatment. Intriguingly, we do observe a higher intensity of SNX10-EGFP when treating cells with DFP, which is puzzling considering that the plasmid construct does not share the same promoter as endogenous SNX10, suggesting either reduced SNX10 degradation or increased membrane recruitment of SNX10.

Point 11: The text at the end of this section describing figure 3 "Taken together, our data suggest a role for SNX10 in mitophagy" is confusing to me since there is no direct evidence for this presented in the figure (i.e. would need to deplete SNX10 and show an increase or decrease in mitophagy). The increased LC3-SNX10 colocalisation could be due to the increased LC3 in the DFP condition. Data suggests that SNX10 localises to the mitophagosome however there is no data to suggest it functions there.

Response: We agree that we did not explain the model very well in the original manuscript. We have now included a model figure (Fig. 10) based on several new experiments that strengthen the model (including using another mitophagy reporter, as described below).

We propose that SNX10 functions as a negative regulator of piecemeal mitophagy of components of the OXPHOS machinery, allowing the cell to regulate oxidative phosphorylation in response to metabolic needs without compromising overall mitochondrial structure. This is based on the following observations:

- i) Endogenous COX-IV, SAMM50, and MitoTracker staining inside SNX10-positive vesicles that are also positive for LC3B, p62 and CD63 upon induction of hypoxia-induced mitophagy using DFP or DMOG (Fig. 3, 4 and 5).
- ii) Increased turnover of endogenous COX-IV (by western blot and immunofluorescence, Fig. 6), as well as a Halo-tagged Su9 (a subunit of the ATP synthase, Fig. 7) in cells lacking SNX10. The levels of Cox-IV and Samm50 are also reduced in zebrafish larvae (Fig. 9).
- iii) SNX10 depletion correlates with reduced respiration, ATP production and citrate synthase activity in cells (Fig. 8), and increased ROS and cell death in zebrafish larvae, which is prevented by the addition of the antioxidant NAC (Fig. 9).
- iv) While SNX10 colocalizes with both EEA1 (early endosomes) and CD63 (late endosomes) in control conditions, there is a shift in the SNX10-positive population to CD63-positive structures containing mitochondrial material and LC3B in cells treated with the hypoxia-mimicking drugs DFP or DMOG. Intriguingly, whereas lysosomal turnover of mitochondrial components increases in SNX10-depleted cells, cells lacking SNX10 show an accumulation of endocytic vesicles and delayed EGFR degradation (Fig. 2), suggesting that SNX10 control cargo trafficking in the endocytic pathway. The exact mechanism underlying this phenotype is not clear, but the presence of enlarged LAMP1-positive vacuoles in cells expressing wild-type SNX10 (Fig. 1) suggests that SNX10 may regulate fission/fusion events in the endocytic pathway. SNX10 is the smallest of the PX domain proteins, containing only a PX domain plus a short IDR (intrinsically disordered region). The fact that mutations in its PX domain are causative of autosomal recessive osteopetrosis

(ARO), a life-threatening rare type of skeletal dysplasia characterized by increased bone density, demonstrates an essential role for this protein.

- v) The fact that SNX10 co-localizes with LC3B and p62 (Fig. 3, 4, and 5), combined with the increased turnover of the inner mitochondrial membrane (IMM) protein pSu9-Halo-EGFP (Fig. 7), with no effect on a mCherry-EGFP-tagged matrix marker (Fig. S3 A-B) in SNX10-depleted cells, indicate that SNX10 functions in piecemeal mitophagy. p62/SQSTM1 is known to be required for piecemeal mitophagy (PMID: 29149599, PMID: 34037656). Indeed, we show that p62 co-localizes with COX-IV containing SNX10-positive vesicles (Fig. 7D) and that the COX-IV level is significantly increased in cells lacking p62. To our surprise, neither the DFP- nor the siSNX10-induced degradation of COX-IV (Fig. 7G-I) or the co-localization of SNX10 with LC3B were reduced by inhibition of the core autophagy machinery components ULK1 or VPS34. The two previous papers published on piecemeal mitophagy (PMID: 29149599, PMID: 34037656) did not investigate the need for these core autophagy machinery components, and it is therefore likely that piecemeal mitophagy may be independent of the ULK1 and VPS34 complexes.

The seemingly selective effect of SNX10 on the turnover of proteins involved in mitochondrial oxidative phosphorylation and ATP production is reminiscent of the recently described VDIM (Vesicles Derived from the Inner Mitochondrial membrane) pathway (PMID: 39169179). VDIMS are formed by IMM herniation through pores in the outer mitochondrial membrane, followed by their engulfment by lysosomes in proximity to mitochondria in a microautophagy-like process. VDIM formation was found to increase upon oxidative stress, leading to selective degradation of IMM proteins (including COX4 and other proteins involved in OXPHOS) while sparing the remainder of the organelle. However, in contrast to MitoTracker/COX-IV containing SNX10-positive vesicles, VDIMS seem to lack LC3 and p62. We also show that mitochondria-derived vesicles form independently of SNX10 (Fig. S3C-D).

Point 12: Figure 4 describes the effect of SNX10 depletion on levels of mitochondrial proteins in steady state and after DFP. Point 12. In my opinion, although SNX10 may indeed localise to structures that contain mitochondria, the functional depletion of SNX10 does not significantly change mitochondrial protein abundance. The WB for COXIV does not reflect the IF images presented. Is this the most representative WB to show? Is there another exposure?

Response: We have now strengthened our data showing an effect of SNX10 depletion on COX-IV levels. We demonstrate by western blot analysis (Fig. 6A and D) and immunofluorescence microscopy analysis (Fig. 6E-F and Fig. 7G-I) that the level of endogenous COX-IV is reduced in SNX10-depleted cells compared to control cells. Importantly, the turnover of the inner mitochondrial membrane (IMM) ATP synthase subunit Su9 (pSu9-Halo-EGFP) is also increased in cells lacking SNX10 (using two different oligos (Fig. 7A-C) and shRNA constructs (see *reviewer Fig. 4* above). Moreover, the levels of Cox-IV and Sams50 are reduced in zebrafish larvae upon knock-out of *snx10ab* (Fig. 9).

Point 13: Supplementary Figure 3A. line intensity scan - specify if the left or right Y axis is green or magenta pixel intensity.

Response: Thank you for pointing this out. Our new Fig. 5B-C includes updated line plots of the same, and we have therefore removed the original Fig. S3A. The new plots depict the co-occurrence of SNX10, LC3, and COX-IV with the Y axis representing the pixel intensity for all. We have provided this data across all conditions (Control, DFP, and DMOG).

Point 14: Supplementary Figure 3C-D. Include a description in methods section or figure legend about how mitolysosomes were measured - is it the magenta only signal or a ratiometric analysis. Stats mentioned in figure legend but not on the graph. If non-sig, include on the graph.

Response: We acknowledge that our previous description regarding the measurement of mitolysosomes (now shown in Fig. S3 A-B) was vague. We have now included a dedicated section in the Methods entitled "Image Analysis", providing details on how mitolysosomes were measured. Additionally, we have added clarifying statements to each figure legend where non-significant results are mentioned. We chose not to depict non-significant data directly on the graphs, as including too many lines and elements could make visualization and interpretation more challenging.

Point 15: 5. Figure 5 looks at the depletion of SNX10 and mitochondrial function. Point 15. Since only one siRNA appears to have a significant effect (that is my interpretation from the graph but I may be wrong), I believe it is especially important to rescue this phenotype using the si-resistant construct or demonstrate with another siRNA. To understand the disease-relevance of the pathway, the authors could rescue with the disease-associated mutants.

Response: We acknowledge the need to rescue the phenotypes observed in SNX10-depleted cells and we have tried several approaches to do so:

- CRISPR-Cas9 mediated KO of SNX10: after several attempts, where cells kept dying, we managed to obtain one clone (validated through sequencing). However, the cells grew very slowly, and we did not observe a phenotype. As mutations in SNX10 are associated with ARO, a disease linked to childhood mortality, it is not surprising that a full knock-out of SNX10 is lethal.
- Next, we generated stable shRNA cells using two different shRNA constructs (SNX10 shRNA #1, SNX10 shRNA #2), with the latter targeting the 3' UTR (to be used for rescue experiments). Both shRNA cell lines showed a 60-80% knockdown of SNX10 expression (see Reviewer Fig. 4A above). We analyzed the total level of COX-IV in these cells by western blot and while we did observe a similar reduction of COX-IV levels (as with SNX10 siRNA) for the first two experiments, this effect was no longer evident in the next experiments, potentially indicating induction of compensatory mechanisms over time or reduced knockdown efficiency. We have therefore chosen to include the data from the first two experiments for the reviewer only (see *Reviewer Fig. 4* above). In these experiments, we also transfected cells (control shRNA, SNX10 shRNA #1, SNX10 shRNA #2) with SNX10-EGFP to address a potential rescue effect of SNX10. As can be seen in the *reviewer Fig. 4*, the expression level of SNX10-EGFP was too high to observe a knockdown phenotype of SNX10 shRNA #1 (targeting the coding sequence), but it is interesting to note that the level of COX-IV was higher in cells expressing SNX10-EGFP

than in non-transfected cells, indicating that SNX10 overexpression has the opposite effect on COX-IV than SNX10 depletion.

- We also tried transient transfection of the shRNA constructs as the 3' UTR shRNA only gives a 60% knockdown efficiency in stable cells. However, upon transient transfection, all cells died, indicating that targeting all SNX10 isoforms is lethal. SNX10 has several isoforms, that in contrast to the canonical isoform used here, do not contain the full PX domain, but share the IDR tail. It makes sense that knockdown of all isoforms could be lethal, as ARO is a serious disease that leads to early life death.
- We would like to point out that the two siRNA oligos used (siSNX10 #1 and siSNX10 #2) both show the same trend for all experiments, although only oligo 1 is significant in several experiments. Importantly, both siSNX10 oligos result in higher DFP-induced degradation of the mitophagy reporter pSu9-Halo-mGFP (Fig. 7A-C) and of COX-IV (Fig. 6E-F). We have anyway tried to generate a siRNA-resistant SNX10 construct to make stable cell lines for transfection with siSNX10 #1. We were, however, not able to amplify the mutant siRNA through several attempts.
- The fact that we see a reduction of Cox-IV (and Samm50) levels in zebrafish larvae lacking *Snx10ab*, with corresponding increased levels of ROS and cell death (Fig. 9) further strengthens our data on the role of SNX10 in the turnover of COX-IV.

Point 16: Figure 6 - zebrafish system. Point 16. The interpretation that SNX10 mediated degradation of inner membrane proteins is not demonstrated by the data presented, therefore the text should be rewritten. It could be through transcription, loss of biogenesis etc. Discussion point: is osteopetrosis seen in mito disease?

Response: We agree that the reduced Cox4 and Samm50 protein levels observed in *Snx10ab* DKO larvae potentially could be due to a reduction of *cox4* and *samm50* transcripts or reduced mitochondrial biogenesis. We have now performed qPCR analysis from control or *Snx10ab* DKO zebrafish larvae RNA, showing a slight, but not significant decrease of *cox4* and *samm50* transcript levels in the *Snx10ab* DKO larvae (Fig. S4D). In line with this, transcript levels of several mitochondrial genes (TIMM23, TOMM20, COX4, BNIP3, NIX, HIF1a) were either unaffected or increased in cells lacking SNX10 or ULK1 (Fig. S2D). The fact that the overall levels of several mitochondrial proteins (including Citrate synthase, TOMM20) are unaffected in SNX10-depleted cells further indicates that SNX10 depletion does not affect mitochondrial biogenesis.

We thank the reviewer for raising the question about a potential link between osteopetrosis and mitochondria disease. Mitochondria are crucial for the regulation of nutrient metabolism and maintenance of bone homeostasis. Indeed, mitochondrial dysfunction (including impaired mitochondrial autophagy and OXPHOS activity, and ROS accumulation) has been linked to osteoporosis (PMID: 38370357), but currently, only a limited number of studies have investigated the association between osteopetrosis and mitochondrial dysfunction. We have now included a few sentences about this in the discussion (p. 13).

January 23, 2025

RE: JCB Manuscript #202404009R

Anne Simonsen
Oslo University Hospital

Dear Prof. Simonsen:

Thank you for submitting your revised manuscript entitled "SNX10 functions as a novel modulator of piecemeal mitophagy and mitochondrial bioenergetics". We would be happy to publish your paper in JCB pending final revisions necessary to meet our formatting guidelines (see details below). In your final revision, please be sure to address the reviewers' final minor comments.

A. MANUSCRIPT ORGANIZATION AND FORMATTING:

- 1) Text limits: Character count for Articles is < 40,000, not including spaces. Count includes abstract, introduction, results, discussion, and acknowledgments. Count does not include title page, figure legends, materials and methods, references, tables, or supplemental legends.
- 2) Figures limits: Articles may have up to 10 main text figures.
- 3) Figure formatting: Scale bars must be present on all microscopy images, including inset magnifications (you may alternatively indicate the diameter of the inset). Molecular weight or nucleic acid size markers must be included on all gel electrophoresis. Aspect ratios of images may not be altered.
- 4) Statistical analysis: Error bars on graphic representations of numerical data must be clearly described in the figure legend. The number of independent data points (n) represented in a graph must be indicated in the legend. Statistical methods should be explained in full in the materials and methods. For figures presenting pooled data the statistical measure should be defined in the figure legends. Please also be sure to indicate the statistical tests used in each of your experiments (either in the figure legend itself or in a separate methods section) as well as the parameters of the test (for example, if you ran a t-test, please indicate if it was one- or two-sided, etc.). Also, if you used parametric tests, please indicate if the data distribution was tested for normality (and if so, how). If not, you must state something to the effect that "Data distribution was assumed to be normal but this was not formally tested."
- 5) Abstract and title: The abstract should be no longer than 160 words and should communicate the significance of the paper for a general audience. The title should be less than 100 characters including spaces. Make the title concise but accessible to a general readership.

* The term 'novel' may not be used in the title as per JCB policy *
- 6) Materials and methods: Should be comprehensive and not simply reference a previous publication for details on how an experiment was performed. Please provide full descriptions in the text for readers who may not have access to referenced manuscripts.
- 7) All antibodies, cell lines, animals, and tools used in the manuscript should be described in full, including accession numbers for materials available in a public repository such as the Resource Identification Portal. Please be sure to provide the sequences for all of your primers/oligos and RNAi constructs in the materials and methods. You must also indicate in the methods the source, species, and catalog numbers (where appropriate) for all of your antibodies. Please also indicate the acquisition and quantification methods for immunoblotting/western blots.
- 8) Microscope image acquisition: The following information must be provided about the acquisition and processing of images:
 - a. Make and model of microscope
 - b. Type, magnification, and numerical aperture of the objective lenses
 - c. Temperature
 - d. Imaging medium
 - e. Fluorochromes
 - f. Camera make and model

g. Acquisition software

h. Any software used for image processing subsequent to data acquisition. Please include details and types of operations involved (e.g., type of deconvolution, 3D reconstitutions, surface or volume rendering, gamma adjustments, etc.).

10) Supplemental materials: There are strict limits on the allowable amount of supplemental data. Articles may have up to 5 supplemental figures. Please also note that tables, like figures, should be provided as individual, editable files. A summary of all supplemental material should appear at the end of the Materials and methods section.

13) ORCID IDs: ORCID IDs are unique identifiers allowing researchers to create a record of their various scholarly contributions in a single place. Please note that ORCID IDs are now *required* for all authors. At resubmission of your final files, please be sure to provide your ORCID ID and those of all co-authors.

Please note that JCB now requires authors to submit Source Data used to generate figures containing gels and Western blots with all revised manuscripts. This Source Data consists of fully uncropped and unprocessed images for each gel/blot displayed in the main and supplemental figures. Since your paper includes cropped gel and/or blot images, please be sure to provide one Source Data file for each figure that contains gels and/or blots along with your revised manuscript files. File names for Source Data figures should be alphanumeric without any spaces or special characters (i.e., SourceDataF#, where F# refers to the associated main figure number or SourceDataFS# for those associated with Supplementary figures). The lanes of the gels/blots should be labeled as they are in the associated figure, the place where cropping was applied should be marked (with a box), and molecular weight/size standards should be labeled wherever possible.

Journal of Cell Biology now requires a data availability statement for all research article submissions. These statements will be published in the article directly above the Acknowledgments. The statement should address all data underlying the research presented in the manuscript. Please visit the JCB instructions for authors for guidelines and examples of statements at (<https://rupress.org/jcb/pages/editorial-policies#data-availability-statement>).

B. FINAL FILES:

**It is JCB policy that if requested, original data images must be made available to the editors. Failure to provide original images upon request will result in unavoidable delays in publication. Please ensure that you have access to all original data images prior

to final submission.**

Thank you for your attention to these final processing requirements. Please revise and format the manuscript and upload materials within 7 days. If you need an extension for whatever reason, please let us know and we can work with you to determine a suitable revision period.

Thank you for this interesting contribution, we look forward to publishing your paper in Journal of Cell Biology.

Sincerely,

Michael Lazarou, PhD
Monitoring Editor

Andrea L. Marat, PhD
Deputy Editor

Journal of Cell Biology

Reviewer #1 (Comments to the Authors (Required)):

This is a resubmission of the manuscript "SNX10 functions as a novel modulator of piecemeal mitophagy and mitochondrial bioenergetics". The authors have thoroughly addressed all of my questions, and have done so in an impressive manner. This was an interesting manuscript upon submission, and I feel the authors have used the revision to substantially elevate the study, which I find to be highly rigorous, thought provoking, and timely. I particularly appreciate the authors' thoughtful additions to their manuscript, including the use of additional hypoxia-inducing drugs throughout their study, the use of a second mitophagy reporter (which showed unexpected but very interesting behaviors), and the in-depth exploration of the contributions of p62 to the SNX10 phenotype.

I have a few minor suggestions for the authors, but these should not preclude acceptance and publication of this manuscript.

Minor comments:

1. Upon re-reading the paper, I had questions about the statistical analysis presented in Figure 3A. I see that Reviewer #3 also had questions on this (point 8), and the authors have addressed this in their methods section. However, I feel as though clarifying some details in the text and the figure legend is warranted, as the authors use language such as "A total of 53 proteins were identified as significant SNX10-EGFP interactors", but no language can be found in the main text or figure legends to describe how this statistical analysis was done.
2. Lines 266-268 read "...our data indicate a role for SNX10 as a negative regulator of lysosomal turnover of selected mitochondrial membrane components in response to HIF1a activation, with no effect on the turnover of matrix localized proteins." The latter statement is likely an overstatement; just because the matrix-localized mitophagy reporter did not turn over does not mean that SNX10 has no effect on any matrix localized proteins. This could be easily modified to focus on the apparent selectivity without the "no effect on the turnover of matrix-localized proteins".
3. The authors describe a reduction in CS activity without reduction in CS levels (lines 302-303), which suggest that CS is not a SNX10 target protein. This seems like a missed opportunity to reiterate the selectivity of SNX10 targets, however, I will leave it up to the authors whether they agree and want to mention this here.

Reviewer #2 (Comments to the Authors (Required)):

The authors have satisfactorily addressed my concerns and congratulations on an intriguing story!

Reviewer #3 (Comments to the Authors (Required)):

The authors have thoughtfully and adequately addressed my reviewer comments, resulting in a much-improved manuscript. During the proofreading stage, I recommend revising lines 291 and 293 to enhance clarity.

2nd Revision - Authors' Response to Reviewers: January 31, 2025

UiO : **CanCell – Senter for kreftcelle-reprogramming**
Universitetet i Oslo

To
Journal of Cell Biology editors

Michael Lazarou, Monitoring Editor
Andrea L. Marat, Senior Scientific Editor

Professor Anne Simonsen
Institute of Clinical Medicine
Division of Cancer Medicine
0317 Oslo, Norway
Phone: +47 99350960
anne.simonsen@medisin.uio.no

Date: 31 January 2025

RE: *JCB Manuscript #202404009R*

Dear Michael and Andrea,

Thank you for the provisional acceptance of our manuscript "SNX10 functions as a novel modulator of piecemeal mitophagy and mitochondrial bioenergetics".

We have now addressed the final minor comments from Reviewer #1 and Reviewer #3 (see our response below). We have also revised the manuscript to adhere to the *JCB* formatting guidelines and have uploaded all requested files to the online submission system.

As the term 'novel' may not be used in the title as per *JCB* policy, we have changed the title of the manuscript to "SNX10 functions as a modulator of piecemeal mitophagy and mitochondrial bioenergetics".

Sincerely,

Prof. Anne Simonsen, PI and co-director of CanCell

Department of Molecular Cell Biology, Institute for Cancer Research, Oslo University Hospital and Centre for Cancer Cell Reprogramming, Institute for Clinical Medicine, University of Oslo

Our response to the Reviewer's comments:**Reviewer #1**

This is a resubmission of the manuscript "SNX10 functions as a novel modulator of piecemeal mitophagy and mitochondrial bioenergetics". The authors have thoroughly addressed all of my questions, and have done so in an impressive manner. This was an interesting manuscript upon submission, and I feel the authors have used the revision to substantially elevate the study, which I find to be highly rigorous, thought provoking, and timely. I particularly appreciate the authors' thoughtful additions to their manuscript, including the use of additional hypoxia-inducing drugs throughout their study, the use of a second mitophagy reporter (which showed unexpected but very interesting behaviors), and the in-depth exploration of the contributions of p62 to the SNX10 phenotype. I have a few minor suggestions for the authors, but these should not preclude acceptance and publication of this manuscript.

Minor comments:

1. Upon re-reading the paper, I had questions about the statistical analysis presented in Figure 3A. I see that Reviewer #3 also had questions on this (point 8), and the authors have addressed this in their methods section. However, I feel as though clarifying some details in the text and the figure legend is warranted, as the authors use language such as "A total of 53 proteins were identified as significant SNX10-EGFP interactors", but no language can be found in the main text or figure legends to describe how this statistical analysis was done.

Response: We have now included information in the text and legend to Fig. 3A about the cut-off used for statistical analysis of the significant candidates from the pulldown of SNX10 wild type and mutant shown in Fig. 3A. The text reads: "A total of 53 proteins were identified as significant SNX10-EGFP interactors compared to the EGFP control, as analyzed by R lima using a cut-off value of $\log_2FC > 1$ and adjusted P-value < 0.05 (Fig. 3 A)."

2. Lines 266-268 read "...our data indicate a role for SNX10 as a negative regulator of lysosomal turnover of selected mitochondrial membrane components in response to HIF1a activation, with no effect on the turnover of matrix localized proteins." The latter statement is likely an overstatement; just because the matrix-localized mitophagy reporter did not turn over does not mean that SNX10 has no effect on any matrix localized proteins. This could be easily modified to focus on the apparent selectivity without the "no effect on the turnover of matrix-localized proteins".

Response: We agree that this is an overstatement and have now changed the text to read: "Thus, our data indicate a role for SNX10 as a negative regulator of lysosomal turnover of selected mitochondrial membrane components in response to HIF1 activation.", as suggested by the reviewer.

3. The authors describe a reduction in CS activity without reduction in CS levels (lines 302-303), which suggest that CS is not a SNX10 target protein. This seems like a missed opportunity to reiterate the selectivity of SNX10 targets, however, I will leave it up to the authors whether they agree and want to mention this here.

Response: We thank the reviewer for pointing out this. We have now modified the sentence to read: "However, the reduction in CS activity did not correlate with reduced CS levels as determined by Western blotting (Fig. 8, E and F), suggesting a lower activity of the Krebs cycle and further demonstrating the selectivity of SNX10 in the degradation of mitochondrial proteins."

Reviewer #3:

The authors have thoughtfully and adequately addressed my reviewer comments, resulting in a much-improved manuscript. During the proofreading stage, I recommend revising lines 291 and 293 to enhance clarity.

Response: We agree and have now modified the text to enhance clarity. The text now reads: "In line with this, the DFP-induced co-occurrence of LC3B with SNX10 and SNX10/Mitotracker-positive structures was not affected in cells treated with the ULK1 inhibitor compared to control cells (Fig. 3, E-G). Taken together, our data indicate that SNX10 modulation of COX-IV turnover is independent of canonical macro-autophagy (Fig. 10)."